# Defective dystrophic thymus determines degenerative changes in skeletal muscle

Andrea Farini[1,8], Clementina Sitzia[2,8], Chiara Villa[1], Barbara Cassani[3,4], Luana Tripodi[1], Mariella Legato[1], Marzia Belicchi[1], Pamela Bella[1], Caterina Lonati[5], Stefano Gatti [5], Massimiliano Cerletti[6,7] & Yvan Torrente [1✉]

In Duchenne muscular dystrophy (DMD), sarcolemma fragility and myofiber necrosis produce cellular debris that attract inflammatory cells. Macrophages and T-lymphocytes infiltrate muscles in response to damage-associated molecular pattern signalling and the release of TNF-α, TGF-β and interleukins prevent skeletal muscle improvement from the inflammation. This immunological scenario was extended by the discovery of a specific response to muscle antigens and a role for regulatory T cells (Tregs) in muscle regeneration. Normally, autoimmunity is avoided by autoreactive T-lymphocyte deletion within thymus, while in the periphery Tregs monitor effector T-cells escaping from central regulatory control. Here, we report impairment of thymus architecture of mdx mice together with decreased expression of ghrelin, autophagy dysfunction and AIRE down-regulation. Transplantation of dystrophic thymus in recipient nude mice determine the up-regulation of inflammatory/ fibrotic markers, marked metabolic breakdown that leads to muscle atrophy and loss of force. These results indicate that involution of dystrophic thymus exacerbates muscular dystrophy by altering central immune tolerance.

[1] Stem Cell Laboratory, Department of Pathophysiology and Transplantation, Università degli Studi di Milano, Unit of Neurology, Fondazione IRCCS Cà Granda Ospedale Maggiore Policlinico, Centro Dino Ferrari, Milan, Italy. [2] Residency Program in Clinical Pathology and Clinical Biochemistry, Università degli Studi di Milano, Milan, Italy. [3] Consiglio Nazionale delle Ricerche-Istituto di Ricerca Genetica e Biomedica (CNR-IRGB), Milan Unit, Milan, Italy. [4] IRCCS Humanitas clinical and research center, Rozzano 20089, Milan, Italy. [5] Center for Surgical Research, Fondazione IRCCS Cà Granda, Ospedale Maggiore Policlinico, Milan, Italy. [6] UCL Research Department for Surgical Biotechnology, University College London, London, UK. [7] UCL Institute for Immunity and Transplantation, University College London, London, UK. [8] These authors contributed equally: Andrea Farini, Clementina Sitzia. ✉email: yvan.torrente@unimi.it

The study of innate and adaptive immune response involvement in muscular dystrophies (MDs) has been attracting the interest of many researchers though the results are so far barely exhaustive and sometimes contradictory. The recruitment of T cells into injured muscle implies an adaptive immune response, which normally depends on antigen exposure. While this mechanism appears conceivable in inflammatory myositis, it is less clear in muscles regenerating after acute injuries or in chronic diseases such as MDs. In Duchenne muscular dystrophy (DMD), dystrophin loss causes sarcolemma fragility and myofiber necrosis[1], leading to cellular debris that constitute a major source of damage-associated molecular patterns (DAMPs) attracting inflammatory cells, such as macrophages and lymphocytes. This condition is reinforced by the upregulation of specific factors as tumour necrosis factor alpha (TNF-α) and transforming growth factor beta (TGF-β), by the over-expression of interleukins (IL-1, IL-6) and oxidative stress[2]. In addition to inflammatory cell invasion determined by DAMPs, different works proposed that immune response in dystrophic muscles can be ascribable to a specific response to muscle antigens[3], resulting into the development of antigen-specific T cells with the ability of oligo-expansion[4]. In a study devising the intramuscularly injection of functional dystrophin transgene in six DMD patients, Mendell et al. demonstrated the unexpected presence of circulating CD4+ T lymphocytes against self-dystrophin epitopes in two patients before treatment. The number of these spontaneously primed T cells in blood was found inversely correlated with steroid treatment[5]. Similarly, Flanigan et al.[6] demonstrated that one-third of 70 DMD patients cohort developed spontaneously a T cell-mediated immune response against dystrophin, to a lesser degree in patients treated with deflazacort or prednisone. Accordingly, it is plausible that in DMD patients not all the dystrophin-reactive T cells are deleted into the thymus and they can be activated by dystrophin expressed from revertant myofibers in muscle tissues. In a dystrophic murine model (mdx mouse), we recently found the presence of anti-dystrophin T lymphocytes and the over-expression of immunoproteasome (IP), an enzymatic complex that cleaves peptides to produce epitopes for antigen presentation to T lymphocytes[7]. In line, we demonstrated that IP inhibition improved dystrophic muscle functions by reducing the number of both circulating and infiltrating activated T cells, confirming a pathogenic role of immune cells[7]. Dystrophic muscle features were also improved by depletion of B and T cells, which turned into reduced TGF-β activation and fibrosis deposition, as we observed in immunodeficient mdx mice (scid/mdx)[8].

The central role of immune mechanisms in tissue repair has been demonstrated by the occurrence of a specialized population of regulatory T cells (Treg) accumulating within injured muscles and promoting muscle regeneration[9,10]. This regulatory population holds a particular interest in mdx mice where it is able to mitigate muscle injury and inflammation or, conversely, to exacerbate the dystrophic pathological features by its expansion or depletion[9,10]. All these findings suggest that muscle regeneration is not only controlled by satellite cell activation, but rather by the existence of a more complex network of inflammatory and immune cells, cytokines and growth factors affecting the proliferation and differentiation of muscle stem cells.

The normal thymus generates mature T lymphocytes from bone marrow-derived progenitors, and screens them for autoreactivity[11]. Potentially auto-aggressive lymphocytes presenting either major histocompatibility complex (MHC) antigens (including cross-presented peptides of autoantigens) or tissue-restricted antigens (TRAgs) are deleted in the thymic medulla[12], respectively, by dendritic cells (DC) and medullary thymic epithelial cells (mTECs) under the control of autoimmune regulator (AIRE) or other analogous regulators. A minority of TRAg-expressing lymphocytes escapes deletion and matures into the natural Treg subset. Like other descendants, Tregs then migrate into the peripheral pool, where they maintain self-tolerance and immunity against infection.

The maintenance of thymic immune functionality is notoriously jeopardized by aging processes in a fashion faster than in other tissues and organs, and even in healthy subjects. The progressive aging leads to a continuous loss of thymic epithelial cells (TECs), which is particularly drastic in the case of mTECs of mdx mice[13,14]. Thymic involution increases the susceptibility to infection and infective diseases and impairs response to vaccines and cancer immunosurveillance[15,16]. While cortical TECs (cTECs) are essential for the early events of T cell differentiation, the primary role of mTECs is establishing the self-tolerance through negative selection and generation of regulatory T cells[17]. As thymic involution is accompanied by the replacement of T cell compartment with adipocytes and increase of pro-inflammatory cytokines[18] and thymic T cells express hormones[19], it was suggested a role for ghrelin (GHR) and its receptor (GHS-R) in thymus functions and, in particular, in generation of naive T cells and in the regulation of inflammatory cytokines.

In this work, we aimed at studying the relationship between dystrophic thymus and impaired muscle function in mdx mice. We found chaotic architecture of mdx thymus and modification of epithelial thymic niche, together with significant downregulation of autophagic machinery and AIRE expression. More importantly, we speculated that the lack of GHR in mdx thymus had a deleterious effect on organelle functions, limiting T cell lymphopoiesis, energy balance and inflammation. In a second extent, we established a model of thymus transplantation from mdx to nude mice—that lack thymus and are unable to produce T cells—to better investigate our hypothesis. Interestingly, mdx thymus transplantation in nude mice allowed the formation of mature T cells that caused the upregulation of pro-inflammatory cytokines and genes. This condition led to the reduction of myofiber area and the rise of fibrosis, turning definitively into loss of skeletal muscle mass and autophagic/metabolic dysfunctions.

All these data recapitulate the role of T lymphocytes in determining the pathogenesis of DMD and, more in general, we highlighted the involvement of thymus-derived immune cells in dysfunctions affecting dystrophic muscles.

## Results

**Altered thymic architecture in mdx mice is accompanied by ghrelin receptor expression loss.** In healthy conditions represented by C57Bl mice, the thymic development of T cells is a multi-steps process relying on the entrance of lymphoid progenitor cells into the thymus and the subsequent formation of the double positive (DP) CD4+CD8+ thymocytes. These DP thymocytes are then transformed into mature thymocytes and pushed out the thymus. The correct T cell maturation, the negative selection and the self-tolerance development are tightly controlled by the thymic stromal architecture, implying that a disorganization of medulla and cortex areas may result in the autoreactive T cell occurrence. Mdx thymus revealed alterations of tissue morphology, as already described[14]. Haematoxylin and eosin (H&E) analysis of C57Bl thymus showed a distinct cortico-medullary junction (CMJ), that is a key site for positive/negative selection of T cell progenitors (dashed white line in Fig. 1a), while the mdx thymus revealed a scattered and faintly defined junction area (dashed white line in Fig. 1a).

The expression of cytokeratins (CK) 14/16, which identify the mTECs with a fundamental role in regulating structural integrity of the cells[20], was also significantly downregulated in the

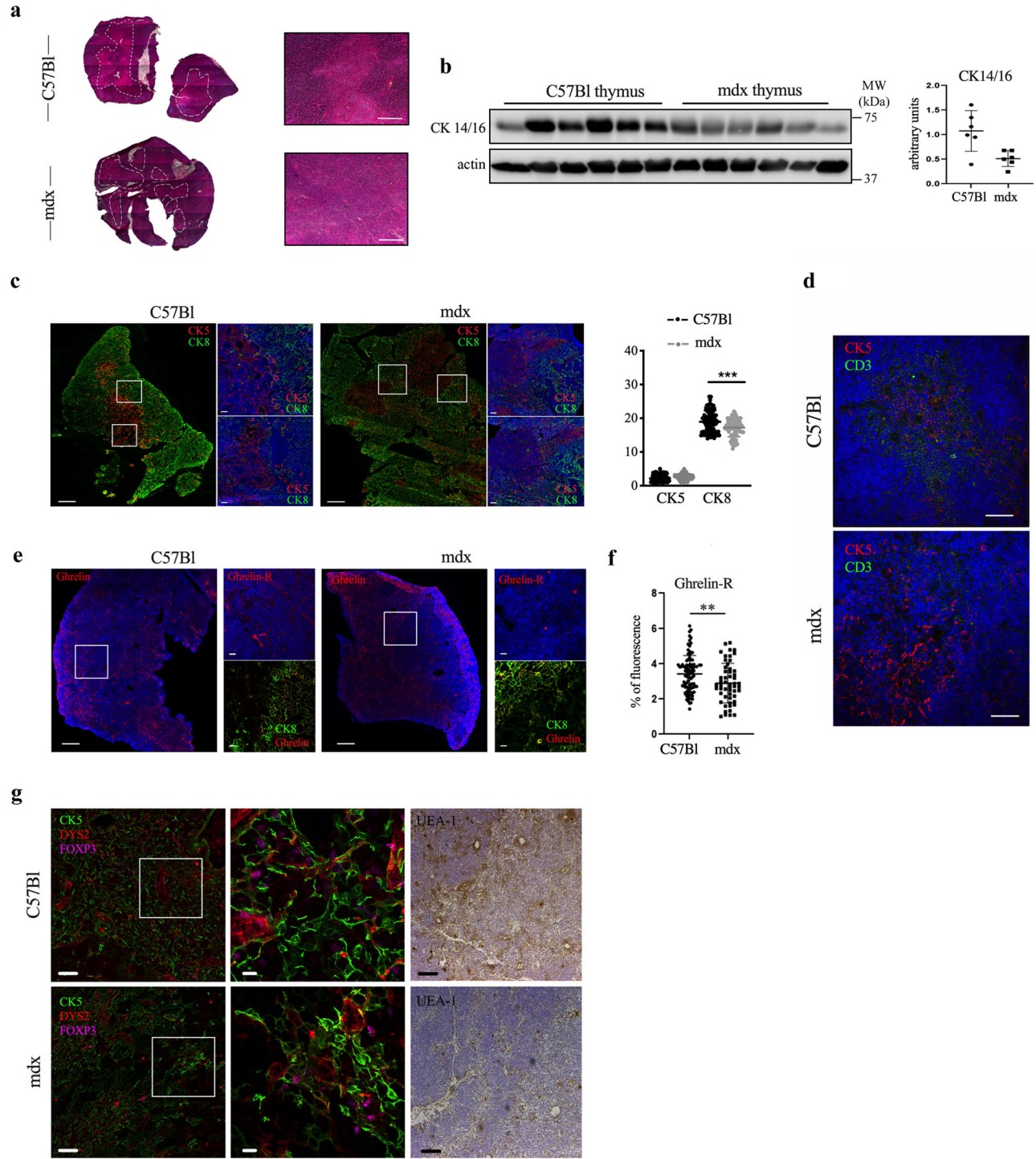

thymus of mdx mice compared to the C57Bl thymic expression (Fig. 1b).

Immunofluorescence staining of thymic medullary (CK5) and cortical (CK8) cytokeratin showed a detectable compartmentalization of cortex and medulla in C57Bl mice, while in mdx mice medullary area branched to the cortex, without clear boundaries associated with a significant reduction of CK8+ cTEC cells (Fig. 1c). No differences in medullary area size neither in the number of CK5+ mTEC were detected by the means of image quantification, although its architecture changes were noticeable, and they may jeopardize an efficient CD3+ cells/mTECs interaction (Fig. 1d). CD3+ lymphocytes were found distantly spread out within dystrophic thymic medulla, suggesting a loosen

contact with TRAg expressing cells (Fig. 1d). On the other hand, we did not appreciate any difference in B-lymphocyte expression by the means of immunofluorescence staining with B-220-specific marker (unpublished observations). According to the role of GHR in mediating T cell compartment and inflammation in thymus, we investigated the presence of this hormone in C57Bl and mdx mice. Fluorescent staining for GHR showed a similar pattern distribution in both mice, and a preferential localization in close proximity of the cortex network at the CMJ (Fig. 1e). GHS-R appeared instead to be hardly detectable in dystrophic thymus compared to C57Bl (Fig. 1f). Since the transcription factor FoxP3 is required for the generation, identity, and suppressive function of CD4+ Treg cells, we analysed the interaction between FoxP3+

**Fig. 1 Altered thymic architecture in mdx mice.** Representative images of H&E staining of thymus of 3-month-old C57Bl and mdx mice revealed differences in medullary/cortex boundaries between animals (dashed white line) (**a**). WB analysis showed the downregulation of cytokeratin 14/16 in mdx thymus related to C57Bl (**b**). Thymic architecture of C57Bl and mdx mice characterized by immunofluorescence staining for cortical cytokeratin CK8 (green) and medullary cytokeratin CK5 (red) confirmed changes in dystrophic thymic environment. Graph displays fluorescence area % occupied by CK5 and CK8, as calculated by ImageJ software (**c**). Double immunofluorescence staining for CK5 (red) and CD3 (green) of C57Bl and mdx thymi portrayed a loosen embedding of CD3+ cells within dystrophic medulla. **d** Staining of ghrelin (GHR) and ghrelin receptor (GHS-R) (red) showed a comparable distribution of GHR between animals, but a prevalent expression of GHS-R in thymus of C57Bl mice. Of note, GHR was preferentially found in proximity of cortical CK8 (green). **e** Graph displays fluorescence area % occupied by GHS-R, as calculated by ImageJ software, in C57Bl and mdx mice (**f**). Expression of FoxP3+ cells (magenta) was evaluated by immunofluorescence staining within CK5+ thymic medulla (green). C-terminal containing dystrophin isoforms were detected by DYS-2 antibody (red) to identify a specific protein distribution within thymus. mTEC maturation level was evaluated by immunohistochemistry staining of C57Bl and mdx thyme with UEA-1. For fluorescence microscopy, nuclei were counterstained with DAPI (**g**). Scale bars: 200 μm (**a**); 100 μm for confocal tile scan reconstruction (left) and 20 μm for higher magnification confocal microscope images (right) (**c, e**); 50 μm (**d**); 50 μm (left) and 10 μm for higher magnification confocal microscope images (right) (**g**). The comparisons between the averages of the two groups were evaluated using two-sided Student's t-test. **b** *$p = 0.0495$. **c** ***$p = 0.0007$. **f** **$p = 0.0046$. Data are presented as mean ± SD of three independent experiments with $n = 6$ mice/group. For Ck5–Ck8 immunofluorescence staining $n = 12$ images/mice have been quantified, GHS-R staining was quantified in $n = 12$ and $n = 8$ images of C57Bl and mdx mice, respectively. Source data are provided as a Source Data file.

cells and CK5+ mTECs (Fig. 1f). Interestingly, FoxP3+ cells were intimate connected to CK5+ mTEC in C57Bl and diffusely distributed around DYS2+ sheath-like structures reminiscent of the fibroblastic reticular cells observed in secondary lymphoid organs structures[21]. However, FoxP3+ cells were sparse and separated from mTECs and DYS2+ structures in mdx and sharply concentrated at CMJ (Fig. 1f). Maturation of mTECs is progressively marked by the expression of UEA-1. Detectable mature UEA1+ mTECs were found at CMJ but reduced in the medullary regions of mdx (Fig. 1f). Collectively, the above data suggest that dystrophic thymus is impaired in its architecture.

**Altered lipidic expression in dystrophic thymus.** Dysfunctions of GHR and GHS-R axis have been demonstrated to cause a reduction in the amount of naive T cells and consequent defects in thymic output[22]. In particular, high expression levels of GHS-R may influence adipogenic lipid-expressing cells, while its absence conversely allows the upregulation of thymic adipocytes[18]. Considering the low level of GHS-R observed in dystrophic thymus, we moved to investigate whether this decrease could be correlated to changes in lipid molecular pattern. Quantitative real-time reverse transcriptase PCR (RT-qPCR) analysis of lipids in C57Bl and mdx thymus showed a downregulation of Phosphoenolpyruvate Carboxykinase (*PECPK*) expression and Angiopoietin-Like Protein 4 (*pgar*) in the cortex of dystrophic thymus compared to C57Bl (Supplementary Fig. 1a). Accordingly, imaging mass spectrometry of lipids in thymic tissues revealed modulation of the amount of several lipids, as demonstrated for the glycerophospholipids indicated by *m/z* values (Supplementary Fig. 1b). These results hint at a potential ghrelin receptor involvement in the modulation of genes associated with dystrophic thymic stromal microenvironment changes and adipogenesis.

**Abnormal T cell development and autophagy impairment of dystrophic thymus.** Based on the above data, we sought to further elucidate the thymocyte commitment, development and/or function in mdx mice. Both C57Bl and mdx mice showed similar absolute numbers of thymic CD4−CD8− double-negative (DN) cells as well as of CD4+CD8+ DP cells and CD4+CD8− and CD4−CD8+ single positive (SP) thymocytes (Fig. 2a, b). Development progression of DN thymocytes is characterized by an ordered sequence of expression of CD44 and CD25 markers: CD44+CD25− (DN1), CD44+CD25+ (DN2), CD44−CD25+ (DN3), and CD44−CD25− (DN4). Analysis of the distribution of DN thymocytes in mdx mice revealed a significant decrease in DN3 cells and significant increase in DN4 cells, suggesting an

accelerated transition through the DN3 and DN4 stages (Fig. 2c). As DN4 are DP precursors, we analysed the DP stage in more detail using TCR-β and CD69 and found a significant increase in the percentage of TCR-β+CD69+ T cells in dystrophic thymus (Fig. 2d). Subsequent stage of development was characterized by the increased percentage of T-regs in dystrophic CD4+ SP cells (Fig. 2e). These results indicate an early activation of central tolerance in the presence of disorganized thymic architecture of mdx mice.

Impairment of NF-κB and STAT3 associated signalling was described in impaired thymus architecture and development of thymocytes[23,24]. Interestingly, we found a significant downregulation of NF-κB in thymus of mdx related to C57Bl mice without differences of STAT1 and STAT3 in their total and phosphorylated isoforms (Fig. 2f). Among the IKK-related kinases that regulate NF-κB activity, the IKKi is predominantly expressed in specific tissues such as the pancreas, thymus, spleen, and peripheral blood leucocytes[25] and its over-expression is known to effectively drive IκBα degradation responsible for NF-κB activation[11,26]. Notably, IKKi was significantly downregulated in thymus of mdx compared with age-matched C57Bl mice and the reduction of IKKi protein level was well correlated with NF-κB reduction (Fig. 2f). To investigate whether autophagy could affect dystrophic thymocytes development as previously reported in mdx muscle tissues[27,28] and in T cell tolerance to self-antigens[29,30], we analysed the mechanistic activation of Atg7-mediated conjugation of microtubule-associated protein 1 light chain 3 (LC3-I) to the membrane lipid phosphatidylethanolamine to form LC3-II and the expression of LC3-binding chaperone *p62* (ref. [31]). Similar levels of p62 and Atg7 were found between dystrophic and healthy thymus both in RT-qPCR (Fig. 2g) and western blot (WB) analysis (Fig. 2h), whereas the LC3-II/LC3-I ratio displayed a significant decrease in mdx compared to C57Bl (Fig. 2h), suggesting altered autophagic flux in dystrophic thymus.

**AIRE signalling pathway dysregulation in mTEC of mdx thymus.** As mentioned above, the TEC architecture disruption in thymus of mdx mice is associated to the dramatic loss of GHS-R, and defects in NF-κB signalling pathways and autophagy machinery which are important regulators of thymocyte selection and T-lymphocyte development[32,33]. This condition likely recalls the pathological phenotype caused by defects in AIRE signalling pathway.

Staining with anti-AIRE antibody revealed a relative abundance of AIRE+ cells in the thymic medulla of C57Bl mice (Fig. 3a). Interestingly, AIRE protein expression was significantly

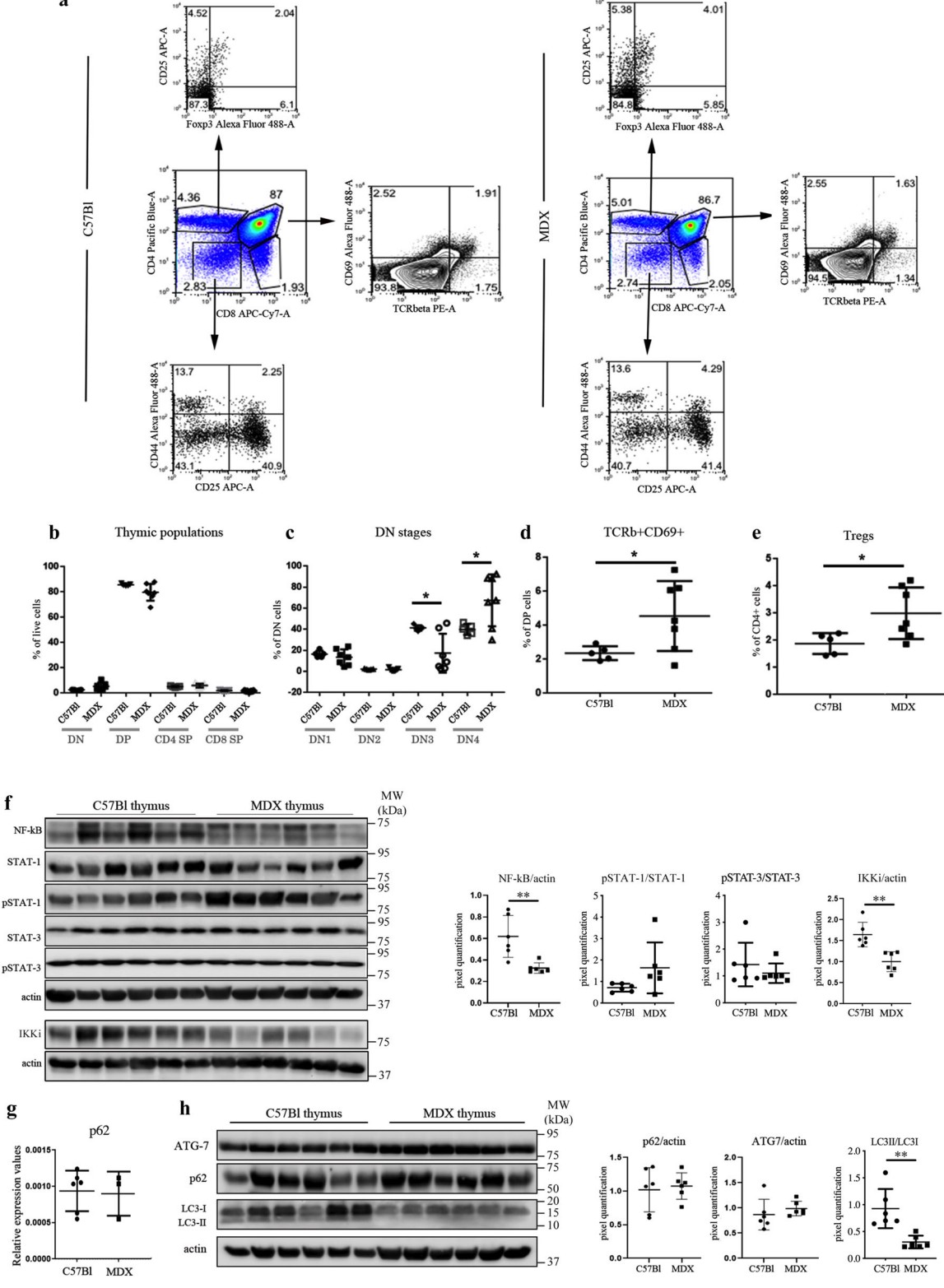

downregulated in mdx thymus such as the protein deacetylase Sirtuin 1 (SIRT-1) (Fig. 3b).

Next, we used fluorescence-activated cell sorting (FACS) to analyse *AIRE* and *AIRE*-dependent genes expression in sorted cTEC and mTEC cells. Sorting analysis using the canonical cortical (Ly5.1) and medullary (UEA1) markers revealed a decreased percentage of mTEC and cTEC in mdx thymus with a similar amount of MHC II expression compared to C57Bl

(Fig. 3c). Analyses of the expression of *AIRE* and *fezf2* and their regulators/regulated genes by real-time qPCR documented a lower expression of both *AIRE* and *fezf2* in dystrophic mTECs related to wild-type mTECs and an upregulation of these genes in C57Bl mTECs compared to C57Bl cTECs (Fig. 3d). Decreased expression of AIRE in mdx mTECs was associated to a significant downregulation of the expression of *Ins-2* related to C57Bl mTEC (Fig. 3f). The expression of *AIRE* up-stream regulators such as the

**Fig. 2 Cellularity, NF-kB/STATs expression, and autophagy in thymus of C57Bl and mdx mice.** FACS analysis of thymus homogenate from mdx and C57Bl mice at 8 weeks of age demonstrates no significant alteration of T cells (**a**, **b**), and few differences in CD4−CD8−DN stages, in particular DN3 (CD44−CD25+) and DN4 (CD44+CD25+) (**c**). The number of TCRβ+CD69+ cells (**d**) and of Foxp3+CD25+ cells (**e**) was significantly increased in thymus of mdx mice. Cropped image of a representative WB and densitometric analysis revealed a downregulation of NF-kB, IKKi, and STAT3 in mdx thymus (**f**). RT-qPCR of *p62* expression is shown in **g**. Autophagy markers such as Atg7, p62 and LC3 were also assessed by WB analysis. Representative WB image and quantification of LC3-II/LC3-I showed the impairment of the autophagic flux (**h**). All protein expression was normalized on actin, as a loading control. The comparisons between the averages of the groups were evaluated using two-sided Student's *t*-test. **c** *$p = 0.0177$ (DN3), *$p = 0.0351$ (DN4). **d** *$p = 0.043$. **e** *$p = 0.0332$. **f** **$p = 0.0048$ (NF-κB), **$p = 0.0018$ (IKKi). **h** **$p = 0.0026$. Data are presented as mean ± SD of three independent experiments with $n = 7$ (mdx) and $n = 5$ (C57Bl) mice (**a–e**); $n = 6$ mice/group (**f**, **h**); $n = 3$ mice (mdx) and $n = 6$ mice (C57Bl) (**g**). Source data are provided as a Source Data file.

*Tnfrsf11a* (or receptor activator of NF-*kB*, RANK) and the *CD40* was similar between the mTEC subpopulations from mdx and C57Bl thymi, suggesting no dysfunctions in cross-talk among the resident epithelial cells and lymphocytes in dystrophic thymus (Fig. 3e). However, mTEC cells showed a significant increase in the level of the *AIRE*, *fezf-2*, *ins-2*, *spt-1* and *mup4* transcripts in C57Bl mice. Mdx mTEC showed only *mup4* upregulation compared to mdx cTEC (Fig. 3d–f).

**Dystrophin isoform expression in C57Bl and mdx thymus.** Thymic stromal cell populations are able to express tissue-specific antigens, either synthetized in peripheral tissues and circulating or isolated self-antigens. This expression depends on a mechanism called promiscuous gene expression that is correlated to T cell negative selection. Since dystrophic skeletal muscles of mdx do not express the full-length dystrophin protein (427 kDa), we verified whether immune tolerance for dystrophin could be achieved by self-dystrophin expression in the thymus. We performed immunofluorescence staining of C57Bl and mdx thymi using two different antibodies: DYS-1, recognizing the mid-rod domain and therefore the full-length 427 kDa dystrophin, and DYS-2, recognizing the C-terminal domains of 71 and 427 kDa dystrophin isoforms. Positive staining for DYS-1 and DYS-2 was clearly detectable in C57Bl thymus lobe (Fig. 4a). Conversely, DYS-1 staining was very weak in mdx mice, and probably attributable to utrophin cross reactivity, while DYS-2 staining was similar to the one depicted in C57Bl (Fig. 4a). Dystrophin isoform detection by RT-PCR analysis in C57Bl thymus confirmed the expression of the ubiquitous dystrophin isoform Dp71 (71 kDa), the full-length Dp427 (427 kDa) (Fig. 4b), and of the Dp140 (140 kDa), appreciable only following the purification of PCR products and to a lower extent (Fig. 4c). Thymus of mdx mice showed the expression of only dystrophin isoform Dp71 (Fig. 4d). WB analysis again proved the expression of the Dp71 isoform in the thymus of both mice, and the expression of Dp427 only in C57Bl thymus (Fig. 4e).

**Adult dystrophic thymus transplantation into nude mice altered T cell development.** T cell involvement in DMD pathogenesis was previously demonstrated[34]. To determine the possible role of the self-reactive T cells in DMD pathology, we subcutaneously transplanted adult thymic tissues of mdx and C57Bl into C57Bl/nude mice (referred to as Tnu$^{MDX}$ nude and Tnu$^{C57Bl}$ mice, respectively) (Fig. 5a). The aim was to provide thymocyte development with two different thymic epithelial niches that differ mainly for the absence of dystrophin. Nude mice treated with PBS (nu$^{PBS}$) were used as control. Since it is accepted that nude mice had no T cell functions[35,36], we monitored the amount of circulating CD3+ T cell subpopulations in the peripheral blood of nude mice over time after transplantation, to determine whether the injection of thymus partially rescued this condition (Fig. 5b). The populations of CD3+CD4+, CD3+CD8+ T cells can be clearly detected in circulation and gradually increased over

time after transplantation and much more than nu$^{PBS}$ mice (Fig. 5c). A significant increase of CD3+CD25+ T cells was observed in the Tnu$^{C57Bl}$ vs Tnu$^{MDX}$ mice at day 116 post-transplantation (pt). Not significant modifications were found in the number of CD3+CD4+ and CD3+CD8+ subpopulations at each time point between Tnu$^{C57Bl}$ vs Tnu$^{MDX}$ mice (Fig. 5c). Overall, the trends of Tnu$^{MDX}$ mice showed a reduction over time in the number of CD3+CD25+ T cells pt and in the number of CD3+CD4+ T cell subset. Conversely, the amount of Tnu$^{MDX}$ CD3+CD8+ T cells incremented from day 88 until day 116 pt. Linear regression of data for Tnu$^{MDX}$ mice also confirmed substantial changes over time in the number of CD3+CD25+ T cells and CD3+CD8+ T cells (Fig. 5c). Previous reports demonstrated that among the infiltrating lymphocytes, the largest subpopulation in the dystrophic muscles is represented by the pro-inflammatory T helper 17 lymphocytes (Th17)[6]. Moreover, the thymic isoform of the RAR-related orphan receptor gamma (RORγt) gene is fundamental to drive the differentiation of the T cells into Th17-positive cells[37]. We found significant upregulation of RORγt expression in Tnu$^{MDX}$ muscles compared to the Tnu$^{C57Bl}$ and nu$^{PBS}$ (Fig. 5d). These data correlated with the expression of Th1-specific gene *T-bet* that was clearly detectable only in the Tnu$^{MDX}$ mice (Fig. 5e). We have also characterized the *tibialis anterior* (TAs) of the injected animals to evaluate the amount of mature infiltrating T cells in skeletal muscles. Interestingly, we observed that the proportion of CD3+ T cells was increased in the muscle of Tnu$^{MDX}$ mice compared to the Tnu$^{C57Bl}$ and nu$^{PBS}$. Among these cells, we found a similar increase for both the CD4+ and the CD8+ lymphocyte compartments (Fig. 5f).

**Tnu$^{MDX}$ mice exhibit dystrophic muscle features and skeletal muscle regression.** We measured the amount of dystrophin protein in Tnu$^{MDX}$ compared to Tnu$^{C57Bl}$ and nu$^{PBS}$ mice. TAs immunostaining with C-terminal DYS-2 antibody depicted a weak intensity of the dystrophin fluorescence in the positive myofibers of the Tnu$^{MDX}$ mice (Fig. 6a). The WB analysis showed downregulation of the dystrophin Dp427 isoform in both the Tnu$^{MDX}$ and Tnu$^{C57Bl}$ compared to nu$^{PBS}$ mice (Fig. 6b), whereas no differences have been found in dystrophin Dp427 mRNA expression (Fig. 6c). Such similarity between Tnu$^{MDX}$ and Tnu$^{C57Bl}$ in dystrophin expression was unexpected as we prevalently found T-infiltrating muscle cells in mdx mice. Because the myosin motor protein is closely associated with dystrophin-based differences in muscle function[38], we verified the fibre compositions of Tnu$^{MDX}$, Tnu$^{C57Bl}$ and nu$^{PBS}$ mice. We observed a statistically significant decrease in the number of the oxidative/glycolytic MyHC type IIx and IIb myofibers in TAs of Tnu$^{MDX}$ mice (Fig. 6d). Similarly, we observed the over-expression of the slow-specific gene slow myosin heavy chain 2 (MyHC-SL2) mice compared to Tnu$^{C57Bl}$ and nu$^{PBS}$ mice which correlated with no modification of the levels of fast-fibre-specific ATPase sarcoplasmic/endoplasmic reticulum Ca$^{2+}$ Transporting (*atp2a*)-1

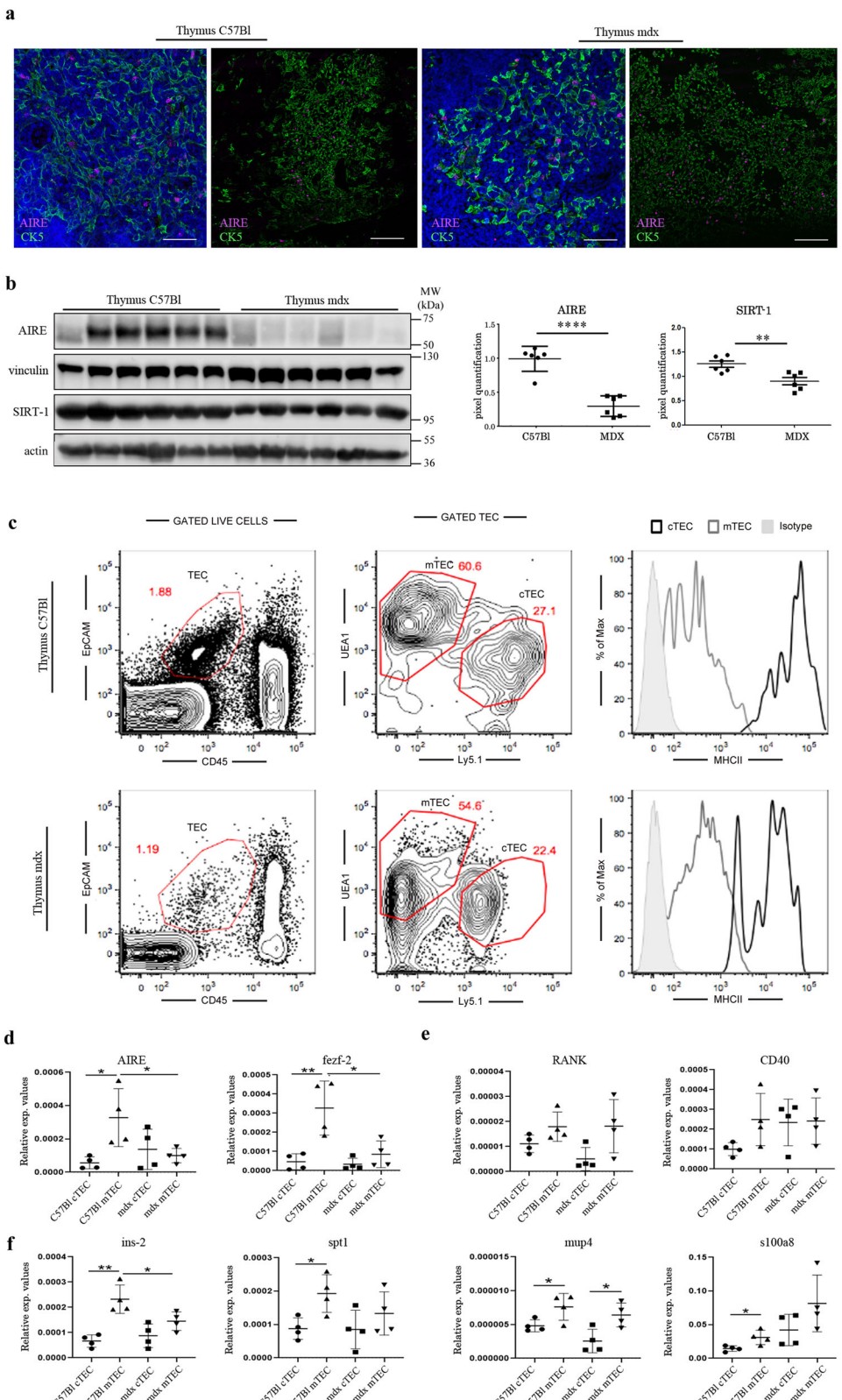

(Fig. 6e). Overall, the Tnu$^{MDX}$ mice developed signs of muscular atrophy as demonstrated by the marked weight loss compared to Tnu$^{C57Bl}$ and nu$^{PBS}$ mice (Fig. 6f). Thus, we measured the expression of muscle RING-finger protein (*MuRF*)-1 as a downstream signalling effect of muscle atrophy and controller of amino acids metabolism and ATP synthesis[39–41]. We found that Tnu$^{MDX}$ mice showed an up-regulated expression of *MuRF*-1

compared to Tnu$^{C57Bl}$ and nu$^{PBS}$ mice, as also demonstrated in TAs of untreated mdx vs C57BL mice used as control (Fig. 6g). Next we examined the expression of the pro-growth factor Akt[42,43] and found no changes in Akt expression between mice (Fig. 6h).

Morphometric analysis of quadriceps (QAs) and TAs of Tnu$^{MDX}$ mice showed centrally nucleated regenerating myofibers and

**Fig. 3 AIRE dysregulation in thymus of 3-month-old mdx mice.** Representative confocal microscope images (left) and tile scan reconstruction (right) of thymic lobes from 3-month-old C57Bl and mdx mice. Despite a comparable AIRE+ cell pattern distribution embedded within CK5+ thymic medulla of both mice, in mdx thymus immunofluorescence staining for AIRE appeared less evident. Nuclei were counterstained with DAPI (**a**). Representative images of western blot analysis showing the expression of the AIRE and SIRT-1 proteins in the thymus of C57Bl and mdx mice. Densitometric analyses are shown as the AIRE/vinculin ratio and SIRT-1/actin ratio (**b**). For the identification and sorting of cTEC and mTEC from C57bl and mdx mice, stained thymus cell suspension was analysed using flow cytometry. Representative FACS profile are shown. The numbers within the panels indicate the percentage of each population of live cells, a gate of CD45-negative and EpCAM-positive events represents TEC cells. Within the TEC gate, two population are separated by level of Ly5.1 and UEA-1. MHC II molecules were highly expressed on cTECs (**c**). Following FACS isolation of mTEC and cTEC, RT-qPCR experiments showed the downregulation of *AIRE* and *Fezf2* in isolated mdx mTEC related to mTEC from C57Bl thymus (**d**). The expression in thymus of *AIRE*-regulators *RANK* and *CD40* was similar between mTEC isolated from C57Bl and mdx mice (**e**). RT-qPCR revealed diminished expression of *AIRE*-dependent genes *Ins2* and *Spt1* in dystrophic mTEC and in control cTEC related to mTEC isolated from C57Bl thymus (**f**). Scale bar: 50 μm (**a**). The comparisons between the averages of the groups were evaluated using two-sided Student's *t*-test. **b** ****$p < 0.0001$ (AIRE), **$p = 0.0050$ (SIRT-1). **d** $p = 0.0436$ (*AIRE*) and $p = 0.0220$ (*fezf2*) in dystrophic mTECs vs C57Bl mTECs; $p = 0.0224$ (*AIRE*) and $p = 0.00089$ (*fezf2*) in C57Bl mTECs vs C57Bl cTECs. **f** $p = 0.0413$ (*Ins-2*) in mdx mTECs vs C57Bl mTECs; $p = 0.0017$ (*Ins-2*), $p = 0.0178$ (*spt1*), $p = 0.0413$ (*mup4*), $p = 0.0271$ (*s100a8*) in C57Bl mTECs vs C57Bl cTECs; $p = 0.0210$ (*mup4*) in C57Bl mTECs vs C57Bl cTECs. Data are presented as mean ± SD of three independent experiments with $n = 6$ mice/group (**a, b**); $n = 8$ (**c**); $n = 8$ mice divided into two independent groups examined in $n = 2$ independent experiments (**d–f**). Source data are provided as a Source Data file.

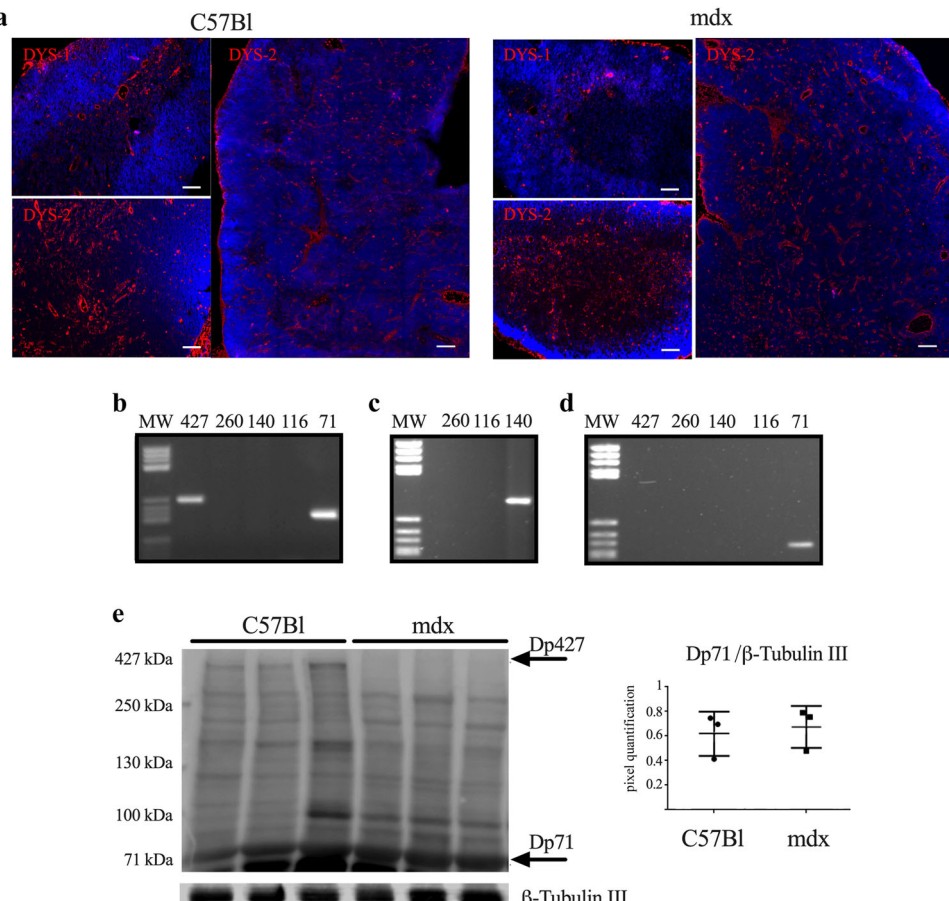

**Fig. 4 Dystrophin isoforms expression in thymus of 3-month-old C57Bl and mdx mice.** Confocal microscope fluorescence images (right) and tile scan reconstructions (left) of dystrophin isoforms in thymi of C57Bl and mdx mice. DYS-1 (mid-rod-domain) (red) and DYS-2 (C-terminal-domain) (red) antibodies were used (**a**). RT-PCR analysis of the thymus of C57Bl mice determined the expression of *Dp71, Dp427* dystrophin isoforms (**b**). Following the purification of the PCR products related to the isoforms *Dp260, Dp140,* and *Dp116,* we performed another PCR on these samples and we found one band corresponding to *Dp140* in C57Bl thymus (**c**). RT-PCR analysis of thymus of mdx mice determined the expression of the only *Dp71* dystrophin isoform (**d**). WB analysis confirmed the absence of Dp427 dystrophin isoform in thymus of mdx mice and the presence of DP71 dystrophin isoform in thymus of both C57Bl and mdx mice (**e**). Densitometric analyses are shown as DP71/β-Tubulin III (**e**). Scale bar: 100 μm for tile scan reconstructions and 50 μm for fluorescence images (**a**). MW stands for molecular weight marker (**b–d**). Data are presented as mean ± SD of three independent experiments with $n = 3$. Source data are provided as a Source Data file.

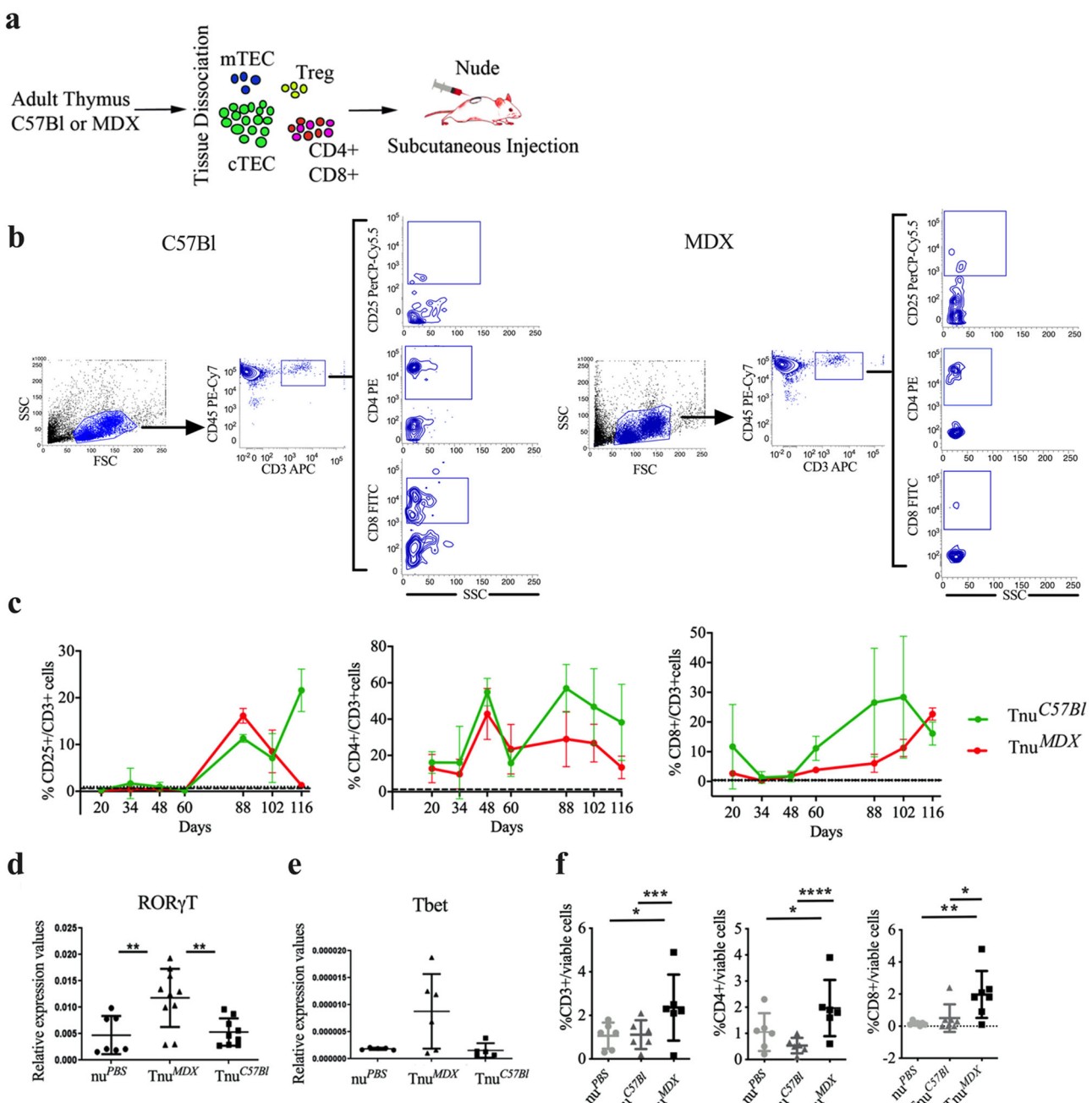

**Fig. 5 Characterization nude mice following adult thymus transplantation.** Schematic overview of the experimental procedure (**a**). Gating strategy to identify CD25+/CD3+, CD4+/CD3+ and CD8+/CD3+ cell subpopulations in Tnu$^{C57Bl}$ and Tnu$^{MDX}$ mice. All subpopulations are detected within CD45+ CD3+ gate (**b**). FACS analysis of CD3-expressing blood-derived subpopulation isolated from Tnu$^{C57Bl}$ and Tnu$^{MDX}$ mice at different days following the thymus transplantation. Dashed lines referred to averaged values of nu$^{PBS}$ mice used as controls. The number of the CD3+, CD4+ and CD8+ T cells were significantly higher in Tnu$^{MDX}$ muscles related to Tnu$^{C57Bl}$ and nu$^{PBS}$ mice (**c**). RT-qPCR analysis revealed significant over-expression of Th17 key gene, *RORγt* (**d**) and the upregulation of Th1 key gene *T-bet* (**e**) in muscles of Tnu$^{MDX}$ mice. **f** FACS analysis of muscles isolated from Tnu$^{C57Bl}$, nu$^{PBS}$ mice and Tnu$^{MDX}$ mice for quantification of mature infiltrating T cells. The number of the CD3+, CD4+ and CD8+ infiltrating T cells were significantly higher in Tnu$^{MDX}$ muscles related to Tnu$^{C57Bl}$ and nu$^{PBS}$ mice. The comparisons among the averages of the groups were evaluated using Linear regression analysis (**b**) and one-way ANOVA (**d–f**). **c** *$p = 0.0085$ (CD3+CD25+ T cells: Tnu$^{C57Bl}$ vs Tnu$^{MDX}$ mice at day 116 pt); CD3+CD25+ T cells in Tnu$^{MDX}$ mice: ****$p < 0.0001$, day 88 vs day 116 pt; ***$p = 0.0077$ day 88 vs day 102 pt; *$p = 0.0383$ day 102 vs day 116 pt; CD3+CD4+ T cells in Tnu$^{MDX}$ mice: *$p = 0.0482$, day 88 vs day 116 pt; CD3+CD8+ T cells in Tnu$^{MDX}$: ****$p < 0.0001$, day 88 vs day 116 pt; *$p = 0.0392$, day 88 vs day 102 pt; ***$p = 0.0010$, day 102 vs day 116 pt; CD3+CD25+ T cells in Tnu$^{MDX}$ mice over time: *$p = 0.026$. **d** **$p = 0.0050$ (Tnu$^{MDX}$ vs Tnu$^{C57Bl}$); **$p = 0.0097$ (Tnu$^{MDX}$ vs nu$^{PBS}$). **f** CD3+ T cells: ***$p = 0.0011$ (Tnu$^{MDX}$ vs Tnu$^{C57Bl}$) and *$p = 0.0203$ (Tnu$^{MDX}$ vs nu$^{PBS}$); CD4+ T cells: ***$p < 0.0001$ (Tnu$^{MDX}$ vs Tnu$^{C57Bl}$) and *$p = 0.0105$ (Tnu$^{MDX}$ vs nu$^{PBS}$); CD8+ T cells: *$p = 0.013$ (Tnu$^{MDX}$ vs Tnu$^{C57Bl}$) and **$p = 0.0016$ (Tnu$^{MDX}$ vs nu$^{PBS}$). Data are presented as mean ± SD of three independent experiments with $n = 4$ mice Tnu$^{C57Bl}$ and $n = 5$ mice Tnu$^{MDX}$ (**c**); $n = 8$ mice Tnu$^{C57Bl}$, $n = 9$ mice Tnu$^{MDX}$ and $n = 7$ mice nu$^{PBS}$ (**d**); $n = 5$ mice Tnu$^{C57Bl}$, $n = 6$ mice Tnu$^{MDX}$ and $n = 5$ mice nu$^{PBS}$ (**e**); $n = 6$ mice/group for % of CD3+ and CD4+, $n = 7$ mice /group for % of CD8+ (**f**). Source data are provided as a Source Data file.

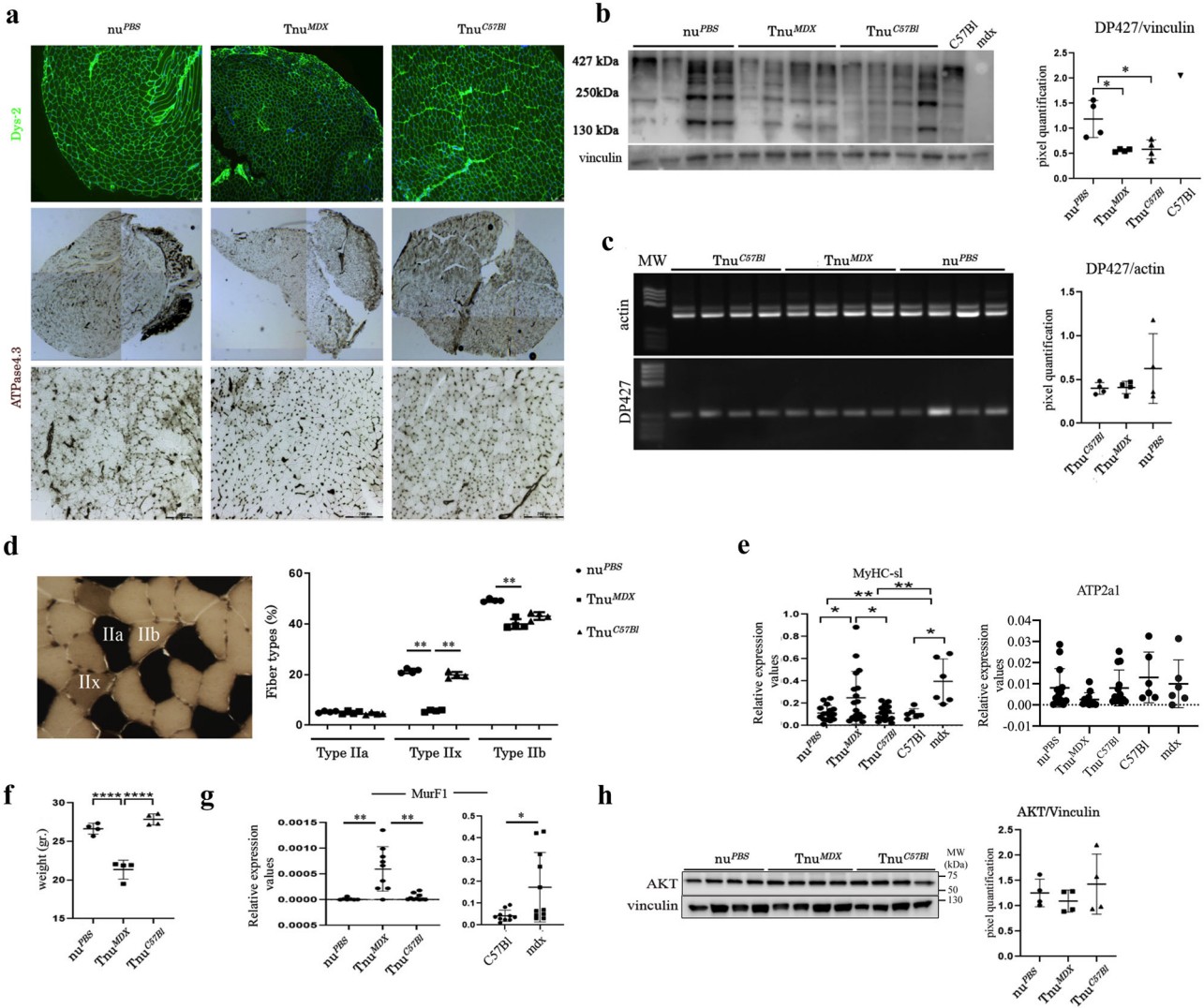

**Fig. 6 Adult dystrophic thymus transplantation determines dystrophic muscle features and skeletal muscle regression.** Representative immune fluorescence staining with DYS-2 (C-terminal-domain) antibody. Overall, confocal microscope images of TAs from nu$^{PBS}$, Tnu$^{MDX}$ and Tnu$^{C57Bl}$ showed weak dystrophin intensity around the myofibers in TA of Tnu$^{MDX}$ mice (**a**). Overview and higher magnification of ATPase (pH 4.3) muscle sections of TAs from nu$^{PBS}$, Tnu$^{MDX}$ and Tnu$^{C57Bl}$ mice (**a**). Densitometric analysis of WB images of dystrophin protein expression showed downregulation of Dp427 dystrophin isoform in TA muscles of Tnu$^{MDX}$ and Tnu$^{C57Bl}$ mice (**b**). RT-PCR analysis of TA of nu$^{PBS}$, Tnu$^{MDX}$ and Tnu$^{C57Bl}$ mice determined the expression of *Dp427* dystrophin isoform (**c**). Representative images of skeletal muscle showed the distribution and composition of the myosin heavy chain (MyHC) isoforms (Type IIa, Type IIx and Type IIb). Graph portrays the percentage of myofibers expressing different MyHC isoforms in TAs of nu$^{PBS}$, Tnu$^{MDX}$ and Tnu$^{C57Bl}$ mice. $n = 10$ images were analysed for each mouse (**d**). RT-qPCR experiments on TA muscles demonstrated the over-expression of *MyHC-sl2* gene together with the downregulation of fast *atp2a1* in TA of Tnu$^{MDX}$ mice related to TAs of other mice (**e**). Tnu$^{MDX}$ mice showed a dramatic weight loss (**f**), which correlated with the over-expression of the atrophy-related *MuRF-1* gene (**g**). Cropped image of a representative WB showing the expression of the Akt and vinculin proteins in TA muscles of nu$^{PBS}$, Tnu$^{MDX}$ and Tnu$^{C57Bl}$ mice. Densitometric analyses are shown as Akt/vinculin ratio (**h**). Scale bar: 50 μm for Dys-2 (**a**); upper images: 500 μm and higher magnification in bottom images: 200 μm for ATPase (**a**). The comparisons among the averages of the groups were evaluated using one-way ANOVA (**b**, **d–g**) and two-sided Student's *t*-test (**e**, **g**). **b** *$p = 0.0218$ (nu$^{PBS}$ vs Tnu$^{MDX}$), *$p = 0.0260$ (nu$^{PBS}$ vs Tnu$^{C57Bl}$). **d** **$p = 0.0098$ Tnu$^{MDX}$ vs Tnu$^{C57Bl}$ and **$p = 0.0024$ Tnu$^{MDX}$ vs nu$^{PBS}$ mice for MyHC type IIx; **$p = 0.0071$ Tnu$^{MDX}$ vs nu$^{PBS}$ mice for MyHC type IIb. **e** *$p = 0.0299$ Tnu$^{MDX}$ vs Tnu$^{C57Bl}$ and *$p = 0.0324$ Tnu$^{MDX}$ vs nu$^{PBS}$ mice for *MyHC-SL2*. **f** ****$p < 0.0001$ Tnu$^{C57Bl}$ and nu$^{PBS}$ vs Tnu$^{MDX}$. **g** ***$p = 0.0005$ Tnu$^{MDX}$ vs Tnu$^{C57Bl}$ and **$p = 0.003$ Tnu$^{MDX}$ vs nu$^{PBS}$ mice; *$p = 0.0189$ (mdx vs C57Bl). Data are presented as mean ± SD of three independent experiments with $n = 4$ mice Tnu$^{MDX}$, Tnu$^{C57Bl}$ and nu$^{PBS}$ (**a–d**); $n = 8$ mice Tnu$^{C57Bl}$, $n = 9$ mice Tnu$^{MDX}$ and $n = 7$ mice nu$^{PBS}$, $n = 6$ C57Bl and $n = 6$ mdx mice (**e**); $n = 4$ mice/group (**f**); $n = 8$ mice Tnu$^{C57Bl}$, $n = 9$ mice Tnu$^{MDX}$ and $n = 7$ mice nu$^{PBS}$, $n = 6$ C57Bl and $n = 6$ mdx mice (**g**); $n = 4$ mice/group (**h**). Source data are provided as a Source Data file.

fibrotic infiltrate suggesting a dystrophic phenotype (Fig. 7a, b). In particular, the fibrosis of QA muscles of Tnu$^{MDX}$ mice was significantly higher than Tnu$^{C57Bl}$ and nu$^{PBS}$ (Fig. 7a, b). According to this "dystrophic-like" phenotype, we found a higher number of necrotic myofibers in the QA of Tnu$^{MDX}$ mice (Fig. 7b) and a

significant increase of the number of centrally nucleated myofibers was also observed for TAs of the same animals (Fig. 7c).

The cross-sectional areas (CSAs) of the myofibers observed in the muscles of Tnu$^{MDX}$ mice were significantly lower than those observed in the muscles of Tnu$^{C57Bl}$ and nu$^{PBS}$ (mean area for

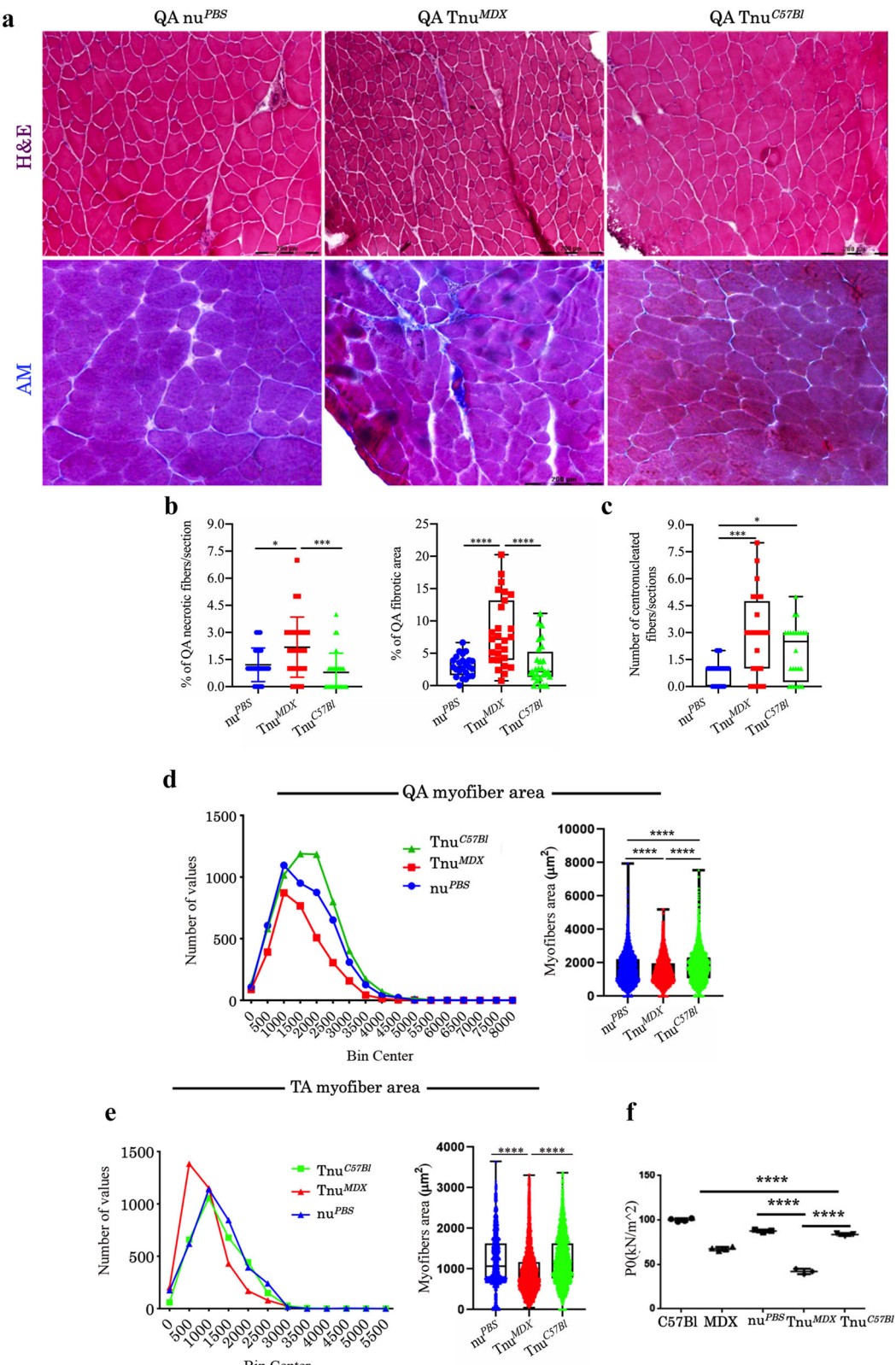

QA: Tnu$^{MDX}$: 1497 ± 13.70 µm$^2$ N = 3155; Tnu$^{C57Bl}$ 1751 ± 11.85 µm$^2$ N = 5597; nu$^{PBS}$: 1645 ± 12.59 µm$^2$ N = 4799)(mean area for TA: Tnu$^{MDX}$: 925.2 ± 10.26 µm$^2$ N = 3458; Tnu$^{C57Bl}$ 1222 ± 10.85 µm$^2$ N = 3078; nu$^{PBS}$: 1204 ± 10.78 µm$^2$ N = 3703) (Fig. 7d,e). Interestingly, the atrophic-like phenotype of QA muscles of Tnu$^{MDX}$ mice was confirmed by the coefficient of variation, as the lower variance for Tnu$^{MDX}$ mice was due to

myofiber area reduction consequent to loss of sarcomeric protein (nu$^{PBS}$: 760340.6; Tnu$^{MDX}$: 621440.8; Tnu$^{C57Bl}$ 785544.3). Furthermore, the values of frequency distribution confirmed the smaller area of myofibers in Tnu$^{MDX}$ mice (25% Percentile: Tnu$^{MDX}$: 936.36; Tnu$^{C57Bl}$: 1086.99; nu$^{PBS}$: 974.73. 75% Percentile: Tnu$^{MDX}$: 1971.2; Tnu$^{C57Bl}$: 2308.25; nu$^{PBS}$: 2230.67). Notably, strength evaluation evidenced a dramatic

**Fig. 7 Morphometric and functional analysis of skeletal muscles of nude mice following adult thymus transplantation.** Representative H&E and AM staining of QA muscles of $nu^{PBS}$, $Tnu^{MDX}$ and $Tnu^{C57Bl}$ mice (**a**). Quantification of the necrotic myofibers, fibrotic areas (**b**) and centrally nucleated myofibers (**c**) of the QA muscles of $nu^{PBS}$, $Tnu^{MDX}$ and $Tnu^{C57Bl}$ mice. Boxes indicate 25th to 75th percentiles; whiskers indicate 5th to 95th percentiles; and the line indicates the median. Quantification of the relative frequency of the myofiber cross-sectional area (CSA) expressed as the frequency distribution of the QA (**d**) and TA (**e**) muscles of the $nu^{PBS}$, $Tnu^{MDX}$ and $Tnu^{C57Bl}$ mice. Boxes indicate 25th to 75th percentiles; whiskers indicate 5th to 95th percentiles; and the line indicates the median. For morphometric analysis, images were quantified with ImageJ software for each mouse (**a–e**). Tetanic force of TA of C57Bl, mdx, $nu^{PBS}$, $Tnu^{MDX}$ and $Tnu^{C57Bl}$ mice is shown in **f**. Scale bar: 200 μm (**a**). The comparisons among the averages of the groups were evaluated using one-way ANOVA (**b–f**) and $F$-test to compare variance (**d, e**). **b** ***$p = 0.0003$ $Tnu^{MDX}$ vs $Tnu^{C57Bl}$ and *$p = 0.0207$ $Tnu^{MDX}$ vs $nu^{PBS}$ mice for necrotic fibres/section; ****$p < 0.0001$ for fibrotic area. **c** *$p = 0.0455$ $Tnu^{MDX}$ vs $Tnu^{C57Bl}$ and ***$p = 0.0001$ $Tnu^{MDX}$ vs $nu^{PBS}$ mice. **d–f** ****$p < 0.0001$. Data are presented as mean ± SD of three independent experiments with $n = 8$ mice/group (**a**); $n = 8$ mice $Tnu^{C57Bl}$, $n = 9$ mice $Tnu^{MDX}$ and $n = 8$ mice $nu^{PBS}$ (**b, c**); $n = 4$ (**d, e**); $n = 3$ mice $Tnu^{C57Bl}$, $n = 3$ mice $Tnu^{MDX}$ and $n = 3$ mice $nu^{PBS}$, $n = 4$ C57Bl and $n = 4$ mdx mice (**f**) mice/group. Source data are provided as a Source Data file.

reduction in the tetanic force of TA muscles of $Tnu^{MDX}$ mice compared to $Tnu^{C57Bl}$, $nu^{PBS}$, C57Bl and untreated mdx mice (Fig. 7f).

**Fibrotic and inflammatory markers are up-regulated in muscles of $Tnu^{MDX}$ mice.** The role of T-lymphocytes in controlling the expression of collagen in dystrophic muscles has been well described[34]. Thus, we sought to investigate the effect of transplantation of dystrophic thymus on muscle inflammation and downstream signalling markers of fibrosis. Immunofluorescence staining of TAs showed a marked collagen X deposition in muscles of $Tnu^{MDX}$ related to the $nude^{PBS}$ and $Tnu^{C57Bl}$ mice (Supplementary Fig. 2a). Collagen I and the idiopathic marker of fibrosis osteopontin (OPN) protein levels of TAs of $Tnu^{MDX}$ mice were not statistically different when compared to TAs of $Tnu^{C57Bl}$ and $nude^{PBS}$ mice (Supplementary Fig. 2b). However, RT-qPCR analysis of TAs revealed an over-expression of *collagen 3a*[44,45] in $Tnu^{MDX}$ related to $Tnu^{C57Bl}$ and $nude^{PBS}$ mice (Supplementary Fig. 2c).

Consistent with the increase of these fibrogenic markers, we found increased expression of IP subunits PSMB8 and PSMB9 in TAs of $Tnu^{MDX}$ related to $nude^{PBS}$ mice (Supplementary Fig. 3). We previously demonstrated the upregulation of PSMB8 and PSMB9 in DMD muscles as key molecules for regulating inflammation and T-lymphocytes infiltration[7]. We thus investigated the expression of the lymphocytes associated cytokines, such as TGF-β and TNF-α, and other engaged to their activity, as NF-kB. TGF-β expression was higher in muscles of $Tnu^{MDX}$ mice related to $nude^{PBS}$ and in $nude^{PBS}$ related to $Tnu^{C57Bl}$. Likewise, muscles of $Tnu^{MDX}$ displayed an upregulation of TNF-α and NF-kB that may turn into MuRF1 induction and contribute to muscle force reduction[39] (Supplementary Fig. 3).

**Autophagy is impaired in the skeletal muscles of $Tnu^{MDX}$ mice.** As lipidation of LC3 and its association with autophagosome membranes has been established as useful sign for autophagy in dystrophic muscle[46], we detected LC3 of treated skeletal muscle by immunoblotting. Importantly, a significant decrease of LC3-II/LC3-I ratio was observed in muscles of $Tnu^{MDX}$ mice compared to $nu^{PBS}$ and $Tnu^{C57Bl}$ mice. Additionally, we found increased expression of p62 and Atg7 in the muscles of $Tnu^{MDX}$ mice, suggesting that autophagy is downregulated in $Tnu^{MDX}$ mice (Supplementary Fig. 4a, b). Since p62 is strictly related to TRAF-6[47] and TRAF-6 is up-regulated in the skeletal muscles of mdx mice[48], we investigated its expression in treated mice. Interestingly, TRAF-6 was increased in $Tnu^{MDX}$ mice (Supplementary Fig. 4a, b). Increased levels of iNOS, characterizing DMD muscles and regulated by NF-kB activation, may contribute to the autophagy process inhibition[39,49,50]. Accordingly, *iNOS* expression was significantly increased in the muscle of $Tnu^{mdx}$ mice related to $nude^{PBS}$ and $Tnu^{C57Bl}$ mice (Supplementary Fig. 4c).

Since autophagy can indirectly impact on glucose metabolism and glycogenolysis, we verified whether impairment of autophagy affected the skeletal muscle metabolism of treated mice. Muscles of $Tnu^{MDX}$ showed an upregulation of pyruvate dehydrogenase kinase-4 (*pdk4*) (Supplementary Fig. 4d). In addition, a statistically significant increase of the antioxidant enzyme glutathione peroxidase 1 (*GP-x1*) was evident in $Tnu^{MDX}$ mice (Supplementary Fig. 4d). Then, we investigated whether the metabolic abnormalities could be associated to mitochondrial dysfunctions as described in mdx mice[51]. However, RT-qPCR analysis of the peroxisome proliferator-activated receptor-γ coactivator (*PGC*)-$1\alpha$, *PPARγ* and nuclear respiratory factor (*NRF*)-1, which coordinates the mitochondrial biogenesis[52], did not show any differences among mice (Supplementary Fig. 4e).

**$Tnu^{MDX}$ dystrophic like phenotype is not induced by systemic transplantation of mdx-derived T lymphocytes.** Since adult transplanted thymus may include donor-derived autoreactive T cells that could reach the host muscles through the circulation, nude mice were treated by intravenous injection of blood-derived CD3+CD4+ (CD4+$^{mdx}$) or CD3+CD8+ (CD8+$^{mdx}$) T lymphocytes isolated from the peripheral blood of 12 weeks-old mdx mice (Fig. 8a) and followed for 120 days. Therefore, in order to rule out the already reported muscle-dependent tissue activation of CD4+$^{mdx}$ and CD8+$^{mdx}$[3,7], we sought to further treat C57Bl mice by intra-arterial transplantation of CD4+$^{mdx}$ and CD8+$^{mdx}$. Morphometric analysis of the TAs of nude injected with CD4+$^{mdx}$ and CD8+$^{mdx}$ displayed significant differences compared to untreated nude mice in myofiber size variability, cross-sectional area (CSAs), with no differences in fibrosis and muscle force (Fig. 8a–e). Additionally, no differences were seen in the expression of dystrophin and various inflammatory molecules (*PDK4, iNOS, PPARγ, NRF1, GP-xl*) (Fig. 8f–h). Interestingly, TAs of C57Bl treated intra-arterially with mdx T lymphocytes highlighted necrotic myofibers and regenerating myofibers 8 weeks after transplantation of CD4+$^{mdx}$ cells (Supplementary Fig. 5b, c). Indeed, morphometric analysis showed approximately 3–5% of fibrotic area as described in age-matched untreated C57Bl mice[53,54] (Supplementary Fig. 5b, c).

**Foetal dystrophic thymus transplantation into nude mice determine altered muscle metabolism.** TEC progenitors are active in mice during embryogenesis and their importance in embryonic thymus development and central immune tolerance has been noted previously[55–57]. To further validate the hypothesis of a dysregulated central tolerance in mdx mice, we next explored the impact of transplantation of foetal E17 thymus in nude mice.

Accordingly, nude mice were followed for 120 days after transplantation of E17 thymus of mdx and C57Bl mice underneath the kidney capsule[58] (hereafter referred to as $Tnu^{E17MDX}$

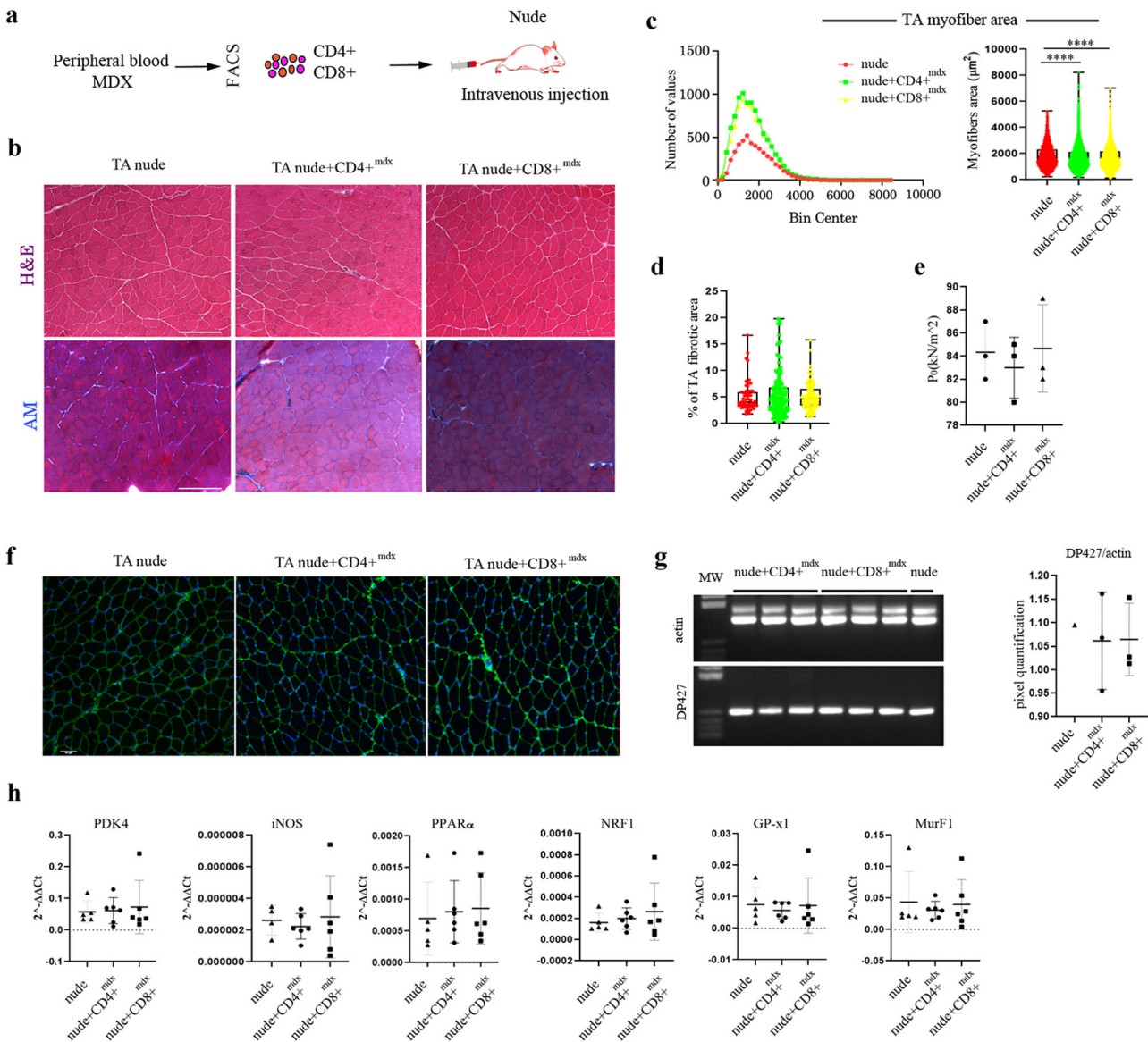

**Fig. 8 Morphometric and functional analysis of skeletal muscles of nude mice following CD4+ and CD8+ lymphocytes transplantation.** Schematic overview of the experimental procedure (**a**). Representative H&E and AM staining of TAs from nude, nude+CD4+$^{mdx}$ and nude+CD8+$^{mdx}$ mice (**b**). Quantification of the relative frequency of the myofiber cross-sectional area (CSA) expressed as the frequency distribution of TA muscles of nude, nude +CD4+$^{mdx}$ and nude+CD8+$^{mdx}$ mice (nude: minimum, median, maximum and range: 247.5, 1666, 5265, 5017, respectively; 25% percentile, 75% percentile, coefficient of variation: 1158, 2320, 46.51%, respectively. nude+CD4+$^{mdx}$: minimum, median, maximum and range: 183.7, 1531, 8239, 8055, respectively; 25% percentile, 75% percentile, coefficient of variation: 1042, 2132, 50.31%, respectively. nude+CD8+$^{mdx}$: minimum, median, maximum and range: 120.9, 1548, 7033, 6912, respectively; 25% percentile, 75% percentile, coefficient of variation: 1065, 2176, 49.21%, respectively) (**c**). Quantification of fibrotic areas of TA muscles of nude, nude+CD4+$^{mdx}$ and nude+CD8+$^{mdx}$ mice (mean area: nude: 5.044; nude+CD4+$^{mdx}$: 5.532; nude+CD8+$^{mdx}$: 5.182) (**d**). For morphometric analysis, images were quantified with ImageJ software for each mouse. Tetanic force of TA of nude, nude+CD4+$^{mdx}$ and nude+CD8+$^{mdx}$ mice is shown in **e**. Representative immunostaining with dys-2 (C-terminal-domain) antibody showed comparable dystrophin intensity around the myofibers in TA of nude, nude+CD4+$^{mdx}$ and nude+CD8+$^{mdx}$ mice (**f**). Representative image of RT-PCR analysis of TA of nude, nude+CD4 +$^{mdx}$ and nude+CD8+$^{mdx}$ mice determined similar expression of *Dp427* dystrophin isoform (**g**). RT-qPCR experiments on TA muscles of treated and untreated mice demonstrated no differences of expression of genes specifically involved in autophagy, skeletal muscle metabolism, mitochondrial biogenesis and muscle atrophy (**h**). Scale bar: 200 μm (**a**), 50 μm (**f**). The comparisons among the averages of the groups were evaluated using one-way ANOVA (**c**). **c** ****$p < 0.0001$. Data are presented as mean ± SD of three independent experiments with $n = 3$ mice/group. Source data are provided as a Source Data file.

and Tnu$^{E17C57Bl}$, respectively) (Fig. 9a). Morphological and immunofluorescence examination of the foetal E17 thymus grafts post-transplantation showed the presence of thymocytes underneath the kidney capsules (Supplementary Fig. 6). The populations of CD3+CD4+, CD3+CD8+ T cells can be clearly detected in circulation of Tnu$^{E17MDX}$ and Tnu$^{E17C57Bl}$ and gradually

increased over time compared to the untreated nude mice (represented as averaged values with dashed black lines) (Fig. 9b–d). Linear regression of CD3+ cell data for Tnu$^{MDX}$ and Tnu$^{E17C57Bl}$ mice also proved the increasing trends compared to nude mice (Fig. 9d). Circulating CD4+ and, partially CD8+ cells, increased over time in Tnu$^{E17MDX}$ (Fig. 9e). No differences

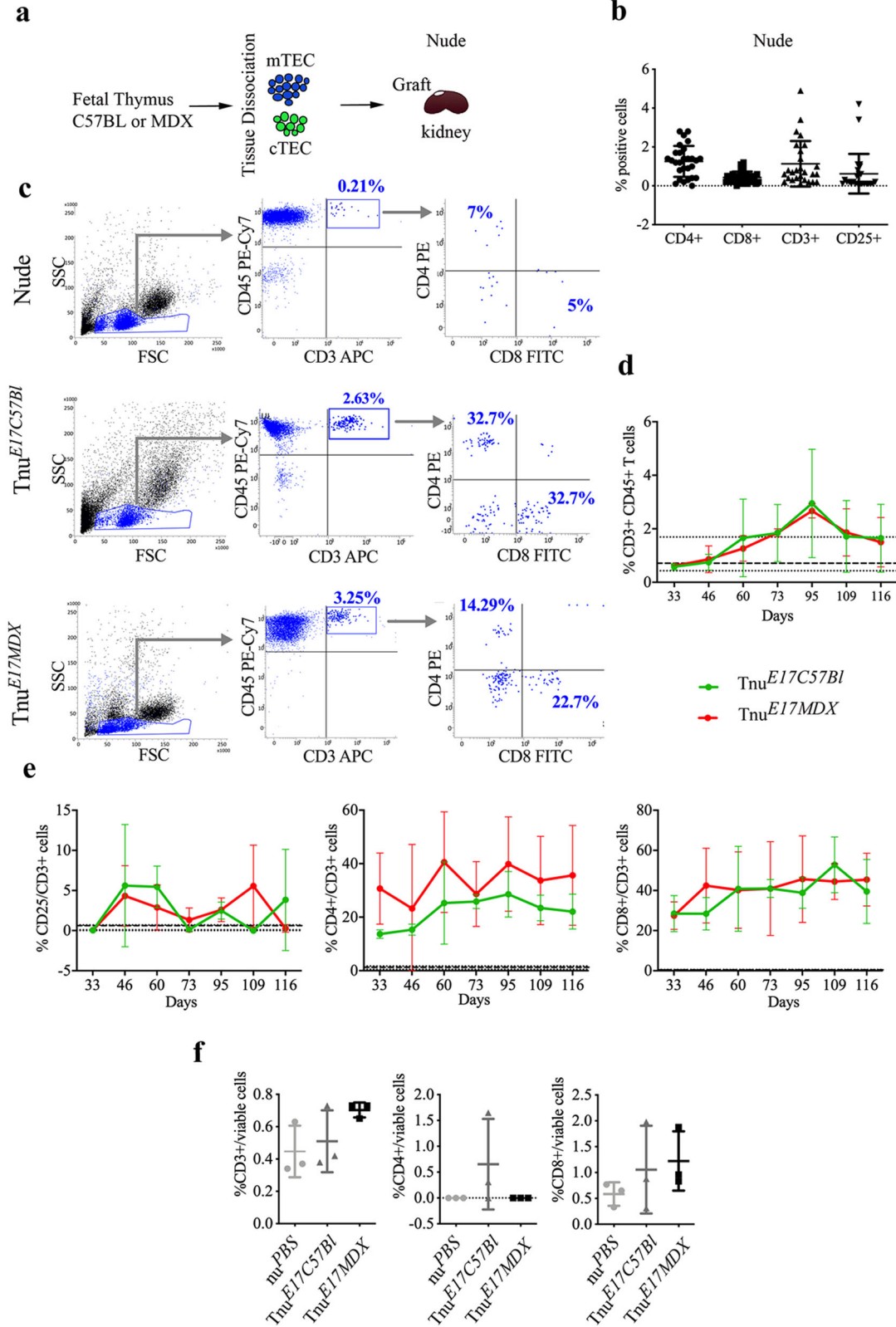

were found in muscle tissues of Tnu$^{E17MDX}$ and Tnu$^{E17C57Bl}$ analysed by cytofluorimetry for CD3+, CD4+ and CD8+ populations (Fig. 9f).

Interestingly, Tnu$^{E17MDX}$ presented significantly smaller CSA with a higher peak of small fibres and strongly reduction of large fibres (over 5000 mm$^2$) compared to the Tnu$^{E17C57Bl}$ and untreated nude mice (Fig. 10a, b). These data are further

supported by significant reduction of the variance of fibres in Tnu$^{E17MDX}$ vs Tnu$^{E17C57Bl}$ and untreated nude (variance Tnu$^{E17C57Bl}$: 810539.3315; variance Tnu$^{E17MDX}$ 636038.7; variance nude 752741.1) (Fig. 10b). Moreover, the Tnu$^{E17MDX}$ had an upregulation of the amount of fibrosis compared to Tnu$^{E17C57Bl}$ (Fig. 10c). Notably, strength evaluation evidenced a dramatic reduction in the tetanic force of TA muscles of

**Fig. 9 Foetal dystrophic thymus transplantation into nude mice determine altered muscle metabolism.** Schematic of the experimental procedure (**a**). Nude mice were transplanted with E17 thymus of mdx (Tnu$^{E17MDX}$) and C57Bl (Tnu$^{C57Bl}$) mice underneath the kidney capsules and sacrificed after 120 days. FACS analysis of blood-derived cells from untreated 8-week-old nude mice confirmed the lack of T cell subpopulations (**b**). Representative image of FACS dot plots showing the expression of CD3+CD4+ and CD3+CD8+ T cells in Tnu$^{E17MDX}$ and Tnu$^{E17C57Bl}$. Lack of circulating CD3+CD4+ and CD3+CD8+ T cells in untreated nude mice was also confirmed (**c**). FACS analysis of transplanted mice showed an increasing percentage of circulating CD3+ cells in Tnu$^{E17MDX}$ and Tnu$^{E17C57Bl}$ compared to the untreated nude mice (represented as averaged values with dashed black lines) (**d**). Circulating CD3+CD4+ and CD3+CD8+ T cells gradually increased over time in Tnu$^{E17MDX}$ whereas in Tnu$^{E17C57Bl}$ compared to the untreated nude mice (represented as averaged values with dashed black lines) (**e**). No differences were found in muscle tissues of Tnu$^{E17MDX}$ and Tnu$^{E17C57Bl}$ analysed by FACS for CD3+, CD4+, and CD8+ populations (**f**). The comparisons among the averages of the groups were evaluated using Linear regression analysis (**d**). **d** ****$p < 0.0001$. Data are presented as mean ± SD of three independent experiments with $n = 4$ with seven time-points each (**b**); $n = 4$ (**d, e**) and $n = 3$ (**f**) mice/group. Source data are provided as a Source Data file.

Tnu$^{E17MDX}$ mice compared to Tnu$^{E17C57Bl}$ and untreated nude while no significant difference was observed between nude and Tnu$^{E17C57Bl}$ (Fig. 10d). Weight variation of grafted mice only evidenced a tendency to loss of weight of Tnu$^{E17MDX}$ vs Tnu$^{E17C57Bl}$ (Fig. 10e). Although dystrophin protein was similarly detected in muscles of Tnu$^{E17MDX}$ mice, Tnu$^{E17C57Bl}$ and untreated nude mice, we found a significant reduction of full-length DP427 dystrophin mRNA in Tnu$^{E17MDX}$ mice compared to Tnu$^{E17C57Bl}$ and untreated nude (Fig. 10f, g). Next, we analysed the presence of damage muscle markers in the serum of grafted Tnu$^{E17MDX}$, Tnu$^{E17C57Bl}$ and untreated mice as control. We observed significant increase of AST and ALT suggesting a sarcolemma fragility in Tnu$^{E17MDX}$ (Fig. 10h). However, RT-qPCR analysis of muscle tissues validated the significant over-expression of inflammatory muscle markers in Tnu$^{E17MDX}$ (Fig. 10i). Notably, Tnu$^{E17MDX}$ muscle tissues showed significant increased expression of metabolic muscle markers as *PDK4*, *GP-x1* and *PPARα* (Fig. 10j). Mitochondrial biogenesis was also affected in Tnu$^{E17MDX}$ by increased expression of *PGC1α* and *NRF-1* (Fig. 10k). Similarly, we found different expression of electron transport chain genes such as *CoxVa* and *CoxVIIb* (Fig. 10l). These data therefore suggest that transplantation of dystrophic E17 thymus prevalently causes altered muscle metabolism.

## Discussion

Inflammation and inflammatory cells, mainly consisting in macrophages, are characteristic features of dystrophin-deficient muscles. Lymphoid cells are also present in the muscle infiltrates where they serve immunomodulatory and cytolitic functions[6]. In particular, CD8+ T lymphocytes occur in elevated numbers in dystrophic muscle suggesting an adaptive immune response, which may contribute significantly to the pathology of DMD. T lymphocytes are rich sources of cytokines and can influence the shift of macrophages to pro- vs. anti-inflammatory phenotype (M1 or M2, respectively) playing opposite roles in muscle repair[59,60]. Interestingly, regulatory T cells (Tregs) participate in muscle regeneration through released cytokines as IL-10 (ref. [9,61]). However, acute inflammatory response to chronic muscle damages in DMD stimulates the innate immunity which provides a rapid mechanism against released muscle DAMPs. The complexity of the immune response to DMD may be attributable to the combination of innate and adaptive immune response which can occur independently or concurrently[62]. Since immunocompetent T cells and Tregs are necessary for the maintenance of immune tolerance and mainly originate in the thymus[63–65], we investigated the role of dystrophic thymus in DMD immunity. Interestingly, we showed a severe involution with chaotic architecture of thymus of 3-month-old mdx mice. Different works demonstrated the fundamental role of NF-kB and STAT3 in the positive selection of CD8+ thymocytes[33], in the regulation of self-

tolerance and autoimmunity[66] and thymus architecture[67]. Here, we also observed a severe impairment in mdx thymus of AIRE and SIRT-1 expression, two master regulators of the immunological self-tolerance. It is known that mutations in *AIRE* cause the autoimmune polyendocrinopathy candidiasis ectodermal dystrophy (APECED): patients suffer from muscle disturbances that resemble those of limb-girdle myopathy[68,69]. Surprisingly, thymus of mdx mice presented significantly a lower amount of ghrelin receptor. While the stomach is considered the major source of peripheral ghrelin, recent studies have demonstrated ghrelin to be widely distributed in organ systems including thymus[22]. The loss of ghrelin protein expression in the thymus with age strongly correlated with thymic involution characterized by impaired architecture and less defined cortical and medullary regions and adiposity[18,70]. Moreover, ghrelin is also involved in T cell maturation and emigration of T cells from thymus[19,71]. To test the prevalence of innate or adaptive immunity in DMD and the role of thymus in this framework, we created a model of foetal and adult thymus transplantation into nude mice. In these experiments we found reduced myofiber area, loss of strength and fibrosis in muscles of nude mice that received thymi of mdx mice. Interestingly the percentage of circulating Treg decreased over time in Tnu$^{E17MDX}$ and Tnu$^{MDX}$. Genetic mutations that affect the development and function of mTECs can compromise T cell tolerance. For example, deletion or mutation of RANK[72,73] leads to defective differentiation of mTECs, resulting in a reduced or absent medulla in mice[74]. Overall, these results suggested that the disorganization of grafted dystrophic thymi of mdx activates a diverse stromal repertoire leading to muscle infiltration of CD4+ and CD8+ lymphocytes and release of pro-inflammatory cytokines in Tnu$^{MDX}$ and Tnu$^{E17MDX}$ mice. Moreover, muscles of Tnu$^{MDX}$ and Tnu$^{E17MDX}$ mice presented metabolic dysfunctions, as determined by the over-expression of *pdk4*—that coordinates the glucose oxidation and it is over-expressed in patients with a drastic skeletal muscle atrophy[75]—and *GPx1*, in accordance with data showed by Messina et al.[76] in DMD muscular biopsies. More interestingly, we showed that immune activation (and the subsequent increased expression of cytokines) was linked to loss of skeletal muscle mass and strength. This condition determined an upregulation of TNF-α and IP pathway in muscle of transplanted nude mice. We found in Tnu$^{MDX}$ and Tnu$^{E17MDX}$ mice that TNF-α promoted muscular atrophy with a mechanism involving the activation of MuRF-1 and increasing proteasomal activity and oxidative state, as described in ref. [39]. Recently, Li et al.[77] showed that the activity of the same pathway could be increased by ROS production and p38 activation. In parallel, TNF-α could have a role in regulating the autophagic machinery[78]. In fact, we found dysregulation of several autophagy mediators as LC3/p62/TRAF-6/iNOS whose functional connections are largely known[47,48,79] and may explain the thymus involution in DMD. Collectively, these data indicate a significant role of dystrophic thymus in modulating immune cell functions in DMD.

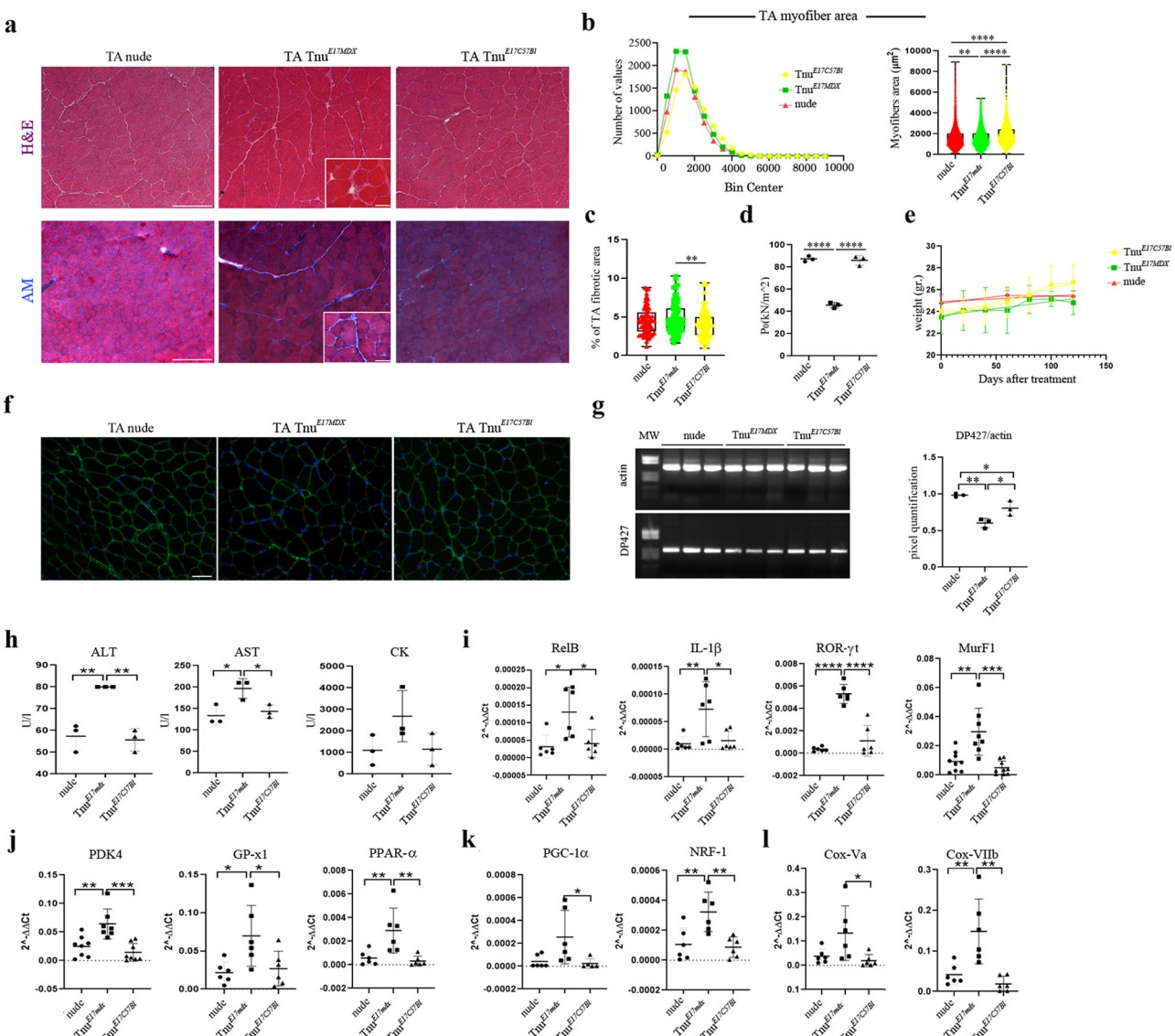

**Fig. 10 Morphometric and functional analysis of skeletal muscles of nude mice following foetal thymus transplantation.** Representative images of H&E and AM staining of TAs from nude, Tnu$^{E17MDX}$, Tnu$^{E17C57Bl}$ mice (**a**). Quantification of the relative frequency of the myofiber CSA expressed as the frequency distribution of TA muscles of nude, Tnu$^{E17MDX}$, Tnu$^{E17C57Bl}$ mice (Tnu$^{E17C57Bl}$: minimum, median, maximum and range: 98.07, 1763, 8676, 8578, respectively; 25% percentile, 75% percentile, coefficient of variation: 1231, 2434, 47.44%, respectively. Tnu$^{E17MDX}$: minimum, median, maximum and range: 81.53, 1421, 5449, 5367 respectively; 25% percentile, 75% percentile, coefficient of variation: 962.9, 2021, 51.27%, respectively. nude: minimum, median, maximum and range: 101.7, 1461, 8946, 8845, respectively; 25% percentile, 75% percentile, coefficient of variation: 982.2, 2029, 54.19%, respectively) (**b**). Quantification of fibrotic areas of TA muscles of nude, Tnu$^{E17MDX}$, Tnu$^{E17C57Bl}$ mice (mean area: nude: 4.534; Tnu$^{E17MDX}$: 4.850; Tnu$^{E17C57Bl}$: 4.055) (**c**). For morphometric analysis, images were quantified with ImageJ software for each mouse. Tetanic force of TA of Tnu$^{E17MDX}$ is dramatically decreased compared to nude and Tnu$^{E17C57Bl}$ mice (**d**). Weight of mice following E17 thymus transplantation is reported in the graph (**e**). Representative immunostaining with dys-2 (C-terminal-domain) antibody showed weak dystrophin intensity around the myofibers in TA of Tnu$^{E17MDX}$ (**f**). Representative image of RT-PCR analysis described lower expression of *Dp427* dystrophin isoform in TA of Tnu$^{E17MDX}$ compared to nude and Tnu$^{E17C57Bl}$ mice (**g**). ALT, AST and CK are measured in the serum of nude, Tnu$^{E17MDX}$, Tnu$^{E17C57Bl}$ mice (**h**). RT-qPCR experiments on TA muscles of nude, Tnu$^{E17MDX}$, Tnu$^{E17C57Bl}$ mice showed differences of expression of genes specifically involved in inflammation/fibrosis and atrophy (**i**); skeletal muscle metabolism (**j**); mitochondrial biogenesis (**k**) and oxidative capacity (**l**). Scale bar: 200 and 40 μm for higher magnification images in the inserted squares (**a**); 50 μm (**f**). The comparisons among the averages of the groups were evaluated using one-way ANOVA (**b**–**l**). **b** **$p = 0.0017$ and ****$p < 0.0001$. **c** **$p = 0.0090$. **d** ****$p < 0.0001$. **g** *$p = 0.0282$ Tnu$^{E17MDX}$ vs Tnu$^{E17C57Bl}$; **$p = 0.0013$ Tnu$^{E17MDX}$ vs nude; *$p = 0.0459$ Tnu$^{E17C57Bl}$ vs nude. **h** AST: *$p = 0.0459$ Tnu$^{E17MDX}$ vs Tnu$^{E17C57Bl}$; *$p = 0.0228$ Tnu$^{E17MDX}$ vs nude; ALT: **$p = 0.0018$ Tnu$^{E17MDX}$ vs Tnu$^{E17C57Bl}$; **$p = 0.0027$ Tnu$^{E17MDX}$ vs nude. **i** *MurF1*: ***$p = 0.0001$, **$p = 0.0011$; *RORγt*: ****$p < 0.0001$; *IL-1β*: *$p = 0.0174$ Tnu$^{E17MDX}$ vs Tnu$^{E17C57Bl}$; **$p = 0.0094$ Tnu$^{E17MDX}$ vs nude; *RelB*: *$p = 0.0110$ nude vs Tnu$^{E17MDX}$; *$p = 0.0203$ Tnu$^{E17MDX}$ vs Tnu$^{E17C57Bl}$. **j** *PDK4*: **$p = 0.0031$ and **$p = 0.0003$; *GP-x1*: *$p = 0.0431$ Tnu$^{E17MDX}$ vs Tnu$^{E17C57Bl}$, *$p = 0.0231$ Tnu$^{E17MDX}$ vs nude; *PPARα*: **$p = 0.0047$ Tnu$^{E17MDX}$ vs Tnu$^{E17C57Bl}$, **$p = 0.0099$ Tnu$^{E17MDX}$ vs nude. **k** *PGC1α*: *$p = 0.0489$; *NRF-1*: **$p = 0.0045$ Tnu$^{E17MDX}$ vs Tnu$^{E17C57Bl}$, **$p = 0.0078$ Tnu$^{E17MDX}$ vs nude. **l** *CoxVa*: *$p = 0.0295$; *CoxVIIb*: **$p = 0.0011$ Tnu$^{E17MDX}$ vs Tnu$^{E17C57Bl}$; **$p = 0.0052$ Tnu$^{E17MDX}$ vs nude. Data are presented as mean ± SD of three independent experiments with $n = 3$ (**b**–**d**); $n = 4$ (**e**); $n = 3$ (**f**–**h**); $n = 3$ with two technical replicates (*RelB, IL-1β, RORγt*) and with two/three technical replicates (*murf-1*) each (**i**); $n = 3$ with two technical replicates (*GP-x1, pparα*) and with two/three technical replicates (*pdk-4*) each (**j**); $n = 3$ with two technical replicates (**k**) mice/group. Source data are provided as a Source Data file.

## Methods

**Animal ethics statement.** Procedures involving living animals were conformed to Italian law (D.L.vo 116/92 and approved by local ethics committees). This work was authorized by the Ministry of Health and Local University of Milan Committee, authorization number 859/2017-PR (5247B.35, 10/07/2017 and additional integration). C57Bl (8-week-old and 3-month-old), mdx (8-week-old and 3-month-old) and BALB/c nude (8-week-old) mice were provided by Charles River and housed in a controlled ambient environment (12 h light/dark cycle) at a temperature between 21 °C/23 °C. All the mice were males except for four C57Bl and four mdx females that were used for mating. The mice had free access to clean water and food.

**FACS analysis and sorting of cells from murine peripheral blood, muscles and thymus.** For experiments involving the characterization and isolation of circulating T cell, peripheral blood (100 μl) was collected from the mouse tail veins. Red blood cells were lysed with ACK solution (NH$_4$Cl 150 mM, KHCO$_3$ 10 mM and Na$_2$EDTA 0.1 mM) to allow cytofluorimetric studies. For five-colours flow cytometry 10$^5$ cells were resuspended in phosphate-buffered saline (PBS) and incubated with 10 μl primary antibodies anti-CD4-phycoerythrin (anti-CD4-PE), anti-CD3-allophycocyanin (anti-CD3-APC), anti-CD8-fluorescein-isothiocyanate (anti-CD8-FITC), anti-CD25-Peridinin Chlorophyll Protein Complex Cy5.5 (anti-CD25-PerCP-Cy5.5) (BD Biosciences-Pharmingen, CA, USA), and anti-7-amino-actinomycin D (anti-7AAD) (BD Biosciences-Pharmingen, CA, USA). The controls were isotype-matched mouse immunoglobulines. After each incubation performed at 4 °C for 20 min, cells were washed in PBS containing 1% heat-inactivated FCS and 0.1% sodium azide. For FACS characterization, data were acquired with the Cytomics FC500 (Beckman-Coulter) machine and analysed with CXP 2.1 software. Each analysis included at least 5–10 × 10$^4$ events for each gate. A light-scatter gate was set up to eliminate cell debris from the analysis. The percentage of positive cells was assessed after correction for the percentage reactive to an isotype control conjugated to relative fluorochromes. The 7AAD was added to exclude non-viable cells from the analysis.

For the examination of thymus cellularity, once harvested from 3-month-old mdx ($n = 7$) and C57Bl mice ($n = 5$), thymi were depleted from fat and connective tissue, transferred to six-well plate containing Liberase (Invitrogen) solution and incubated at 37 °C for 20 min. Following dissociation as described in detail by Xing and Hogquist[80] we identified the main thymic cellular subpopulations according to the combined expression of CD4 (Pacific Blue-A) and CD8 (APC-Cy7-A). CD4+/CD8+ DP cells were subsequently characterized for the expression of CD69 (Alexa Fluor 488-A) and TCRbeta (PE-A). Similarly, the DN cells were studied for the expression of CD44 (Alexa Fluor 488-A) and CD25 (APC-A) while the CD4+ subpopulation was characterized for Foxp3 (Alexa Fluor 488-A) and CD25 (APC-A) in order to identify the T-regs. All the antibodies are from Biolegend (San Diego, CA, USA). For the isolation of cTEC and mTEC, cells isolated from 3-month-old C57bl and mdx mice ($n = 8$/animal group) were immediately enriched by thymocyte depletion—since thymus-derived cells are mainly composed of over 95% thymocytes. Accordingly, these cells were incubated with anti-CD45 antibody at a final concentration of 2.5 μg/ml: thymocytes depleted TECs were then resuspended in FACS sorting buffer and centrifuges. Cells were incubated with an antibody cocktail containing anti-CD45 (Pacific Blue, clone 30-F11), anti-EpCAM (PE-Cy7), anti-MHC-class II (PerCP/Cy5.5), anti-Ly51 (FITC)—all from Biolegend—and the UEA-1 fluorescein (from Vector Laboratories, Burlingame, CA, USA) and analysed by FACS. cTEC and mTEC were then pulled randomly in two groups/animal ($n = 4$ animals per group of C57Bl and mdx mice) with similar number of cells in order to have enough amount of sample to perform RNA extraction and RT-qPCR experiments.

For sorting, cells from peripheral blood were isolated with A-FACS Aria machine (BD Bioscience, New Jersey). For FACS characterization of lymphocytes from muscles, TA, gastrocnemius and QA muscles were excised and extensively washed in PBS to removed blood contaminants[9]. Muscles were cut in small pieces, digested for 1 h with Liberase 0.2 mg/ml (Invitrogen) and filtered with 70 μm mesh filters. Undigested tissues were mashed with a plunger through the filters and washed with DMEM in addition to serum. Histopaque (Sigma Aldrich) gradient was performed to separate lymphocyte fraction and centrifuges for 25 min. The T cells containing interphase was aspirated carefully, washed in PBS and stained with antibody cocktails as already described.

**Imaging mass spectrometry.** Adult thymus tissues of 3-month-old mdx and C57Bl mice were frozen for preparation of cryosections (thickness of 10 μm) with the use of a cryostat (CM 1900; Leica Microsystems, Wetzlar, Germany). For imaging mass spectrometry, the sections were thaw-mounted on indium–tin oxide slides (Bruker Daltonik, Bremen, Germany), dried in silica gel-containing plastic tubes and then sprayed with 9-aminoacridine (5 mg in 4 ml of 80% ethanol) with the use of a 0.2-mm nozzle calibre airbrush (Procon Boy FWA Platinum; Mr Hobby, Tokyo, Japan) for matrix-assisted laser desorption–ionization (MALDI) imaging mass spectrometry in negative-ion mode. Adjacent sections were stained with H&E. Imaging mass spectrometry was performed with iMScope TRIO Mass Microscope (Shimadzu, Kyoto, Japan). MALDI mass spectra were acquired with a laser diameter of 50 μm, 200 shots/spot, scanning pitch of 20 μm, and scanning $m/z$

range of 615–931. Regions of tissue samples exposed to the laser radiation were determined by light and fluorescence microscopic observations.

**Transplantation of animals**

*Subcutaneous injections.* Nude mice were injected with thymus isolated from 3-month-old C57Bl or mdx mice as described in details in ref. [81], and sacrificed 120 days after transplantation. Briefly, thymi were collected from mdx and C57Bl mice and placed individually into sterile tubes with PBS, then minced roughly with scissors: the fragments were implanted subcutaneously into the scruff of nude mice. Each mouse received equal amount in weight of thymic tissue. Transplanted nude mice were regularly tail-bled. Nude mice transplanted with PBS were used as controls. Two independent experiments of subcutaneous thymic transplantation were performed for a total of nu$^{PBS}$ $n = 8$, Tnu$^{MDX}$ $n = 9$, Tnu$^{C57BL}$ $n = 8$.

*Intra-arterial and intra-tail vein injections.* Three-month-old C57Bl mice were transplanted into the femoral artery with CD3+CD4+ (CD4+$^{mdx}$, $n = 3$) or CD3+CD8+ (CD8+$^{mdx}$, $n = 3$) T lymphocytes (5 × 10$^5$ cells/leg) were isolated from 3-month-old mdx peripheral blood as described in detail in ref. [82]. Untreated C57Bl mice ($n = 3$) were used as controls. Similarly, 8-week-old nude mice were transplanted into tail vein with CD4+$^{mdx}$ ($n = 3$) or CD8+$^{mdx}$ ($n = 3$) (5 × 10$^6$ and 2.9 × 10$^6$ cells, respectively). Untreated nude mice were used as control ($n = 3$). Mice were sacrificed 8 weeks after transplantation.

*Kidney capsule transplants.* To assure that muscular degeneration observed in transplanted mdx nude mice depends on dysfunctions of mdx thymic stroma, we transplanted 8-week-old nude mice with thymic stroma from mdx or C57Bl embryos under the kidney capsule. $N = 4$ nude mice were transplanted for each embryonic cell type. Untreated nude mice ($n = 4$) were used as controls. Briefly, we isolated thymus from E17 embryos of mdx or C57Bl pregnant mice, we exposed kidney of recipient nude mice and we injected embryonic tissues under the renal capsule, as described in ref. [83]. Transplanted nude mice were regularly tail-bled. Mice were sacrificed at 120 days after transplantation.

**WB analysis.** Total proteins from skeletal muscles isolated from transplanted nude mice (nu$^{PBS}$ $n = 4$; Tnu$^{MDX}$ $n = 4$; Tnu$^{C57BL}$ $n = 4$) were obtained as in ref. [84]. Samples were resolved on polyacrylamide gels (ranging from 6 to 10%) and transferred to nitrocellulose membranes (Bio-Rad Laboratories, CA, USA). Filters were incubated overnight with following antibodies: Vinculin (1:600, MA5-11690, Invitrogen); Actin (1:600, A2066, Sigma Aldrich); GAPDH (0411) (1:600, sc-47724, Santa Cruz Biotechnology); Phospho-p38 (Thr180) (1:500, ab195049, Abcam); p38 (1:500, ab31828, Abcam); PSMB5 (1:500; ab3330, Abcam); PSMB8 (1:500, Proteasome 20S LMP7, ab3329, Abcam); PSMB9 (1:500; Proteasome 20S LMP2 [EPR13785] ab184172, Abcam); dystrophin (1:50, Novocastra, UK); Collagen I (1:500, ab6308, Abcam); TRAF6 (D-10) (1:500, sc-8409, Santa Cruz Biotechnology); IKKi (A-11) (1:500, sc-376114, Santa Cruz Biotechnology); NF-kB p65 (A-12) (1:500, sc-514451, Santa Cruz Biotechnology); TGFβ (1:500, E-AB-33090, Elabscience); TNFα (1:500, E-AB-40015, Elabscience); p62 (1:600, P0067, Sigma Aldrich); OPN (1:550, R&D); Ghrelin (1:500, PA1-1070, Invitrogen); GHS-R (1:500, PA5-28752, Invitrogen); LC3B (1:500, L7543, Sigma Aldrich); IL-10 (1:500, sc-1783, Santa Cruz Biotechnology); AKT (1:500, ab179463, Abcam); ATG7 (1:500, 126M4822V, Sigma Aldrich). Dystrophic ($n = 6$) and wild-type ($n = 6$) thymi were characterized for: LC3B; Cytokeratin 14–16 (1:500, PA5-36061, Invitrogen); STAT1 (1:500, ab47425, Abcam); Phospho-STAT1 (1:500, ab10946, Abcam); STAT3 (1:500, ab68153, Abcam); Phospho-STAT3 (1:500, ab76315, Abcam); AIRE (1:500, 14-5934-82, 5H12, eBioscience); SIRT-1 (1:500, 2192247, Millipore); NF-kB; autophagy markers (LC3B, p62, ATG7); dystrophin. Page ruler plus pre-stained protein ladder (Thermo Fisher #26619) and Protein Kaleidoscope precision plus (Bio-rad #161-0375) were used as molecular weight markers.

Membranes were incubated with primary antibodies ON at 4 °C, then followed by washing, detection with horseradish peroxidase (HRP)-conjugated secondary antibodies (DakoCytomation, USA) and developed by enhanced chemiluminescence (Amersham Biosciences, USA). Bands were visualized using an Odyssey Infrared Imaging System (Li-COR Biosciences, USA). Densitometric analysis was performed using ImageJ software version 1.46i (http://rsbweb.nih.gov/ij/).

**Histological analysis.** For immunohistochemistry and immunofluorescence analysis murine tissues were collected from treated and untreated animals, frozen in liquid nitrogen cooled isopentane and cut on a cryostat into 8 μm. AM and H&E staining were performed as described ref. [7]. Thymi for 3-month-old C57Bl ($n = 6$) and mdx ($n = 6$) mice were characterized by immunofluorescence staining to determine the area occupied by cortex (cytokeratin 8, CK8) vs. medulla (cytokeratin 5, CK5; AIRE) antigens, dystrophin isoforms (DYS-1 and DYS-2) and the presence of T cells (CD3 and Foxp3). Expression of dystrophin and collagen X was also evaluated on TAs of transplanted nude mice. With the exception of dystrophins, sections were fixed with 4% paraformaldehyde for 10 min, permeabilized with 0.3% Triton X-100 for 15 min and incubated with 10% donkey serum to block non-specific binding for 1 h. Slides were then incubated with the primary antibodies (overnight at 4 °C) diluted in blocking solution. Fluorochrome-conjugated secondary antibodies were diluted in PBS and added for 1 h at room temperature.

For dystrophin staining, slides were not fixed, and non-specific binding was blocked with 2% horse serum and 5% foetal bovine serum. Primary antibodies were used at the following dilutions: CK5, CK8 and collagen X 1:150 (Abcam, Cambridge UK), FOXp3 and AIRE 1:100 (Thermo Fisher, Carlsbad, CA), CD3 1:50 (Abcam), dystrophin isoforms (NCL-DYS-1 and NCL-DYS-2, Novocastra, Wetzlar, Germany). Slides were then mounted with Prolong Gold® Antifade Reagent with DAPI (Thermo Fisher, Carlsbad, CA). A Leica SP8 confocal microscope and a Leica DMi8 fluorescence microscope were used for acquiring images. mTECs were also detected by immunohistochemistry with a biotinylated UEA-1 antibody (Vector Laboratories, Burlingame, CA), diluted 1:100. Endogenous peroxidase activity was blocked in 0.3% alcoholic hydrogen peroxide for 30 min, followed by antigen retrieval in 0.01 M sodium citrate buffer for 1 h, at pH 6 and 100 °C. Sections were blocked with 2% horse serum and 5% foetal bovine serum for 30 min at room temperature and incubated with UEA-1 ON. HRP-conjugated ABC kit (Vector Laboratories) was used to detect the primary antibody. Nuclei were counterstained with haematoxylin for 1 min and slides mounted with DPX (Sigma Aldrich, St. Louis, Missouri).

To investigate the morphology of transplanted nude muscles, we evaluated the percentage of fibrosis by means of AM staining.

To investigate the morphology of transplanted nude muscles, we evaluated the percentage of fibrosis by means of AM staining. We quantified the amount of necrosis and regenerating fibres as in ref. [85]. Briefly, we performed the immunohistochemistry with anti-mouse IgG antibody, and we counted the IgG+ fibres/total number of fibres. The regenerative fibres were easily identified as those that are small and centrally nucleated in H&E staining.

**Myofibrillar ATPase histochemistry**. Myofibrillar ATPase staining with pre-incubation at pH 4.3 was used to identify fibre types—slow, fast (IIa, IIx, IIb), intermediate—in TA of transplanted nude mice (nu$^{PBS}$ $n = 4$; Tnu$^{MDX}$ $n = 4$; Tnu$^{C57BL}$ $n = 4$) as described in ref. [86]. After staining, muscle sections were washed with tap water, dehydrated with ethanol, cleared in xylene and mounted as performed in ref. [87]. Different images were analysed by graph distribution of grey values in the muscle-selected area—presented as pixel counts for each grey value—and each value was assigned to specific intervals, representing different fibre types.

**Analysis of tetanic force**. Tetanic force of TA of transplanted nude mice (nu$^{PBS}$ $n = 3$; Tnu$^{MDX}$ $n = 3$; Tnu$^{C57BL}$ $n = 3$ was determined as described in ref. [7] and expressed as kN/m$^2$. Same protocol was applied for TA of nude mice (nude $n = 3$, nude+ CD4+$^{mdx}$ $n = 3$; nude+ CD8+$^{mdx}$ $n = 3$). Tetanic force of C57Bl ($n = 4$) and mdx ($n = 4$) mice were also measured for comparison.

**Quantitative (RT-PCR) and qualitative (RT-qPCR) experiments**. Total RNA was extracted from skeletal muscles of transplanted nude mice and cDNA generated using the Reverse Transcriptase Kit (Thermo Fisher Scientific, California, USA). Classical RT-PCR was carried out with 2 μg of cDNA using Invitrogen kit with a mix constituted of 1× Taq buffer, 1.5 mM MgCl₂, 0.2 mM dNTPs, 2.5 units of the Platinum Taq DNA polymerase, 0.2 mM of primers specific for dystrophin isoforms (Supplementary Table 1). Thirty-six cycles of amplification (94 °C/2 min, 92 °C/1 min, 60 °C/2 min, and 72 °C/2 min) were performed and PCR products analysed on 2% agarose gels. The PCR products that failed to show the bands were purified using the Jetquick PCR product purification spin kit (Genomed) and a second round of amplification (30 cycles of amplification: 94 °C/2 min, 92 °C/1 min, 59 °C/2 min, 72 °C/2 min) was performed: we used 5 ml of purified PCR and the same mix except for 1.5 mM MgCl₂. Again, PCR products were analysed on 2% agarose gels.

We quantified the expression of genes through SYBR-Green method. All the samples were tested in duplicate and the threshold cycles (Ct) of target genes were normalized against the housekeeping gene, glyceraldehyde 3-phosphate dehydrogenase (GAPDH). Relative transcript levels were calculated from the Ct values as $X = 2^{-\Delta\Delta Ct}$ where $X$ is the fold difference in amount of target gene versus GAPDH and $\Delta Ct = Ct_{target} - Ct_{GAPDH}$. The sequence of primers used is listed in Supplementary Table 2. The sequences of primers for AIRE experiments described in Fig. 3 were gently provided by Takayanagi group[88] while the sequence of primer for the RT-qPCR described in Supplementary Fig. 1 were those of Youm et al.[22].

**Statistics and reproducibility**. Histological images were captured by a Leica microdissector, a fluorescent microscope and confocal microscopy. Quantitative analyses were performed by ImageJ Software (NIH) version 1.46i. Threshold colour Plug in of ImageJ Software was used to quantify the Azan Mallory staining as the percentage of area over a fixed grid area and the percentage of fluorescence signal for GSH-R immunofluorescence staining. Representative micrographs experiments were repeated independently three times with similar results. Data were analysed by GraphPad Prism$^{TM}$ and expressed as means ± SD. To determine the significance of the variation of cellular concentration throughout the time, we used the linear regression for repeated measures. To compare multiple group's means, one-way ANOVA followed by Tukey's multiple comparison test was used to determine significance (*$p < 0.05$, **$p < 0.01$, ***$p < 0.001$; ****$p < 0.0001$). To compare two groups, two-sided Student's $t$-test was applied assuming equal variances.

The difference among groups was considered significant *$p < 0.05$; **$p < 0.01$; ***$p < 0.001$; ****$p < 0.0001$.

**Reporting summary**. Further information on research design is available in the Nature Research Reporting Summary linked to this article.

## Data availability
The authors declare that the data supporting the findings of this study are available within the paper and its supplementary information files. Source data are provided with this paper.

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

## Acknowledgements

This study was supported by the Associazione Centro Dino Ferrari, a French Telethon AFM grant (No. 21104), Ricerca corrente FR230 Policlinico Hospital. This paper presents independent research funded by Roby and OPSIS Foundations. Work in Torrente lab also received support from Italian Regenerative Medicine Infrastructure (IRMI)—Italian Ministry of Education, Universities and Research (MIUR), Ricerca Finalizzata 2016 (Linea di ricerca: "Theory-enhancing"). Funders of the study had no role in study design, data analysis, data interpretation, or writing of the report. We thank Ann-Christin Niehoff from Shimadzu for the outstanding assistance in the acquisition and interpretation of Imaging mass spectrometry data. We thank Silvia Erratico from Novystem for technical and scientific assistance.

## Author contributions

Y.T., A.F., C.S. and M.C. conceived and designed the experiments. A.F., Y.T., C.V. and M.C. wrote the paper. B.C. and C.V. interpreted and analysed the data. S.G. and A.F. performed animal transplantation. C.V., M.B., P.B., C.L., M.L., L.T., B.C. and S.G. performed the experiments and acquired the data. All the authors stated were involved in the critical revision of the manuscript and approved the final version of the article, including the authorship list. The corresponding author had full access to all the data in the study and had final responsibility for the decision to submit for publication.

## Competing interests

The authors declare no competing interests.
