## [Peer Review File · Nature Communications]

Reviewers' comments:

Reviewer #1 (Remarks to the Author):

Minor points:

- There are a number of grammatical errors that make the paper difficult to read. The authors would be well served to have a collaborator or writing center help edit for grammar.
- In the introduction the authors state: "Mendell et al demonstrated that the majority of DMD patients presented CD4+ T lymphocytes directed against self dystrophin epitopes, whose number was inversely correlated with steroid treatment."
However, only 2/6 (not a majority of) patients had self-primed immune responses. 1 of those patients was not on steroids and 1 was, but another patient not on steroids did not have a pre-existing response to dystrophin, so I'm not really sure the numbers are there to make this assertion. The Flanigan paper did show that patients on steroids were less likely to have self-directed anti-dystrophin T cell responses, although responses were still seen in less than a majority of patients: Flanigan KM1, Campbell K, Violet L, Wang W, Gomez AM, Walker CM, Mendell JR. Anti-dystrophin T cell responses in Duchenne muscular dystrophy: prevalence and a glucocorticoid treatment effect. Hum Gene Ther. 2013 Sep;24(9):797-806. doi: 10.1089/hum.2013.092.
- Figure 1F: does yellow indicate co-staining? Assuming so, mention that in the figure legend.
- Fig 2A: In the text description of this figure the authors state that the C57 and mdx thymic populations are significantly different from one another in the DP and SP populations, but there is no such indication on the graph, and the figure legend states otherwise (and the dot plot doesn't appear to show a significant difference).
- The authors change the style of their graphs not only throughout the paper or a figure, but even within a graph itself (ex Fig 2F). Sometimes the individual animals are shown and sometimes what appears to be the average of each individual experiment is graphed, and presumably those numbers are what the statistics were run on. However, running a student T test on experimental averages is a poor method since it greatly alters the calculated deviations. The authors should make all of their graphs dot plots, show all the animals (not the experimental averages), and run student T tests on all of their animals rather than treating experimental averages as individual data points. Otherwise it appears that data is being obscured and p values manipulated.
- "According to these results the expression of Aire-dependent genes Ins-2 and Spt-1 was significantly impaired in mdx thymus, whereas Mup4 and S100a8 showed similar trend even not statistically significant"—this is not a true statement. In Fig 3D the average expression values of s100a8 is HIGHER in mdx mice. Additionally, the authors then state that these trended toward significance while RANK and CD40 differences did not, yet the mup4 differences don't seem strikingly different than the differences seen with RANK and CD40. I do not think it is appropriate to draw conclusions with RANK, CD40, mup4, and s100a8, other than just to say they are not significantly different.
- "We found marked expression of dystrophin in C57Bl thymus...while dys-1 expression was dramatically decreased and dys-2 expression was absent..." The authors are measuring dys-1 and dys-2 detection, not expression, as these are antibodies. These antibodies may have different sensitivities, so the authors cannot say for sure that expression is absent.
- Figure 4B, C, D: The authors mention in the paper text (but not the figure legend) that for 4C the PCR products were purified. Presumably this was not done for 4B. What about for 4D? If not, why not? The legend should clearly indicate whether the PCR products were purified.

Major points:

- The figure legend for Fig 3 indicates that 3A has the densitometric analyses for AIRE and SIRT-1,

but only the pixel quantification for AIRE is shown. The SIRT-1 graph is the expression level graph, which the legend indicates should be in 3B. The SIRT-1 graph shown needs to be moved to 3B and replaced with the correct graph.

- Figure 5A: in the text the authors indicate that there is not a significant "modification" in CD3+CD4+ and CD3+CD8+ subpopulations, but not significant compared to what? For the CD3+CD25+ it appears from the graph that the comparison is d88, 102 and 116, but why are these the important time points? Shouldn't the authors be comparing the changes after engraftment (d88-d116) to before engraftment? If nu mice have no T cells, wouldn't the presence of any CD3+CD4+ and/or CD3+CD8+ cells indicate significant modification? Of course in Figure 5C the authors then show nu mice as having non-zero % of CD3, CD4 and CD8 T cells. This is all the more reason to have baseline comparisons or nuPBS comparisons for 5A (and 5B).

- Figure 5B: as mentioned above, comparing these numbers to a PBS control or baseline would be important. Also interesting though would be a direct comparison of TnuC57 and TnuMDX mice. Are they significantly different at the three time points assayed? Is it the kinetics that are different or the final engraftment or something else?

- Fig 5A-C, since the authors are discussing mice without T cells, having % events is only partially illuminating. How many T cells are actually being produced by these mice? Knowing the raw event #s and/or % of total cells that are CD3+ would be helpful.

- Fig 5C: A comparison to WT C57 mice would be helpful.

- Figure 5C: if up to 2.5% of CD4+ T-cells are seen infiltrating muscle in nuPBS mice, doesn't that mean that the background is 2.5% since those mice are supposed to have no T cells? With that in mind, is the level seen with TnuMDX really positive?

- Figure 6: Knowing how these things compare to regular WT C57 and mdx mice would be beneficial as a point of comparison to the Tnu mice.

Reviewer #2 (Remarks to the Author):

In this manuscript, authors address whether impaired muscle function in mdx mice is ascribed to their disorganized thymic structure. They investigated the structure, gene expressions, and protein expressions in the thymus of these mice. Most importantly, they performed transplantation of mdx thymus into athymic nude mice to ask if the thymic dysfunction is sufficient for inducing degenerative changes in the skeletal muscle of mdx mice. From these data, authors claim that the dysfunction of thymic stroma exacerbates muscular dystrophy in mdx mice. The idea is interesting. However, because there are some concerns, this reviewer thinks that authors' conclusion might not be fully supported by the data.

Two major concerns:

1. Author transplanted whole thymus from mdx (and control) mice into nude mice. In this method, all cells in the thymus (stroma cells, immature T cells and other cells) would be transferred into recipient nude mice. If mdx mice show autoimmunity and/or other immune diseases, autoreactive T cells (or other pathogenic lymphocytes) could circulate and enter into the thymus. These pathogenic cells in the thymus might be also transferred into the recipient and could provoke symptoms.

To overcome this problem, this reviewer strongly recommends authors to perform transplantation of fetal thymic stroma from mdx embryo into kidney capsule of nude mice, which is a typical method to address this type of question.

2. Thymic phenotypes are unclear. Author should perform FACS analysis of thymic stroma cells. Moreover, authors used whole thymus to evaluate gene expression levels by qPCR, which makes data interpretation complicated. This reviewer recommends qPCR analysis of "sorted" TECs to

evaluate expression level of Aire and other genes.

Minor comments:

Fig. 1

- Data quality of Fig. 1B is low.
- Keratin-8 is predominantly expressed in cortical TECs, however, Fig. 1D does not appear to be such typical staining pattern. Can authors do double staining with CK5 and CK8?
- Fig. 1E data do not support the conclusion that thymic medulla of mdx is smaller than that of C57BL/6 because the difference is not significant.
- I can not understand what authors concluded in Fig. 1F data.
- Intensity of GHR staining is quite low in control for Fig. 1H. Please improve it.

Fig. 2.

- Please show FACS profile for Fig. 2A, B, C, and D
- Please exhibit cell number of each fraction in Fig. 2A, B, C, and D.
- NF- κ B family consists of several members. Did author use NF- κ B1 in Fig. 2E or others? This reviewer recommends WB of RelB, which is expressed in mTECs. Moreover, although STAT3 is reduced in the mdx thymus, p-STAT3 is not. This is puzzling.

Fig. 3.

- Please do immunostaining of the thymic section with Aire.

Fig. 4.

- What is the blue staining in Fig. 4A?

Fig. 5

- Please do immunostaining or FACS analysis of Foxp3+Tregs in the muscle of the transplanted mice
- Please analyze the structure of transplanted thymi in the recipient.

Fig. 6.

- The reduction in Dys-2 intensity in Fig 6A is unclear. Moreover, only two samples each were evaluated by WB in Fig. 6B. As the difference is not statistically significant, the conclusion is not warranted. Please address them.
- Please try to detect autoantibody against muscle antigens in serum of the transplanted mice.

Reviewers' comments:

Reviewer #1 (Remarks to the Author):

Minor points:

- **There are a number of grammatical errors that make the paper difficult to read. The authors would be well served to have a collaborator or writing center help edit for grammar.**

Grammatical errors have been corrected as requested.

- **In the introduction the authors state: “Mendell et al demonstrated that the majority of DMD patients presented CD4+ T lymphocytes directed against self dystrophin epitopes, whose number was inversely correlated with steroid treatment.” However, only 2/6 (not a majority of) patients had self-primed immune responses. 1 of those patients was not on steroids and 1 was, but another patient not on steroids did not have a pre-existing response to dystrophin, so I’m not really sure the numbers are there to make this assertion. The Flanigan paper did show that patients on steroids were less likely to have self-directed anti-dystrophin T cell responses, although responses were still seen in less than a majority of patients: Flanigan KM1, Campbell K, Viollet L, Wang W, Gomez AM, Walker CM, Mendell JR. Anti-dystrophin T cell responses in Duchenne muscular dystrophy: prevalence and a glucocorticoid treatment effect. Hum Gene Ther. 2013 Sep;24(9):797-806. doi: 10.1089/hum.2013.092.**

We have reworked the introduction as suggested.

- **Figure 1F: does yellow indicate co-staining? Assuming so, mention that in the figure legend.**

Yellow of previous Figure 1F was the result of overlapping Z stack of the confocal pictures. We performed new confocal microscopy imaging and inserted them in the new Figure 1.

- **Fig 2A: In the text description of this figure the authors state that the C57 and mdx thymic populations are significantly different from one another in the DP and SP populations, but there is no such indication on the graph, and the figure legend states otherwise (and the dot plot doesn’t appear to show a significant difference).**

Text and Figure 2 legend have been corrected as requested.

- **The authors change the style of their graphs not only throughout the paper or a figure, but even within a graph itself (ex Fig 2F). Sometimes the individual animals are shown and sometimes what appears to be the average of each individual experiment is graphed, and presumably those numbers are what the statistics were run on. However, running a student T test on experimental averages is a poor method since it greatly alters the calculated deviations. The authors should make all of their graphs dot plots, show all the animals (not the experimental averages), and run student T tests on all of their animals rather than treating experimental averages as individual data points. Otherwise it appears that data is being obscured and p values manipulated.**

All graphs and writing style have been edited as suggested. Student T test and One-way ANOVA were made on all the animals represented as dot plots in the graphs.

- **“According to these results the expression of Aire-dependent genes Ins-2 and Spt-1 was significantly impaired in mdx thymus, whereas Mup4 and S100a8 showed similar trend even**

not statistically significant”—this is not a true statement. In Fig 3D the average expression values of s100a8 is higher in mdx mice. Additionally, the authors then state that these trended toward significance while RANK and CD40 differences did not, yet the mup4 differences don't seem strikingly different than the differences seen with RANK and CD40. I do not think it is appropriate to draw conclusions with RANK, CD40, mup4, and s100a8, other than just to say they are not significantly different.

In order to clarify this point and better sustain our conclusion, we performed qRT-PCR analysis of mTEC and cTEC cells isolated by in situ laser-capture microdissection (LCM). Differences between Aire and Aire-Dependent gene expression are now included in the text and new Figure 3.

• **“We found marked expression of dystrophin in C57Bl thymus...while dys-1 expression was dramatically decreased and dys-2 expression was absent...”** The authors are measuring dys-1 and dys-2 detection, not expression, as these are antibodies. These antibodies may have different sensitivities, so the authors cannot say for sure that expression is absent.

We agree with this referee that antibodies have different sensitivities and thus we are measuring the detection of the dystrophin protein. As suggested we corrected these sentences in the text.

• **Figure 4B, C, D: The authors mention in the paper text (but not the figure legend) that for 4C the PCR products were purified. Presumably this was not done for 4B. What about for 4D? If not, why not? The legend should clearly indicate whether the PCR products were purified.**

We corrected the legend of Figure 4 reporting that all PCR products have been purified as specified in the text.

Major points:

• **The figure legend for Fig 3 indicates that 3A has the densitometric analyses for AIRE and SIRT-1, but only the pixel quantification for AIRE is shown. The SIRT-1 graph is the expression level graph, which the legend indicates should be in 3B. The SIRT-1 graph shown needs to be moved to 3B and replaced with the correct graph.**

We performed new experiments by laser-capture microdissection (LCM) and data are now shown in the new Figure 3.

• **Figure 5A: in the text the authors indicate that there is not a significant “modification” in CD3+CD4+ and CD3+CD8+ subpopulations, but not significant compared to what? For the CD3+CD25+ it appears from the graph that the comparison is d88, 102 and 116, but why are these the important time points? Shouldn't the authors be comparing the changes after engraftment (d88-d116) to before engraftment? If nu mice have no Tcells, wouldn't the presence of any CD3+CD4+ and/or CD3+CD8+ cells indicate significant modification? Of course in Figure 5C the authors then show nu mice as having non-zero % of CD3, CD4 and CD8 T cells. This is all the more reason to have baseline comparisons or nuPBS comparisons for 5A (and 5B).**

Previous Figure 5 has been reworked to show all the data through days after transplantation and compare differences between animals transplanted. We also performed new experiments of fetal thymus transplantation to be consistent with our conclusions. Progression of the percentages of blood CD3+, CD4+, CD8+ T cell and CD3+CD25+ Treg cell were analyzed after transplantation at different time points (from 20-33 to 116 days). We inserted in all progression graphs a dashed line showing the percentage found in control nude animals as baseline: nude PBS-treated (for the whole thymus transplantation) and nude untreated (for the E17 thymus transplantation). Data are now presented as mean \pm SEM for each time point.

• **Figure 5B:** as mentioned above, comparing these numbers to a PBS control or baseline would be important. Also interesting though would be a direct comparison of TnuC57 and TnuMDX mice. Are they significantly different at the three time points assayed? Is it the kinetics that are different or the final engraftment or something else?

Both kinetics of Tnu^{C57} and Tnu^{MDX} are now shown in each graph of the new Figures 5 and 9.

• **Fig 5A-C,** since the authors are discussing mice without T cells, having % events is only partially illuminating. How many T cells are actually being produced by these mice? Knowing the raw event #s and/or % of total cells that are CD3+ would be helpful.

As suggested, we now included in the new Figures 5 and 9 the percentages of total CD3+ cells of transplanted animals and a dashed line showing the percentage found in untreated nude mice as baseline.

• **Fig 5C:** A comparison to WT C57 mice would be helpful.

• **Figure 5C:** if up to 2.5% of CD4+ T-cells are seen infiltrating muscle in nuPBS mice, doesn't that mean that the background is 2.5% since those mice are supposed to have no T cells? With that in mind, is the level seen with TnuMDX really positive?

The percentages of infiltrating T-cells found in the muscle of nude untreated mice were less than 1% in all the experiments performed (see new Figures 5 and 9) and similar to what described for the C57Bl muscle in our previous works ¹.

• **Figure 6:** Knowing how these things compare to regular WT C57 and mdx mice would be beneficial as a point of comparison to the Tnu mice.

To sustain our quantifications we included in the new Figure 6 new data of WT C57Bl and mdx mice as controls for dystrophin, MyHC-slow, ATP2a, and Murf1.

Reviewer #2 (Remarks to the Author):

In this manuscript, authors address whether impaired muscle function in mdx mice is ascribed to their disorganized thymic structure. They investigated the structure, gene expressions, and protein expressions in the thymus of these mice. Most importantly, they performed transplantation of mdx thymus into athymic nude mice to ask if the thymic dysfunction is sufficient for inducing degenerative changes in the skeletal muscle of mdx mice. From these data, authors claim that the dysfunction of thymic stroma exacerbates muscular dystrophy in mdx mice. The idea is interesting. However, because there are some concerns, this reviewer thinks that authors' conclusion might not be fully supported by the data.

Two major concerns:

1. Author transplanted whole thymus from mdx (and control) mice into nude mice. In this method, all cells in the thymus (stroma cells, immature T cells and other cells) would be transferred into recipient nude mice. If mdx mice show autoimmunity and/or other immune diseases, autoreactive T cells (or other pathogenic lymphocytes) could circulate and enter into the thymus. These pathogenic cells in the

thymus might be also transferred into the recipient and could provoke symptoms. To overcome this problem, this reviewer strongly recommends authors to perform transplantation of fetal thymic stroma from mdx embryo into kidney capsule of nude mice, which is a typical method to address this type of question.

We agree with this referee and really appreciate this comment, which gave us the chance to better support our conclusions. As suggested, we performed new experiments transplanting nude mice with fetal E17 thymus of mdx or C57Bl mice. Moreover, to verify the role of pathogenic lymphocytes, which could circulate and enter into the adult thymus inducing dystrophic symptoms, a control group of nude mice have been treated intravenously with peripheral blood CD4+ or CD8+ T lymphocytes isolated from mdx mice. From nude mice injected with fetal E17 thymus, populations of CD3+CD4+, CD3+CD8+ T-cells can be clearly detected in circulation and gradually increased over time. This was much more evident than in nude mice which received the adult whole thymus. Instead, the percentage of T lymphocytes remains low in animals injected only with mdx T lymphocytes. Interestingly the percentage of circulating Treg decreased over time in nude mice transplanted with E17 or adult thymus of mdx whereas in animals receiving C57Bl E17 thymus was the opposite. Importantly, nude mice treated with E17 thymus recapitulated the muscular dystrophic features of animals treated with adult whole thymus with significant impact on the muscle force. Overall, these results suggested that the transplantation of fetal E17 thymus of mdx amplify the degenerative changes observed in animals treated with whole mdx thymus, and supported the conclusion that the structure disorganization found in dystrophic thymi of mdx activates a diverse stromal repertoire. The whole thymus is organized into two morphologically and functionally distinct compartments: the cortex and the medulla, which house two distinct populations of thymic epithelial cells, cortical TECs (cTECs) and the medullary TECs (mTECs). Other thymic stromal cells (TSCs) include thymic fibroblasts, endothelial cells, as well as antigen presenting cells (APCs) like macrophages and dendritic cells (DCs). Overall, this network of thymic cells provides both homing signals for the immigration of lymphocyte progenitors originated from the bone marrow (BM) and trophic factors necessary for the differentiation and maturation of thymocytes. Otherwise, the fetal thymus is particularly enriched of mTECs and DCs, which provide a specialized microenvironment dedicated to the establishment of T-cell tolerance. mTECs contribute to the intrathymic development of CD4+CD25+ T reg cells²⁻⁴, which control autoimmunity by dampening the immune responses of self-reactive T cells in peripheral tissues. Genetic mutations that affect the development and function of mTECs can compromise T cell tolerance. For example, deletion or mutation of RANK^{5,6} leads to defective differentiation of mTECs phenotypically, resulting in a reduced or absent medulla in mice⁷. Disruption of the three-dimensional medullary architecture in these mice correlates with development of autoimmunity, manifested as lymphocytic infiltration in multiple tissues and autoantibody production. In addition, the absence of tissue-restricted antigen (TRA) expression and presentation by mTECs in mice lacking the transcriptional regulator Aire results in similar autoimmune manifestations⁸. Other study⁹ found that Tregs are elevated in human DMD/BMD and mouse mdx muscle and display an activated phenotype. Regulatory T cells (Tregs) are candidate immunosuppressive lymphocytes that have the functional capacity to modulate dystrophinopathy by regulating the balance between type 1 and type 2 inflammatory responses. However, in our experimental settings, the number of circulating Tregs decreased over time after thymus transplantation (both fetal and adult) and no Foxp3 positive cells have been found in muscles of transplanted nude mice. Although mTECs express a diverse repertoire of TRAs that largely contribute to the induction of T-cell tolerance, they cannot encompass the spectrum of all peripheral self-antigens. Migratory DCs have been shown to reinforce the deletion of autoreactive thymocytes by sampling peripheral self-antigens that would otherwise be undetectable to developing thymocytes. Studies based on Rag2^{-/-} OTII TCR-transgenic mice have shown that migratory cDCs induce the negative selection of autoreactive CD4+ thymocytes¹⁰. Interestingly, in co-culture assays, Sirpα+ cDCs efficiently convert CD4+CD25- thymocytes into CD4+CD25+Foxp3+ Tregs¹¹. Migratory cDCs were also found to efficiently induce nTreg cells in vivo. Since tolerance could be classified into central tolerance or peripheral tolerance depending on where the state is originally induced—in the thymus and bone marrow (central) or in muscle (peripheral), it is possible that the evidences of increased Treg activities found in DMD/BMD and mdx mice correlates with the migratory cDCs ability to transport antigens captured in the dystrophic muscles. This process may contribute to the

establishment of tolerance by deleting autoreactive CD4⁺ thymocytes and inducing nTreg cells. The combinatorial role between central and peripheral tolerance should also explain the variability of muscle inflammation shown between patients and the different response to the treatment with corticosteroids. Our investigation represents the first evidence of impaired central tolerance determined by the disorganization of dystrophic thymus. This assumption is also confirmed by the absence of autoantibodies against dystrophin in transplanted nude mice (see below). In fact, circulating DCs of transplanted nude mice expressing dystrophin should induce the tolerance for dystrophin protein.

2. Thymic phenotypes are unclear. Author should perform FACS analysis of thymic stroma cells. Moreover, authors used whole thymus to evaluate gene expression levels by qPCR, which makes data interpretation complicated. This reviewer recommends qPCR analysis of “sorted” TECs to evaluate expression level of Aire and other genes.

In the morphological and IF analysis of thymus of mdx we noted a complexity enhanced by the presence in mdx of cortical tissue in the medullary and vice-versa, which in agreement with this referee may influence our qPCR quantification. Fluorescence-activated cell sorting (FACS) is usually capable of separating multiple cell populations simultaneously, which should resolve the contamination from others non stromal cells. However, FACS of mTEC or cTEC has some limitations: i) the high number of animals needed to obtain sufficient amount of cells for all experiments, ii) sorting efficiency may be reduced, causing significant cell loss, iii) antibody binding may activate intracellular signal transduction that induces functional changes of the sorted cell population which influences the qPCR analysis. Among the other methods existing for the cell enrichment besides FACS we opted for using laser-capture microdissection (LCM) which offers a robust and reliable tool to isolate specific cell types from their harboring tissues. The tissues can be then subjected to molecular analyses that allow elucidation of biological features with associated molecular mechanisms. LCM is well suited for downstream genomic analyses of minute amounts of material, allowing spatial detection of transcripts of specific cell population out of heterogeneous biological samples such as thymus tissue. Differences between Aire and Aire-Dependent gene expression were evaluated by LCM and results have been included in the text and new Figure 3.

Minor comments:

Fig. 1

- Data quality of Fig. 1B is low.

In agreement with this referee we performed extensive characterization of thymus of C57 and mdx mice and included new representative confocal pictures in the new Figure 1.

- Keratin-8 is predominantly expressed in cortical TECs, however, Fig. 1D does not appear to be such typical staining pattern. Can authors do double staining with CK5 and CK8?

We thank this reviewer for this comment, which gave us the chance to better understand the abnormalities of thymic stroma of dystrophic mdx mice. We included CK5/CK8 images in the new Figure 1.

- Fig. 1E data do not support the conclusion that thymic medulla of mdx is smaller than that of C57BL/6 because the difference is not significant.

In agreement with this referee we performed more immune-fluorescence experiments to increase the sample size and confirmed that the CK8⁺ area (cortex) is significantly reduced in mdx vs C57 mice whereas no statistical differences of CK5⁺ area (medulla) have been found between mdx and C57 mice. This is now reported in the new Figure 1 and in text.

-I can not understand what authors concluded in Fig. 1F data.

The previous Figure 1F (now 1D) showed CD3⁺ developing thymocytes and CK5⁺ mTEC. Developing thymocytes interact with the thymus stromal cells from the cortex to the medulla of thymus. Interaction between self-antigen loaded thymic stromal cells in the medulla and newly generated T cells expressing self-reactive TCR specificities leads to the induction of central T cell tolerance via clonal deletion of self-reactive T cells (negative selection). It is interesting to note that CD3⁺ developing thymocytes of mdx are prevalently restricted to the cortical area suggesting delay in their maturation. This data is harmonized by the others evidences provided in the new set of experiments.

-Intensity of GHR staining is quite low in control for Fig. 1H. Please improve it.

We improve all the images of Figure 1 by confocal analysis.

Fig. 2.

-Please show FACS profile for Fig. 2A, B, C, and D

-Please exhibit cell number of each fraction in Fig. 2A, B, C, and D.

As suggested, FACS profile and cell number are now provided in the new Figure 2.

-NF- κ B family consists of several members. Did author use NF- κ B1 in Fig. 2E or othres? This reviewer recommends WB of RelB, which is expressed in mTECs. Moreover, although STAT3 is reduced in the mdx thymus, p-STAT3 is not. This is puzzling.

Activation of canonical and non-canonical NF- κ B signaling pathways results in activation of respectively NF- κ B1/p50/p65 (NF- κ B1/RelA) and p52/RelB complexes. We used NF- κ B1 to evaluate the canonical pathway. As suggested we also tried to evaluate the non-canonical pathway for RelB expression. In a new set of experiments we first tested the rabbit anti-mouse RelB (sc-226) obtained from Santa Cruz but we found lots of unspecific bands. We then decided to move forward using the EMD Millipore RelB antibody (Milipore, 06-1105) as previously described by Finkin et al ¹². However, not conclusive results have been found. Representative gel is shown below for this referee.

Our results may depend on the fact that RelB is an unstable protein *in vivo* and requires highly specific partners for its stabilization ¹³. Expression of STAT3 in normal thymic tissue, which is negative for pSTAT3, was previously described ¹⁴. Moreover, STAT3/p-STAT3 expression is prevalently detected in cortical compared to medullary TEC. These findings indicate that the phosphorylated (activated) p-STAT3, is differentially involved in TEC differentiation. Our STAT3/p-STAT3 results may be related on the reduced cortical area of thymus of mdx. Moreover p-STAT3 expression of thymus of mdx is also reduced even not significantly.

Fig. 3.

-Please do immunostaining of the thymic section with Aire.

The IF staining for Aire have been now included in the new Figure 3.

Fig. 4.

-What is the blue staining in Fig. 4A?

Blue corresponded to the DAPI. New staining for dystrophin detection has been performed and included in the new Figure 4.

Fig. 5

-Please do immunostaining or FACS analysis of Foxp3+Tregs in the muscle of the transplanted mice

As suggested we first performed immunostaining of Foxp3 in the muscle of transplanted animals but we failed to detect any positive cells. This was also related by the low amount of infiltrating cells found in muscle sections of transplanted nude mice. Moreover, Foxp3+ Tregs were never obtained by FACS analysis from 3 independent experiments. These data are in agreement with previous observations made by Burzyn et al¹⁵ that isolated few amount of Tregs from the muscle of mdx (which are more infiltrated than nude mice).

-Please analyze the structure of transplanted thymi in the recipient.

For the experiments of transplantation of adult thymus we injected tissue fragments subcutaneously into the scruff of nude mice. This procedure leads to a dispersion of the injected material under the skin not allowing its characterization after 116 days. However, the transplantation of the fetal thymus underneath the kidney capsule gave us the chance to characterize the transplanted donor thymi (see Supplementary Figure 6).

Fig. 6.

- The reduction in Dys-2 intensity in Fig 6A is unclear. Moreover, only two samples each were evaluated by WB in Fig. 6B. As the difference is not statistically significant, the conclusion is not warranted. Please address them.

As suggested, the IF staining for Dys-1 and Dys-2 has been repeated and results are now commented in the new Figure 4. We also increased the number of samples analyzed by WB in the new Figure 6.

-Please try to detect autoantibody against muscle antigens in serum of the transplanted mice.

As requested we used serum of treated nude for western blot detection of anti-dystrophin antibodies. Diluted 1:100 serum samples from Tnu^{MDX} and Tnu^{E17MDX} transplanted mice were applied to nitrocellulose membrane strips generated by electrotransfer from gels through which C57BL/10 mouse muscle proteins were separated. Anti-dystrophin antibodies captured by dystrophin on the membrane strip were detected using an anti-mouse secondary antibody. A monoclonal DYS-2 anti-dystrophin antibody served as a positive control (CTR). Serum from PBS-injected nude mice (sham) served as a negative control. The investigator performing this assay was blind to the identity of the treatment groups. Sera from treated animals have not antibody responses against dystrophin. Representative gel is shown below for this referee.

Timing of the formation of autoantibodies against the muscle may be critical, and our detection after thymus transplantation may be warranted. Moreover, development of autoantibodies may depend to the combinatorial role between central and peripheral tolerance. Typically, the mdx muscle is characterized by remodeling/increased inflammation, which might trigger the reactivity to dystrophin in regions that would only be expressed by revertant fibers. However nude muscle is undamaged. Whether an immune response against dystrophin can be enhanced by the transplantation of thymus from mdx in nude mice, cannot be determined at this time.

References

- 1 Banfi, S. *et al.* Supplementation with a selective amino acid formula ameliorates muscular dystrophy in mdx mice. *Scientific reports* **8**, 14659, doi:10.1038/s41598-018-32613-w (2018).
- 2 Apostolou, I., Sarukhan, A., Klein, L. & von Boehmer, H. Origin of regulatory T cells with known specificity for antigen. *Nat Immunol* **3**, 756-763, doi:10.1038/ni816 (2002).
- 3 Aschenbrenner, K. *et al.* Selection of Foxp3⁺ regulatory T cells specific for self antigen expressed and presented by Aire⁺ medullary thymic epithelial cells. *Nat Immunol* **8**, 351-358, doi:10.1038/ni1444 (2007).
- 4 Jordan, M. S. *et al.* Thymic selection of CD4⁺CD25⁺ regulatory T cells induced by an agonist self-peptide. *Nat Immunol* **2**, 301-306, doi:10.1038/86302 (2001).
- 5 Akiyama, T. *et al.* The tumor necrosis factor family receptors RANK and CD40 cooperatively establish the thymic medullary microenvironment and self-tolerance. *Immunity* **29**, 423-437, doi:10.1016/j.immuni.2008.06.015 (2008).
- 6 Hikosaka, Y. *et al.* The cytokine RANKL produced by positively selected thymocytes fosters medullary thymic epithelial cells that express autoimmune regulator. *Immunity* **29**, 438-450, doi:10.1016/j.immuni.2008.06.018 (2008).
- 7 Derbinski, J. *et al.* Promiscuous gene expression in thymic epithelial cells is regulated at multiple levels. *The Journal of experimental medicine* **202**, 33-45, doi:10.1084/jem.20050471 (2005).
- 8 Anderson, M. S. *et al.* Projection of an immunological self shadow within the thymus by the aire protein. *Science* **298**, 1395-1401, doi:10.1126/science.1075958 (2002).
- 9 Villalta, S. A. *et al.* Regulatory T cells suppress muscle inflammation and injury in muscular dystrophy. *Science translational medicine* **6**, 258ra142, doi:10.1126/scitranslmed.3009925 (2014).
- 10 Proietto, A. I. *et al.* Dendritic cells in the thymus contribute to T-regulatory cell induction. *Proceedings of the National Academy of Sciences of the United States of America* **105**, 19869-19874, doi:10.1073/pnas.0810268105 (2008).
- 11 Guerri, L. *et al.* Analysis of APC types involved in CD4 tolerance and regulatory T cell generation using reaggregated thymic organ cultures. *Journal of immunology* **190**, 2102-2110, doi:10.4049/jimmunol.1202883 (2013).

- 12 Finkin, S. *et al.* Ectopic lymphoid structures function as microniches for tumor progenitor cells in hepatocellular carcinoma. *Nat Immunol* **16**, 1235-1244, doi:10.1038/ni.3290 (2015).
- 13 Fusco, A. J. *et al.* Stabilization of RelB requires multidomain interactions with p100/p52. *The Journal of biological chemistry* **283**, 12324-12332, doi:10.1074/jbc.M707898200 (2008).
- 14 Chang, K. C., Wu, M. H., Jones, D., Chen, F. F. & Tseng, Y. L. Activation of STAT3 in thymic epithelial tumours correlates with tumour type and clinical behaviour. *The Journal of pathology* **210**, 224-233, doi:10.1002/path.2041 (2006).
- 15 Burzyn, D. *et al.* A special population of regulatory T cells potentiates muscle repair. *Cell* **155**, 1282-1295, doi:10.1016/j.cell.2013.10.054 (2013).

REVIEWER COMMENTS

Reviewer #1 (Remarks to the Author):

The authors have accumulated an extensive and impressive collection of data that will greatly add to the field's understanding of how the immune system is directly involved in DMD disease pathology. This area is generally not well understood, and publishing this paper will help move the field forward.

Minor comments:

- 193: "TRA" should be "TRAg"
- 203: "Sheat" was maybe meant to be "sheet"?
- All references to Figure 1G in the text should be changed to 1E.
- "Histogram" refers to a particular kind of bar graph. Replace "histogram" with "graph" in all places.
- Is the Y axis correctly labeled for Figure 2E? With the way it is now it appears that you are saying Tregs are 1-5% of FoxP3+CD25+ cells, but I think you meant that 1-5% of CD4+CD8- cells are FoxP3+CD25+ Tregs. If that's the case your Y axis should read "% of CD4 SP cells" or something similar.
- 257: There is a missing ").".
- 257: I'm not sure I see the difference between C57 and mdx in terms of AIRE+ cells being "confined to focal areas of cortico-medullary demarcation." You yourselves wrote that there is "a comparable AIRE+ cell pattern distribution embedded within CK5+ thymic medulla of both mice" in the figure legend. I would recommend removing the part about them being confined to the cortico-medullary demarcation from the body text.
- 378: "collaged" should be "collagen"
- 512: "recept" should be "receptor"

Reviewer #2 (Remarks to the Author):

I really thank the authors for their thorough reply and for performing additional experiment in response to my requests.

What remains as one of major concern is that author used LCM instead of FACS-sorted cells to determine some gene expression levels of mTECs and cTECs. Normal LCM is used for gene expression in the particular "region" in sliced tissues (or other monolayer cells). Since frequency of TECs in the thymus is quite low, it is unlikely that normal LCM isolates selectively TECs. Especially, selective isolation of cTECs from cortex region might be very difficult due to their low frequency (~0.1 % cTEC in above 70% of DP in total thymic cells). Authors could utilize a recent LCM technology, which may make it possible to isolate cells at single cell level. However, in any cases, authors should clearly describe the method (how many single cells are collected and so on) and show the regions (or cells) cut by LCM in Figure. Furthermore, in the case using single cell LCM, since mTECs and cTEC are heterogenous as recently described in the studies using scRNA-seq (PMID 31804611), isolated single cells should be characterized by specific gene markers.

Moreover, although I agree that immunostaining study gives some information on TEC development and localization, FACS analysis would be necessary for a full characterization of the thymus disorganization as it has been used in many previous studies on TECs.

Reviewer #3 (Remarks to the Author):

The manuscript by Farini et al. reports some effects on muscle structure, function and gene expression that result from transplanting thymus derived tissue and cells into wild-type and mdx

mice. The question they address is very interesting: do thymus-derived immune cells affect the pathology of mdx dystrophy?" Although the question is interesting and the quantity of data is large, the investigation is weakened by limitations of experimental design, poor quality of some key reagents, insufficient statistical power to provide sufficient support for many conclusions, questionable use of sampling and statistics, overstatement of many findings and the discordance between the reported findings and numerous other previously-validated findings concerning the pathology of mdx muscular dystrophy.

Major concerns.

1. The major finding of the investigation, and the title, is that defective dystrophic thymus determines degenerative changes in skeletal muscle. In fact, the authors conclude that thymus tissue transplanted into healthy mice will cause muscular dystrophy. This would be an important discovery, but it is hard to make sense of the conclusion in light of hundreds of other investigations that have generated findings that have been reproduced by multiple labs over the past 30 years. If the conclusions in the current study are correct, why does the mdx pathology resolve after a few weeks and then the mice are essentially normal for many months after? If cells derived from the mdx thymus cause atrophy of healthy muscle fibers, then why is mdx muscle mass greater than wild-type muscles? If mdx thymus cells determine mdx pathology, why does transduction of mdx muscle cells with numerous therapeutic genes (utrophin, alpha-integrin, nNOS) greatly reduce pathology? Why would fusion of transplanted myogenic cells with mdx muscle fibers in vivo protect those fibers from pathology, if the presence of mdx thymus derived cells in the mice causes pathology? Why does lymphocyte-deficiency (nude mouse background) not prevent or diminish mdx pathology, other than reducing muscle fibrosis, if thymus-derived cells in mdx mice determine the pathology? Why is tetanic tension produced in mdx muscle not less than wild-type muscles at 3-months of age, if the presence of thymus cells from mdx mice causes a reduction in tetanic tension in muscles?

1. The authors show in Fig 1A a disruption of thymus architecture in 3 mon mdx which corresponds with the findings of Savino in 1995 (ref 14). In Fig 1B, they show that CK14/16 is reduced in mdx thymus (used as marker of mTECs) which they say is necessary for "structural integrity" of the cells. This conclusion is limited because, as shown in the raw data file, the antibodies to CK14/16 and vinculin are highly-non-specific and it would not be possible to make a conclusive comparison on relative protein levels by densitometry of the blots they illustrate. This is even further limited by the small sample size (n = 3). In addition, Savino saw increased CK8/18 in mdx thymus which they related to the same morphological defects that are reported in the current study. How do the authors relate the two sets of observations? The investigators also show in Fig. 1C immunohistochemistry that indicates no change in CK5, but reduction in CK8. However, in Figure 1D there appears to be show more CK5 in mdx thymus and Fig 1E appears to show more CK8 in mdx. Why do the data sets not appear more congruent?

2. Fig 1E and Suppl Fig 1 are purported to show that GHS-R expression is "critical for the correct stromal microenvironment of dystrophic thymus" (lines 224-5) and that there is a "dramatic loss of GHS-R" (line 252). The evidence offered to validate the claim is one high-magnification field of anti-GHS-R labeled thymus of one control and one mdx mouse that shows less labeling in mdx. The conclusion is insufficiently supported because there are no quantitative data and the sample size is one microscopic field. Suppl fig 1 then shows down-regulation of PECCK and Pgar mRNA and imaging mass spec data of one control and one mdx thymus to show altered distribution of glycerophospholipids. These are insufficiently sampled, unquantified data that provide no evidence that GHS-R is playing a "critical" role in the thymus of dystrophic mice.

3. Figure 2 is intended to show "Abnormal T cell development and autophagy impairment of dystrophic thymus" (line 228). However, the authors state in the results (lines 236, 237) that "these results were not conclusive for a dysregulated thymocyte development in thymus of mdx mice." In the legend for the figure, they state that "thymus homogenate from mdx and C57Bl demonstrates no significant alteration of T-cells " (lines 977-8) but the data show that the proportion of double positive T-cells that were TCRb+ and CD69+ was about two time greater in mdx and, more importantly, that the proportion of CD25 cells in mdx that were FoxP3+ also nearly doubled. The data and statements are inconsistent and the data are not sufficient to support the

conclusion.

The authors also claim that Figure 2 shows "a significant down-regulation of NF- κ B1 ($P=0.0459$) and STAT3 ($p=0.0479$) in dystrophic thymus of mdx related to C57Bl mice (Figure 2F). These data are supportive of the altered architecture of the dystrophic thymus, since impairment of NF- κ B and STAT3 associated signalling has been described in impaired cellularity and development of mTECs and thymus dimension " (lines 240-243). However, the quantification of the data in the westerns is questionable (the densitometry data don't reflect the images of the blots) and the entire blots are not shown in the raw data files so it is not possible to assess the quality of the antibody. In addition, they investigators did not assay NF κ B activity and their data show that there is not an increase in pSTAT3, so there is no support for the conclusion that either of these pathways is experiencing a change in activation.

4. Figure 3 is purported to show "AIRE signaling pathway dysregulation in mTEC of mdx thymus." The rationale for assaying the AIRE pathway is their claim that their preceding data sets showed that there was "dramatic loss of GHS-R, and defects in NF κ B signaling pathways and autophagy machinery " (line 252) although that assertion was not supported by strong data (NF κ B protein, not activation, was assayed). Given the shockingly high levels of expression of AIRE and SIRT1 that they show in whole tissue extracts in their blots, the authors should provide evidence to confirm the specificity of the antibodies, e.g. preabsorption of the antibodies with the antigen before repeating the blots and show the preabsorption specifically eliminates the band they identify as the antigens they are assaying.

The western blots for dystrophin proteins show the antibody is highly non-specific which would make any attempts to quantify antigens by densitometry of blots unreliable.

The evidence for impaired autophagy consists of showing that the ratio of LC3-II/LC3-I is lower in mdx than in WT. However, the evidence is very weak, based on one western with $n=3$ and a technical defect in lane 2 of the WT (Fig. 2H) would make accurate quantification impossible. In addition, the quantified data in the scatter gram don't resemble the western. In each mdx sample the LC3-II band appears more dense than the LC3-I band, but the plotted data indicate that the ratio of the band densities for each sample is ~ 1 . In addition, they report that there is no difference in the expression of p62 between mdx and WT in Fig 2G, and as they state in line 245, "expression of p62 - master regulator of the autophagy." Thus, the data seem to support the conclusion that there is not an impairment in autophagy in mdx thymus.

Figure 5 is intended to show that "Adult dystrophic thymus transplantation into nude mice altered T cell development. " However, whether transplantation altered development was not tested. The investigators showed some differences in the relative proportion of different T-cell populations in nude mice that received minced thymus from wild-type or mdx mice, but those differences may have reflected differences in the cells that occurred prior to their transplantation, not changes in development caused by transplantation. Furthermore, interpretation of the data would be facilitated if the authors reported how many cells were transplanted. If it varied between treatment groups that would affect the outcomes in subsequent experiments. Also, please report numbers of cells at each time point in figure 5b in addition to percentages, so changes in total numbers over time can be assessed. The western blot results of whole muscle extracts in figure 5c are very surprising because Tregs are exceeding few in muscle, even in fully-inflamed mdx muscle, which has made many other labs unable to detect FoxP3 in whole muscle extracts, even under pathological conditions when Treg numbers were elevated. The surprise is also increased by the finding of a strong signal for FoxP3 in the control nude mice, which do not have T-cells. Those observations support the possibility that the band that is labeled "FoxP3" is not actually FoxP3. It will be especially important to validate this antibody and blot. It would also be more convincing to other investigators in the field if there were also immunohistochemistry data with double-labeling for FoxP3 with CD4 or CD25 to validate these proposed differences in FoxP3 in these muscles. It is also puzzling that the authors show in figure 5f that some nude mice that did not receive transplanted cells had more CD3+ and CD4+ T-cells than mice that received transplantation. How would this be possible when nude mice have no T-cells, as confirmed in Figure 5b in this investigation?

The raw data files for Figure 5b does not contain the data from the nu/PBS mice.

The raw data files for Figure 5b show great variability between the proportion of CD3+ expressing different T-cell lineage markers between successive reads on a single mouse and between two mice in the same treatment group at a single sampling time. For example, in one mouse the percentage of CD3+ cells that expressed CD4 was 9.6 then 0 then 60.6 then 10.1 then 53.1 then 45.7 and then 53.1. This huge variability would be unlikely to be the result of biology and may reflect a technical issue.

In Figure 5C, only a portion of the FoxP3 blot is shown in the raw data file and no densitometry data are provided. In addition, there are only 2 samples per condition in the blot, although there were four mice in each treatment group. Which 2 mice were chosen for the blot would have a great influence on the outcome because the raw data files for Figure 5b show that at the last sample point for CD25+/CD3+ cells (which would express FoxP3), 0% of the cells were CD25+ in two mice but in the other two mice at that time point, 18.4 or 24.8% of the cells were FoxP3+.

Figure 6. Figures 6a and b purportedly show that there is no difference in Dp427 in nu/PBS mice and wild-type mice, but transplantation of wild-type thymus into nu mice reduces Dp427. The raw data file for figure 6b shows that the dystrophin antibody is so non-specific that it would not be possible to do a meaningful quantification of dystrophin concentration by densitometry of the blot. In addition, the densitometry data are not provided in the raw files.

They authors conclude that transplantation of mdx thymus changes myosin isoform expression but this surprising observation is not place in the context of mdx pathology or current knowledge about regulation of myosin isoform expression. Do published data show that mdx mice experience the same shift in MHC isoform expression at the onset of pathology? Do published data show that the MHC isoform switch in mdx reversed when the pathology subsides? Do published data show that immunosuppression in mdx mice causes changes in MHC expression?

Data in Figure 6D is presented as a histogram with bars, instead of a scatter diagram. Please clarify what the data sets are in the raw data file for Figure 6D. It appears that 3 mice in each group were analyzed (T1, T2, T3)? Which mice were selected for analysis and what were the selection criteria?

The raw data for Figure 6E suggest that the following treatment groups had the following number of mice:

N+pbs 16 mice
N+C57 16
N+mdx 18
C57 6
Mdx 6

But when reporting animal weight in the raw data files for figure 6F, the data were collected from the following number of mice:

N+pbs 2 mice
N+C57 4
N+mdx 4

Why were data from so few mice used for weight measurements? Which mice were selected for reporting weight data and what were the selection criteria?

The authors report in figure 6F that the mass of n+mdx mice was less that n+pbs and n+C57, at $p < 0.01$ using one way ANOVA. First, it is not possible to do meaningful statistical analysis with one way ANOVA with $n = 2$ for a treatment group. The authors do not provide individual data points in figure, just histogram bars.

The raw data for Figure 6G suggest that the following treatment groups had the following number of mice:

N+pbs 7 mice
N+C57 10

N+mdx 9
C57 10
Mdx 11

Given the differences in the numbers of mice assayed for MHC and ATP2a expression by QPCR in Figure 6E and for Murf1 by QPCR in Figure 6G and what were the selection criteria? Presumably, the same RNA/cDNA samples could have been used for QPCR of all transcripts.

Figure 6H shows a western blot for Akt quantity in N+pbs, N+C57 and N+mdx mice purporting to show a significant reduction in AKT in N+mdx. However, only 2 mice in each group were analyzed and it is not possible to do meaningful statistical analysis with one way ANOVA with $n = 2$ for a treatment group. Also, which mice were selected for assaying Akt and what were the selection criteria?

Data in Figures 7B, 7C 7D and 7E should be presented as a scatter diagram.

The authors don't state how they identified necrotic or regenerative fibers. They do not state how they sampled to determine % fibrosis. They express data for necrosis as percentage per section. However, the raw data files show that they analyzed the following numbers of sections:

N+pbs 25 sections
N+C57 28
N+mdx 28

This means that they assayed multiple sections from individual mice and that they used different numbers of sections for different mice. This is inappropriate sampling that would bias the data toward animals from which data were collected from more sections. It also appears that they treated each section as a sample, which would be inappropriate sampling and lead to data bias. A sample should be all the data from a single mouse, not from each section from an individual mouse. How many mice were used in each group?

The raw data files for Figure 7B indicate the same sampling flaws for assaying regenerative fibers, which could also lead to biasing the results.

The raw data files for Figures 7D and 7E suggest that all fibers area measurements for a single treatment group were combined in a single data set and each fiber was treated as an independent sample. This would be inappropriate sampling and data analysis that could bias the results and lead to invalid conclusions if more fibers were sampled from one muscle than from another muscle. In addition, treating each fiber as an independent sample (they aren't) artificially creates huge statistical power with an $n =$ several thousand (fibers), instead of the actual $n = 3$ or 4 (mice). This would be reflected in the extremely low p-values in figures 7B, C, D and E (e.g. $p < 0.0001$ which is not possible with only a few mice per group), despite the gigantic errors shown in the graphs of the data.

The raw data files for Figure 7F show $n = 3$ for each of the n+pbs, n+C57 and n+mdx groups. Is each "n" a single tetanic contraction from a single mouse? How were the mice selected, given the concerns expressed regarding Figure 5C above?

Figure 8 is intended to show that "TnuMDX dystrophic like phenotype is not induced by systemic transplantation of mdx-derived T-lymphocytes." (lines 419-420). However, the investigators do not show that the transplanted CD4+ and CD8+ cells actually survived the transplantation and were present in the transplant recipients, so it is not possible to conclude whether the entirely negative data set shown in the figure was the result of an unsuccessful transplantation.

Figure 9. Many of the concerns expressed for Figure 7 pertain to Figure 9. For example, Figure 10B illustrates a potential consequence of inappropriately treating each fiber in a section as an independent sample (where $n = 7000$ to 9000) instead of treating each mouse as an independent sample (where n is not specified, but base on other data sets it may be $n = 3$). Figure 10B shows little difference between mean values of the groups, with huge standard deviations, but the authors conclude the groups differ at $p = 0.0001$. The appropriate analysis would be to sample the

same number of fibers in each muscle, calculate the mean value for that animal, and treat that value as one sample. Similarly, in the raw data file for figure 10C for assaying fibrosis, that each group consisted of an "n" of about 100. It is unlikely that 100 mice were analyzed in each group, in which a sample was not appropriately identified and whether a sufficient number of animals were sampled is not identified and whether sampling was uniform between different animals was not shown.

In Figures 10i through k, QPCR data for several transcripts are shown for the mice receiving transplantations of fetal thymus or controls. Based on data shown in the raw data files for Figure 9, 4 mice received mdx thymus and 3 mice received C57 thymus. However, in the raw data files for Figures 10i through k, some data sets for an individual group analyzed for QPCR (e.g., n+C57) had 3 samples and other QPCR data sets for the same group had 9 samples. Were there actually nine mice in the n+C57 group? If so, which 3 were selected for analysis for some QPCR reactions and not in other reactions? What were the exclusion criteria? If there weren't 9 mice in the n+C57 group, what were the 9 samples in the Murf1 reaction?

Suppl Figure 2 purportedly shows an increase in fibrosis with a western (n= 2) of dubious quality showing an increase in collagen 1. ["statistically significant over-expression of collagen I"; line 379]. However, that increase in the proportion of the whole muscle that was comprised of collagen could just be secondary to a decrease in fiber size since controlled for mass loaded on gel for blot. Thus, there are no data to show that collagen over-expression occurred.

Suppl Fig 3 purportedly reports data concerning "Inflammatory marker expression in muscles of TnuMDX mice." by showing westerns of whole muscle extracts with extremely high levels of expression of TGFb, TNF, NFkB and IL10 even in control animals. This is highly improbable and validation of the antibody specificity needs to be provided. In addition, they analyze 2 samples per treatment and then claim to show statistical significance using one-way Anova. No meaningful analysis by one-way ANOVA can be made with a sample size of 2. The raw data files only show a portion of the western blots and no densitometry data are provided.

Suppl Fig 4 is purported to show that "Autophagy is impaired in the skeletal muscles of TnuMDX mice" and is founded upon the assertion that "The up-regulation of pro-inflammatory cytokines driven by lymphocytes is likely responsible for the severe reduction of myofiber area and muscle strength observed in TnuMDX mice" (lines 395-396). However, the only data to support that foundation is provided in Supplemental figure 3, which shows only an increase in TNF in a western blot with a sample of n = 2 per treatment using a TNF antibody that showed improbably high levels of TNF in whole extracts of muscles from TNF. If the antibody is specific and lymphocytes are its source, why are there extraordinarily high levels of TNF protein in the muscles of nude mice? The data in Suppl fig 4 then builds on that assertion by assaying for changes in the ratio of LC3III/LC3II, concluding that the ratio is lower in TnuMDX mice, indicating impaired autophagy. However, only 2 samples were analyzed in each treatment group and the strength of any statistical assay would be very limited, the raw data files show only a slice of the western and no raw data for the densitometry were provided. Thus, the basis for the conclusion is not strong.

Reviewer #1

The authors have accumulated an extensive and impressive collection of data that will greatly add to the field's understanding of how the immune system is directly involved in DMD disease pathology. This area is generally not well understood, and publishing this paper will help move the field forward.

Minor comments:

· 193: *“TRA” should be “TRAg”*

The typo was corrected in the new version of the Manuscript

· 203: *“Sheat” was maybe meant to be “sheet”?*

The typo was corrected in the new version of the Manuscript: we meant “sheath” concerning the fibroblastic reticular cells (FRCs)

· *All references to Figure 1G in the text should be changed to 1E.*

As suggested, the typo was corrected in the new version of the Manuscript

· *“Histogram” refers to a particular kind of bar graph. Replace “histogram” with “graph” in all places.*

As suggested, we revised “histogram” in the new version of the Manuscript

· *Is the Y axis correctly labeled for Figure 2E? With the way it is now it appears that you are saying Tregs are 1-5% of FoxP3+CD25+ cells, but I think you meant that 1-5% of CD4+CD8- cells are FoxP3+CD25+ Tregs. If that's the case your Y axis should read “% of CD4 SP cells” or something similar.*

As suggested by this reviewer, we corrected the axis as “% of CD4+ cells” in the new version of Figure 2

· 257: *There is a missing “).”.*

As suggested, we corrected this typo in the new version of the Manuscript

· 257: *I'm not sure I see the difference between C57 and mdx in terms of AIRE+ cells being “confined to focal areas of cortico-medullary demarcation.” You yourselves wrote that there is “a comparable AIRE+ cell pattern distribution embedded within CK5+ thymic medulla of both mice” in the figure legend. I would recommend removing the part about them being confined to the cortico-medullary demarcation from the body text.*

As suggested by this reviewer, we modified the sentence in the body of the text

· 378: *“collaged” should be “collagen”*

The typo was corrected in the new version of the Manuscript

· 512: *“recept” should be “receptor”*

The typo was corrected in the new version of the Manuscript

Reviewer #2

I really thank the authors for their thorough reply and for performing additional experiment in response to my requests. What remains as one of major concern is that author used LCM instead of FACS-sorted cells to determine some gene expression levels of mTECs and cTECs. Normal LCM is used for gene expression in the particular “region” in sliced tissues (or other monolayer cells). Since frequency of TECs in the thymus is quite low, it is unlikely that normal LCM isolates selectively TECs. Especially, selective isolation of cTECs from cortex region might be very difficult due to their low frequency (~0.1 % cTEC in above 70% of DP in total thymic cells). Authors could utilize a

recent LCM technology, which may make it possible to isolate cells at single cell level. However, in any cases, authors should clearly describe the method (how many single cells are collected and so on) and show the regions (or cells) cut by LCM in Figure. Furthermore, in the case using single cell LCM, since mTECs and cTEC are heterogenous as recently described in the studies using scRNA-seq (PMID 31804611), isolated single cells should be characterized by specific gene markers.

Moreover, although I agree that immunostaining study gives some information on TEC development and localization, FACS analysis would be necessary for a full characterization of the thymus disorganization as it has been used in many previous studies on TECs.

As suggested by this reviewer, we performed new FACS analysis and RT-qPCR analysis on mTEC and cTEC isolated by FACS cell sorting from the thymus of age-matched 3-months old mdx and C57Bl mice (n=8 animals/group). As described in details in Material and Methods Section, cells isolated from dissociated thymi were depleted from thymocytes in order to enrich the subpopulations of interest. In a second step, as performed in ([doi:10.3791/51780](https://doi.org/10.3791/51780)), these cells were incubated with an antibody cocktail containing anti-CD45 (Pacific Blue, clone 30-F11), anti-EpCAM (PE-Cy7), anti-MHC-class II (PerCP/Cy5.5), anti-Ly51 (FITC) – all from Biolegend –, and the UEA-1 fluorescein (from Vector Laboratories, Burlingame, CA, USA) and analyzed by FACS. Precisely, we obtained $1,13 \times 10^6$ C57Bl cTEC; $1,24 \times 10^6$ mdx cTEC; $7,14 \times 10^5$ C57Bl mTEC and $7,27 \times 10^5$ mdx mTEC. cTEC and mTEC were then pulled randomly in two groups/animal (n=4 animals/group of C57Bl and mdx mice) with similar number of cells in order to have enough amount of sample to perform RNA extraction and RT-qPCR experiments and we investigated the expression of different genes as reported in the new version of the Figure 3C-E.

Reviewer #3

Major concerns.

1. The major finding of the investigation, and the title, is that defective dystrophic thymus determines degenerative changes in skeletal muscle. In fact, the authors conclude that thymus tissue transplanted into healthy mice will cause muscular dystrophy. This would be an important discovery, but it is hard to make sense of the conclusion in light of hundreds of other investigations that have generated findings that have been reproduced by multiple labs over the past 30 years. If the conclusions in the current study are correct, why does the mdx pathology resolve after a few weeks and then the mice are essentially normal for many months after? If cells derived from the mdx thymus cause atrophy of healthy muscle fibers, then why is mdx muscle mass greater than wild-type muscles? If mdx thymus cells determine mdx pathology, why does transduction of mdx muscle cells with numerous therapeutic genes (utrophin, alpha-integrin, nNOS) greatly reduce pathology? Why would fusion of transplanted myogenic cells with mdx muscle fibers in vivo protect those fibers from pathology, if the presence of mdx thymus derived cells in the mice causes pathology? Why does lymphocyte-deficiency (nude mouse background) not prevent or diminish mdx pathology, other than reducing muscle fibrosis, if thymus-derived cells in mdx mice determine the pathology? Why is tetanic tension produced in mdx muscle not less than wild-type muscles at 3-months of age, if the presence of thymus cells from mdx mice causes a reduction in tetanic tension in muscles?

More recently, a crosstalk between immune and muscle cells has been proposed suggesting a role of immune system in muscle diseases making the environment inimical to myogenesis. For instance, human DMD patients and mdx mice, although sharing many pathological features, differ in the rate of the disease progression and the severity of muscle deterioration and loss of muscle function. It is interesting to speculate that such variations in clinical presentation may be due to differences in the phenotypic character, functional state, or altered kinetics of immune cell recruitment in human versus murine muscle tissues. Among immune cells, regulatory T cells (Tregs) are candidate immunosuppressive lymphocytes that have the functional capacity to modulate dystrophinopathy by regulating the balance between type 1 and type 2 inflammatory responses. Tregs are elevated in human DMD/BMD and mouse mdx muscles and display an activated phenotype. mTECs contribute to the intra-thymic development of Tregs. Although mTECs express a diverse repertoire of TRAs that largely contribute to the induction of T- cell tolerance, they cannot encompass the spectrum of all peripheral self-antigens. Migratory conventional dendritic cells (cDCs) have been shown to reinforce the deletion of autoreactive thymocytes by sampling peripheral self-antigens that would otherwise be undetectable to developing thymocytes. Since tolerance could be classified into central tolerance or peripheral tolerance depending on where the state is originally induced—in the thymus and bone marrow (central) or in muscle (peripheral), it is possible that the evidences of increased Treg activities found in DMD/BMD and mdx mice correlates with the migratory cDCs ability to transport antigens captured in the dystrophic muscles. This process may contribute to the establishment of tolerance by deleting autoreactive CD4+ thymocytes and inducing Treg cells. The

combinatorial role between central and peripheral tolerance should also explain the variability of muscle inflammation shown between patients and the different response to the treatment with corticosteroids. Our investigation represents the first evidence of impaired central tolerance in mdx determined by the disorganization of dystrophic thymus. The contribution of this thymus disorganization in early stage muscular dystrophy should explain differences of muscle damage and inflammation observed in different models of mdx such as mdx3cv, mdx4cv, and mdx-nude mice. For instance, the mdx-nude mouse showed lower fibrogenesis than mdx and transplantation of normal thymic tissue into the mdx-nude mice replenished their T-cells and concomitantly altered the collagen content in their tissues to levels comparable with those in immunocompetent mdx mice. However, it is unknown how the muscle function of the mdx-nude differs from mdx; in addition, if differences exist, their physio-pathological meaning should be clarified. Although several evidences suggest differences in tetanic tension between 3-months old mdx and healthy mice (10.1038/mt.2016.162; 10.1111/j.1469-7793.2000.t01-3-00457.x; 10.1038/mt.2009.322; 10.1016/j.ajpath.2011.07.009; 10.1152/japplphysiol.00776.2015; 10.1093/hmg/ddu469) instead of what is affirmed by this referee, the biomolecular mechanisms involved in the chimeric model of dystrophic thymus transplantation into nude mice cannot be compared to the ones involved in mdx or healthy itself. Moreover, the membrane fragility, consequent to dystrophin absence, does not fully explain the divergent expression signature of both dystrophic mice (mdx) and DMD patients at different stages of the disease versus healthy muscles in terms of muscle wasting, inflammation and fibrosis (10.1093/hmg/11.3.263; 10.1016/s0960-8966(02)00092-5; PMID: 10998753). The inflammation, macrophage infiltration, degenerating and regenerating fibers, and stretch-induced damage are commonly described in DMD as secondary pathology. An understanding of how dystrophin deficiency initiates changes in muscle functions is required for rational development of therapies to address these secondary consequences of dystrophin deficiency. The examples of therapeutic treatments raised by this referee have an impact on the secondary consequences of the dystrophin deficiency when defect in muscle regeneration, increased inflammation and fibrosis have already developed. Here, we pointed out the consequences of primary defects in dystrophic thymus for the development of some features of muscular dystrophy. In our experiments we demonstrated that DMD does not affect only muscle cells but thymus as well with consequences on the immune muscle homeostasis. By means of dystrophic thymus transplantation in nude mice it has been possible to investigate the biomolecular mechanisms involved in central immune tolerance and muscle function. The comprehension of the dynamics underlying mdx mice immune regulation could represent a precious step forward to better understand which central-immune controlled activities are affected by the lack of dystrophin and how. Moreover, from a translational perspective, an important contribution could be given for the development and/or the refinement of efficient therapeutic approaches.

1. The authors show in Fig 1A a disruption of thymus architecture in 3 mon mdx which corresponds with the findings of Savino in 1995 (ref 14). In Fig 1B, they show that CK14/16 is reduced in mdx thymus (used as marker of mTECs) which they say is necessary for “structural integrity” of the cells. This conclusion is limited because, as shown in the raw data file, the antibodies to CK14/16 and vinculin are highly-non-specific and it would not be possible to make a conclusive comparison on relative protein levels by densitometry of the blots they illustrate. This is even further limited by the small sample size (n = 3). In addition, Savino saw increased CK8/18 in mdx thymus which they related to the same morphological defects that are reported in the current study. How do the authors relate the two sets of observations? The investigators also show in Fig. 1C immunohistochemistry that indicates no change in CK5, but reduction in CK8. However, in Figure 1D there appears to be show more CK5 in mdx thymus and Fig 1E appears to show more CK8 in mdx. Why do the data sets not appear more congruent?

Savino work was a pioneering demonstration of the involvement of thymus mdx in 1995. In that manuscript, the authors found “*intense immunolabelling for cytokeratins pair 8/18*” and they did not performed quantification as we have done by counting CKs positive cells and by WB analysis. Moreover, they used a different set of antibodies to recognize immune histological differences. In agreement with recent literature in mice, we used antibodies specific to CK5, CK8, and CK14/16 that have been most commonly used in studies to identify the main TEC subtypes (10.1002/eji.200839191; 10.1073/pnas.95.20.11822; 10.4049/jimmunol.169.6.2842; 10.1371/journal.pgen.0020146). In addition, we performed FACS sorting experiments as requested by referee n2 and results confirmed reduction of both mTEC and cTEC of dystrophic mdx thymus (see new Figure 3C). Thus, we improved the earlier findings of Savino work using a combination of well characterized protocols from the recent literature of thymus field of research. Among selected antibodies for thymus characterization, the CK5, CK8, CK14/16 and vinculin antibodies are well documented and validated in different species including mouse as described in datasheets. Our WB data have been repeatedly validated and more data are now added in support of our conclusions (n=6 animals/group). The appearance discrepancy between immunohistochemistry and change in CK5 or 8 expression is congruent with the quantification of their relative staining obtained from 12 sections/mouse (n=6 animals/group) that have been selected randomly to exclude any data manipulation or bias in providing the information (see new Raw data and

Figure 1). To be congruent with CK5/CK8 immune quantification, representative confocal images of whole thymus sections are now added in the Figure 1.

2. Fig 1E and Suppl Fig 1 are purported to show that GHS-R expression is “critical for the correct stromal microenvironment of dystrophic thymus” (lines 224-5) and that there is a “dramatic loss of GHS-R” (line 252). The evidence offered to validate the claim is one high-magnification field of anti-GHS-R labeled thymus of one control and one mdx mouse that shows less labeling in mdx. The conclusion is insufficiently supported because there are no quantitative data and the sample size is one microscopic field. Suppl fig 1 then shows down-regulation of PECCK and Pgar mRNA and imaging mass spec data of one control and one mdx thymus to show altered distribution of glycerophospholipids. These are insufficiently sampled, unquantified data that provide no evidence that GHS-R is playing a “critical” role in the thymus of dystrophic mice.

In order to clarify the GHS-R differences and better sustain our conclusion, in the new version of the manuscript we included GHS-R fluorescence quantification of 12 thymic sections/C57Bl mouse and 9 sections/mdx mouse (n=6 animals/group). The quantitative RT-PCR analysis in Suppl Fig 1 provides evidences of the up-regulation of genes involved in thymic adipocytes (N=4-6 for C57Bl and mdx mice); however mass spec data provide a qualitative supportive image representation of the differences in the site distribution across the cortical and medullary regions of the thymus. Besides, we modified the sentence concerning the “critical role” of GHS-R in the development of thymus.

3. Figure 2 is intended to show “Abnormal T cell development and autophagy impairment of dystrophic thymus” (line 228). However, the authors state in the results (lines 236, 237) that “these results were not conclusive for a dysregulated thymocyte development in thymus of mdx mice.” In the legend for the figure, they state that “thymus homogenate from mdx and C57Bl demonstrates no significant alteration of T-cells “ (lines 977-8) but the data show that the proportion of double positive T-cells that were TCRb+ and CD69+ was about two time greater in mdx and, more importantly, that the proportion of CD25 cells in mdx that were FoxP3+ also nearly doubled. The data and statements are inconsistent and the data are not sufficient to support the conclusion.

This part of work was already reviewed by the two initial referee’s addressing specific comments. Flow cytometry data analysis was performed in n=7 mdx and n=5 C57Bl mice (see Raw data file) which is sufficient to observe 0.5 significant differences as power calculation. To determine sample sizes for FACs analysis, power analysis was carried out using preliminary data sets. Considering the independent sample t-test, a sample size of 5 achieved >95% power to reject null hypothesis at the significance level of 0.05. All the data meet assumptions of the statistical test and different groups had similar variance. Developing thymocytes interact with the thymus stromal cells from the cortex to the medulla. Interaction between self-antigens- loaded thymic stromal cells in the medulla and newly generated T cells expressing self-reactive TCRs leads to the induction of central T cell tolerance via clonal deletion of self-reactive T cells (negative selection). Thymus homogenates were not indicative of differences in the total number and side scatter of T lymphocytes shown in Figure 2A, however the thymocyte development refers to differences in DN3 and DN4 stages (Figure 2C), TCRb/CD69+ cells and Tregs (Figure 2D-E) of mdx. These data suggest an accelerated transition through the DN3 and DN4 stages leading to an early activation of central tolerance in the presence of disorganized thymic architecture of mdx mice. To better understand the complexity of thymus subpopulation development please see recent single cell publication (10.1038/s41577-019-0238-0).

In agreement with this referee we better described the FACS analysis of Figure 2A-E and improved our comments in the new version of the manuscript. To better support our conclusions we provide new WB analysis with n=6 animals/group (see new Figure 2F and H).

The authors also claim that Figure 2 shows “a significant down-regulation of NF-kB1 (P=0.0459) and STAT3 (p=0.0479) in dystrophic thymus of mdx related to C57Bl mice (Figure 2F). These data are supportive of the altered architecture of the dystrophic thymus, since impairment of NF-kB and STAT3 associated signalling has been described in impaired cellularity and development of mTECs and thymus dimension “ (lines 240-243). However, the quantification of the data in the westerns is questionable (the densitometry data don’t reflect the images of the blots) and the entire blots are not shown in the raw data files so it is not possible to assess the quality of the antibody. In addition, they investigators did not assay NFkB activity and their data show that there is not an increase in pSTAT3, so there is no support for the conclusion that either of these pathways is experiencing a change in activation.

This part was already commented for referee n1 as follows.

Activation of canonical and non-canonical NF- κ B signaling pathways results in activation of respectively NF- κ B1/p50/p65 (NF- κ B1/RelA) and p52/RelB complexes. We used NF- κ B1 to evaluate the canonical pathway. As suggested, we also tried to evaluate the non-canonical pathway for RelB expression. In a new set of experiments, we first tested the rabbit anti-mouse RelB (sc-226) obtained from Santa Cruz but we found lots of unspecific bands. We then decided to move forward using the EMD Millipore RelB antibody (Millipore, 06-1105) as previously described by Finkin et al.¹². However, not conclusive results have been found. Representative gel is shown below for this referee.

Our results may depend on the fact that RelB is an unstable protein in vivo and requires highly specific partners for its stabilization¹³. Expression of STAT3 in normal thymic tissue, which is negative for pSTAT3, was previously described¹⁴. Moreover, STAT3/p-STAT3 expression is prevalently detected in cortical compared to medullary TEC. These findings indicate that the phosphorylated (activated) p-STAT3, is differentially involved in TEC differentiation. Our STAT3/p-STAT3 results may be related on the reduced cortical area of thymus of mdx. Moreover p-STAT3 expression of thymus of mdx is also reduced even not significantly.

Since this part of experiments seems to be not conclusive, we increased the number of tested animals by WB providing a new set of data in the revised manuscript (n=6 animals/group) (new Figure 2F).

4. Figure 3 is purported to show "AIRE signaling pathway dysregulation in mTEC of mdx thymus." The rationale for assaying the AIRE pathway is their claim that their preceding data sets showed that there was "dramatic loss of GHS-R, and defects in NF κ B signaling pathways and autophagy machinery " (line 252) although that assertion was not supported by strong data (NF κ B protein, not activation, was assayed). Given the shockingly high levels of expression of AIRE and SIRT1 that they show in whole tissue extracts in their blots, the authors should provide evidence to confirm the specificity of the antibodies, e.g. preabsorption of the antibodies with the antigen before repeating the blots and show the preabsorption specifically eliminates the band they identify as the antigens they are assaying.

In order to obtain quantitative data from western blots, a rigorous methodology have been used for all antibodies tested as previously described (10.1007/s12033-013-9672-6). In detail, we calculated the dilution factor of samples that is required for protein loading in the quantitative linear dynamic range for each antibody. Furthermore, we selected the most appropriate normalization method based on reference signals obtained either by housekeeping proteins after immunochemical staining or total protein intensity on blotting membranes after total protein staining. Our WB protocols were standardized considering the sensitivity and specificity of the antibody as referred in literature (in this case for AIRE and SIRT1) to assure that the reported fold changes of the target protein are not an artifact of reference signal. However, we added more WB results (n=6 animals/group) to address the point raised by this referee.

The western blots for dystrophin proteins show the antibody is highly non-specific which would make any attempts to quantify antigens by densitometry of blots unreliable.

We used a Western blot protocol that allows sensitive and accurate dystrophin quantification in preclinical and clinical studies (10.3233/JND-150113; 10.3233/JND-180357). Moreover, the antibodies tested are the same used for dystrophin diagnostic purpose in patients followed at the Neuromuscular Center of the Policlinico Hospital of Milan (Italy). Please see as example of similar dystrophin pattern of distribution the last publication of the Jerry Mendell group in JAMA Neurology (doi:10.1001/jamaneurol.2020.1484) published online June 15, 2020.

The evidence for impaired autophagy consists of showing that the ratio of LC3-II/LC3-I is lower in mdx than in WT. However, the evidence is very weak, based on one western with n=3 and a technical defect in lane 2 of the WT (Fig. 2H) would make accurate quantification impossible. In addition, the quantified data in the scatter gram don't resemble the western. In each mdx sample the LC3-II band appears more dense than the LC3-I band, but the plotted data indicate that the ratio of the band densities for each sample is ~1. In addition, they report that there is no difference in the expression of p62 between mdx and WT in Fig 2G, and as they state in line 245, "expression of p62 -

master regulator of the autophagy." Thus, the data seem to support the conclusion that there is not an impairment in autophagy in mdx thymus.

LC3II/LC3I ratio was obtained as the ratio of LC3II/actin and LC3I/actin thus the denser band observed in LC3II mdx have to be normalized to his actin housekeeping marker. Several evidences described decrease in autophagy as diminished LC3II/I ratio and increase in p62 and decreased atg7. In these experiments we evaluated the p62 expression by qRT-PCR which probably mislead our conclusion. However, to better support these observations, we added more WB results (n=6 animals/group) for LC3, p62 and atg7 (new Figure 2H).

Figure 5 is intended to show that "Adult dystrophic thymus transplantation into nude mice altered T cell development. " However, whether transplantation altered development was not tested. The investigators showed some differences in the relative proportion of different T-cell populations in nude mice that received minced thymus from wild-type or mdx mice, but those differences may have reflected differences in the cells that occurred prior to their transplantation, not changes in development caused by transplantation. Furthermore, interpretation of the data would be facilitated if the authors reported how many cells were transplanted. If it varied between treatment groups that would affect the outcomes in subsequent experiments. Also, please report numbers of cells at each time point in figure 5b in addition to percentages, so changes in total numbers over time can be assessed. The western blot results of whole muscle extracts in figure 5c are very surprising because Tregs are exceeding few in muscle, even in fully-inflamed mdx muscle, which has made many other labs unable to detect FoxP3 in whole muscle extracts, even under pathological conditions when Treg numbers were elevated. The surprise is also increased by the finding of a strong signal for FoxP3 in the control nude mice, which do not have T-cells. Those observations support the possibility that the band that is labeled "FoxP3" is not actually FoxP3. It will be especially important to validate this antibody and blot. It would also be more convincing to other investigators in the field if there were also immunohistochemistry data with double-labeling for FoxP3 with CD4 or CD25 to validate these proposed differences in FoxP3 in these muscles. It is also puzzling that the authors show in figure 5f that some nude mice that did not receive transplanted cells had more CD3+ and CD4+ T-cells than mice that received transplantation. How would this be possible when nude mice have no T-cells, as confirmed in Figure 5b in this investigation?

T cell maturation was tested after thymus transplantation using a published method which is based on the transplantation of minced thymus (10.1038/jid.2012.492). The limitation of this approach is related to the fact that transplantation of thymic cells does not work because thymus cannot be reproduced without stromal tissue. Supplementary Figure 6 showed the presence of transplanted thymus under the kidney capsule. The weights of the transplanted minced thymi were similar in all experiments. Raw data of FACS analysis (Figure 5B) reporting the number of cells per time point/mouse were added in the additional file. As suggested by referee n2 we performed immunostaining of Foxp3 in the muscle of transplanted animals but we failed to detect any positive cells. This was also related by the low number of infiltrating cells found in muscle sections of transplanted nude mice. Moreover, Foxp3+ Tregs were never obtained by FACS analysis from 3 independent experiments. These data are in agreement with previous observations made by Burzyn et al that isolated few amounts of Tregs from the muscle of mdx (which are more infiltrated than nude mice) (10.1016/j.cell.2013.10.054). Specificity of FoxP3 band in WB was demonstrated by using preadsorbed antibody that led to band disappearance, meaning that antibody (FJK-16s eBioscience) is well suited for detection of Foxp3 by WB analysis. To be consistent with our conclusions we added more WB results (n=4 animals/group) for Foxp3 (new Figure 5C). The percentages of infiltrating T-cells found in the muscle of nude untreated mice were less than 1% in all the experiments performed (see also Figure 9) and similar to what described for the C57Bl muscle in our previous works (10.1038/s41598-018-32613-w). As suggested by referee n1, we included in the Figures 5 and 9 a dashed line showing the percentage of CD25, CD4, CD8 found in untreated blood of nude mice as baseline. As previously published by other groups the blood of nude mouse showed T cell expression between 0.1-2%.

The raw data files for Figure 5b does not contain the data from the nu/PBS mice.

The raw data of nu/PBS mice are in line with the values of the dashed line and have been included in the raw data files.

The raw data files for Figure 5b show great variability between the proportion of CD3+ expressing different T-cell lineage markers between successive reads on a single mouse and between two mice in the same treatment group at a single sampling time. For example, in one mouse the percentage of CD3+ cells that expressed CD4 was 9.6 then 0 then 60.6 then 10.1 then 53.1 then 45.7 and then 53.1. This huge variability would be unlikely to be the result of biology and may reflect a technical issue.

The amount of blood used and number of events analyzed for these experiments were reproduced similarly between different time points. Moreover, the up and down trend described by this referee is not related to one single animal but different groups had similar variance (see Raw data file) meaning that is a biological effect of the transplanted thymus that is taking time for growing and functioning.

In Figure 5C, only a portion of the FoxP3 blot is shown in the raw data file and no densitometry data are provided. In addition, there are only 2 samples per condition in the blot, although there were four mice in each treatment group. Which 2 mice were chosen for the blot would have a great influence on the outcome because the raw data files for Figure 5b show that at the last sample point for CD25+/CD3+ cells (which would express FoxP3), 0% of the cells were CD25+ in two mice but in the other two mice at that time point, 18.4 or 24.8% of the cells were FoxP3+.

As indicated in the legend of Figure 5, the data shown in B are from the blood and data from WB analysis in C are from the muscle meaning that the comparison for FoxP3 is unjustified. However, we included in the new version of the manuscript (and in the Raw data) the WB bands (n=4 animals/group).

Figure 6. Figures 6a and b purportedly show that there is no difference in Dp427 in nu/PBS mice and wild-type mice, but transplantation of wild-type thymus into nu mice reduces Dp427. The raw data file for figure 6b shows that the dystrophin antibody is so non-specific that it would not be possible to do a meaningful quantification of dystrophin concentration by densitometry of the blot. In addition, the densitometry data are not provided in the raw files. They authors conclude that transplantation of mdx thymus changes myosin isoform expression but this surprising observation is not place in the context of mdx pathology or current knowledge about regulation of myosin isoform expression. Do published data show that mdx mice experience the same shift in MHC isoform expression at the onset of pathology? Do published data show that the MHC isoform switch in mdx reversed when the pathology subsides? Do published data show that immunosuppression in mdx mice causes changes in MHC expression?

As mentioned above, we used a Western blot protocol that allows sensitive and accurate dystrophin quantification in preclinical and clinical studies (10.3233/JND-150113; 10.3233/JND-180357). Moreover, the antibodies tested are the same used for dystrophin diagnostic purpose in patients followed at the Neuromuscular Center of the Policlinico Hospital of Milan (Italy). Figure 6D showed a decreased number of fiber Type IIX in Tnu^{MDX} which correspond to what observed in mdx mouse where the number of fiber Type IIX is decreased compared to wild type (10.1038/s41598-018-32613-w). Several works confirmed that atp2a1 is down-regulated in dystrophic mice (10.1038/s41598-019-38609-4), affecting muscle fiber type diversity (10.1016/j.ydbio.2011.08.010; 10.4238/2014.February.28.13) and over-expression of ATP2A1 was suggested to mitigate the dystrophic pathology (10.1172/JCI43844). Similarly, MyHCsI gene is expressed in mdx mice according to the development of pathology (10.1152/physrev.00031.2010; 10.1093/ejo/cjq113). Moreover, MyHCsI expression is influenced by muscle fiber degeneration, necrosis, and regeneration (10.1007/s10974-006-9066-5).

Data in Figure 6D is presented as a histogram with bars, instead of a scatter diagram. Please clarify what the data sets are in the raw data file for Figure 6D. It appears that 3 mice in each group were analyzed (T1, T2, T3)? Which mice were selected for analysis and what were the selection criteria?

In the new version of the manuscript (and in the Raw data) we add new counts in the graph of Figure 6D. We also specified the number of animals per group in the Raw data section and how we identified the different MyHC isoforms. In the new Figure 6D, the graph is now presented as a scatter diagram.

The raw data for Figure 6E suggest that the following treatment groups had the following number of mice:

N+pbs 16 mice

N+C57 16

N+mdx 18

C57 6

Mdx 6

For adult thymus transplantation experiments, we performed two different set of experiments (n=4 per group for the first set of experiments, n=4 per N+C57 and N+pbs groups and n=5 per N+mdx group for the second set of experiments; total number of animals: 8 from N+pbs; 8 from N+C57; 9 from N+mdx). N=6 mice were tested for each control group of C57 and Mdx mice.

But when reporting animal weight in the raw data files for figure 6F, the data were collected from the following number of mice:

N+pbs 2 mice

N+C57 4

N+mdx 4

Why were data from so few mice used for weight measurements? Which mice were selected for reporting weight data and what were the selection criteria?

All missing weight data were included in the revised text and in raw data files.

Figure 6H shows a western blot for Akt quantity in N+pbs, N+C57 and N+mdx mice purporting to show a significant reduction in AKT in N+mdx. However, only 2 mice in each group were analyzed and it is not possible to do meaningful statistical analysis with one way ANOVA with $n = 2$ for a treatment group. Also, which mice were selected for assaying Akt and what were the selection criteria?

We provided more AKT data in the Figure 6H and in raw data files.

Data in Figures 7B, 7C 7D and 7E should be presented as a scatter diagram.

These data are now presented as scatter diagram as suggested.

The authors don't state how they identified necrotic or regenerative fibers.

To compare among thymus-injected nude mice we performed the immunohistochemistry with anti-mouse IgG antibody and we counted the IgG+ fibers/total number of QA fibers (mean number of fiber=4500)(10.1038/srep38371). In the graph of Figure 7B, we indicated our values as the % of necrotic fiber/section. The regenerative fibers are easily identified as those that are small and centrally-nucleated in H&E staining. Accordingly, we indicated our values in the graph of Figure 7C as the number of centronucleated myofibers/total fibers. All these data are now better represented in the Raw data file.

They do not state how they sampled to determine % fibrosis. They express data for necrosis as percentage per section. However, the raw data files show that they analyzed the following numbers of sections:

N+pbs 25 sections

N+C57 28

N+mdx 28

This means that they assayed multiple sections from individual mice and that they used different numbers of sections for different mice. This is inappropriate sampling that would bias the data toward animals from which data were collected from more sections. It also appears that they treated each section as a sample, which would be inappropriate sampling and lead to data bias. A sample should be all the data from a single mouse, not from each section from an individual mouse. How many mice were used in each group?

Regarding the problem of the sampling of data, rising from Figure 7 to 10, we included all pooled data in the graphs of TA and QA myofiber area and represented them as scatter diagram in the revised text of the manuscript and Figures. The graphs indicating the counting per single mice are shown below for this referee. For the analysis of necrotic and regenerating fibers and fibrosis, we characterized 3-4 sections per animal (n=8 N+pbs; n=8 N+C57; n=9 N+mdx). Sections with artefacts or marked levels in background were excluded from the analysis (please see the new Raw data file).

B

C

D

TA myofiber area

E

Figure 7

Figure 8

The raw data files for Figure 7B indicate the same sampling flaws for assaying regenerative fibers, which could also lead to biasing the results.

N+pbs 20 sections

N+C57 28

N+mdx 25

The raw data files for Figures 7D and 7E suggest that all fibers area measurements for a single treatment group were combined in a single data set and each fiber was treated as an independent sample. This would be inappropriate sampling and data analysis that could bias the results and lead to invalid conclusions if more fibers were sampled from one muscle than from another muscle. In addition, treating each fiber as an independent sample (they aren't) artificially creates huge statistical power with an n = several thousand (fibers), instead of the actual n = 3 or 4 (mice). This would be reflected in the extremely low p-values in figures 7B, C, D and E (e.g. $p < 0.0001$ which is not possible with only a few mice per group), despite the gigantic errors shown in the graphs of the data.

We performed the analysis shown in Figure 7D-E as previously described (10.1186/s13395-017-0141-y; 10.1186/s13395-016-0092-8; 10.1186/s13395-018-0186-6; 10.14814/phy2.12391; 10.1126/scitranslmed.aan5662). Data were plotted per group of treated animals (n=4 per group). As suggested by this referee we represented data as dots per animals.

The raw data files for Figure 7F show n = 3 for each of the n+pbs, n+C57 and n+mdx groups. Is each "n" a single tetanic contraction from a single mouse? How were the mice selected, given the concerns expressed regarding Figure 5C above?

3 out of 4 animals per group of the first set of experiments were randomly selected for tetanic muscle force analysis and one used for qualitative mass spectrometry tissue imaging (not included at this stage in this manuscript).

Figure 8 is intended to show that "TnuMDX dystrophic like phenotype is not induced by systemic transplantation of mdx-derived T-lymphocytes." (lines 419-420). However, the investigators do not show that the transplanted CD4+ and CD8+ cells actually survived the transplantation and were present in the transplant recipients, so it is not possible to conclude whether the entirely negative data set shown in the figure was the result of an unsuccessful transplantation.

Previous evidences demonstrated long-term survival of intravenous injected CD4+ and CD8+ in nude mice (PMID: 7790023). Persistence of injected CD4+ and CD8+ cells has been not tested by FACS analysis or using a reporter GFP gene. Since haplotype of mdx (H2-k^b) differs from the one of nude (H2-k^d) we performed haplotype-specific RT-PCR analysis for H2-k^b in the muscle of transplanted nude mice at the time of sacrifice. Representative gel is shown below for this referee.

Moreover, results in Figure 8 show a modification of the myofiber area distribution compared to the nude untreated animals suggesting that these cells survived but not induce relevant effects on muscles of transplanted animals.

Figure 9. Many of the concerns expressed for Figure 7 pertain to Figure 9.

Probably this referee is referring to Figure 7 and 8.

In the Figure 9, CD4/CD8/CD25/CD3 FACS analysis were performed in groups of 4 animals followed at different ages for 7 times points. For the group of untreated nude mice (Figure 9B) we added one value in mouse 1 (see Raw data) at the starting point of experiments. These data were used as untreated control and represented as averaged values with dashed black lines in Figure 5B and in Figure 9D-E.

For example, Figure 10B illustrates a potential consequence of inappropriately treating each fiber in a section as an independent sample (where $n = 7000$ to 9000) instead of treating each mouse as an independent sample (where n is not specified, but base on other data sets it may be $n = 3$). Figure 10B shows little difference between mean values of the groups, with huge standard deviations, but the authors conclude the groups differ at $p = 0.0001$. The appropriate analysis would be to sample the same number of fibers in each muscle, calculate the mean value for that animal, and treat that value as one sample. Similarly, in the raw data file for figure 10C for assaying fibrosis, that each group consisted of an “ n ” of about 100. It is unlikely that 100 mice were analyzed in each group, in which a sample was not appropriately identified and whether a sufficient number of animals were sampled is not identified and whether sampling was uniform between different animals was not shown.

As indicated in our previous response for Figure 7, we performed these analysis as previously described; (10.1186/s13395-016-0092-8; 10.1186/s13395-018-0186-6; 10.14814/phy2.12391; 10.1126/scitranslmed.aan5662). Data were plotted per group of animals treated ($n=3$ animals/group). As suggested by this referee we represented pooled data dots per group of treatment in the text of the revised manuscript and per animals only for this referee (see above).

Figure 10

In Figures 10i through k, QPCR data for several transcripts are shown for the mice receiving transplantations of fetal thymus or controls. Based on data shown in the raw data files for Figure 9, 4 mice received mdx thymus and 3 mice received C57 thymus. However, in the raw data files for Figures 10i through k, some data sets for an individual group analyzed for QPCR (e.g., n+C57) had 3 samples and other QPCR data sets for the same group had 9 samples. Were there actually nine mice in the n+C57 group? If so, which 3 were selected for analysis for some QPCR reactions and not in other reactions? What were the exclusion criteria? If there weren't 9 mice in the n+C57 group, what were the 9 samples in the Murf1 reaction?

For fetal dystrophic thymus transplantation experiments, we analyzed both TA muscles per mouse in two different set of experiments (n=3 animals/group per experiments; total number of samples: 6 from nude; 6 from N+E17C57; 6 from N+E17mdx). We observed some technical problems regarding the expression of RelB and IL-1Beta (the samples always performed in duplicate) that prevented us to set up the correct baseline and/or to identify the threshold settings. Moreover, Outliers, identified as being more than 1.5 times the interquartile range below the 1st quartile or above the 3rd quartile in each treatment group were omitted. Accordingly, we were not able to accurately quantify few samples so that we did not include them in the graph of the previous version of the manuscript (and consequently in the Raw Data). We now performed more experiments and included new data of RelB and IL-1Beta (new Figure 10i). For Murf1 and PDK4 we decided to increase the number of samples including both TAs of animals to reduce the intra/inter variability and SD.

Suppl Figure 2 purportedly shows an increase in fibrosis with a western (n= 2) of dubious quality showing an increase in collagen 1. ["statistically significant over-expression of collagen I "; line 379]. However, that increase in the proportion of the whole muscle that was comprised of collagen could just be secondary to a decrease in fiber size since controlled for mass loaded on gel for blot. Thus, there are no data to show that collagen over-expression occurred.

As suggested, we included in the Suppl Figure 2 new representative WB quantification of collagen I and OPN. Ponceau S staining of the proteins transferred to the membrane confirmed similar amounts of proteins between specimens (see below). Results in the text are expressed as the ratio of relative intensity of collagen I and OPN protein expression normalized to GAPDH as a loading control.

Suppl Fig 3 purportedly reports data concerning "Inflammatory marker expression in muscles of TnuMDX mice. " by showing westerns of whole muscle extracts with extremely high levels of expression of TGFb, TNF, NFkB and IL10 even in control animals. This is highly improbable and validation of the antibody specificity needs to be provided. In addition, they analyze 2 samples per treatment and then claim to show statistical significance using one-way Anova. No meaningful analysis by one-way ANOVA can be made with a sample size of 2. The raw data files only show a portion of the western blots and no densitometry data are provided.

To sustain our quantifications, we included in the new Suppl Figure 3 new data of wb analysis (n=4 animals/group). Precise reference for each antibody used in this Manuscript are now specified in the new version of the Manuscript.

Suppl Fig 4 is purported to show that “Autophagy is impaired in the skeletal muscles of TnuMDX mice” and is founded upon the assertion that “The up-regulation of pro-inflammatory cytokines driven by lymphocytes is likely responsible for the severe reduction of myofiber area and muscle strength observed in TnuMDX mice” (lines 395-396). However, the only data to support that foundation is provided in Supplemental figure 3, which shows only an increase in TNF in a western blot with a sample of n = 2 per treatment using a TNF antibody that showed improbably high levels of TNF in whole extracts of muscles from TNF. If the antibody is specific and lymphocytes are its source, why are there extraordinarily high levels of TNF protein in the muscles of nude mice? The data in Suppl fig 4 then builds on that assertion by assaying for changes in the ratio of LC3III/LC3II, concluding that the ratio is lower in TnuMDX mice, indicating impaired autophagy. However, only 2 samples were analyzed in each treatment group and the strength of any statistical assay would be very limited, the raw data files show only a slice of the western and no raw data for the densitometry were provided. Thus, the basis for the conclusion is not strong.

We repeated our analysis providing more reliable WB data in Suppl Figure 3 (n=4 animals/group). To sustain our quantifications of autophagic markers, we included in the new Suppl Figure 4 new data of wb analysis for LC3II/I, p62 and atg7 (n=4 animals/group).

REVIEWER COMMENTS

Reviewer #2 (Remarks to the Author):

The authors have addressed all my concerns. I have no additional comments.

EDITORIAL NOTE: Reviewer #2 was asked to comment on Reviewer #3's previous remarks, as Reviewer #3 was unable to re-review this round.

[Comment 3] The authors also claim that Figure 2 shows “a significant down-regulation of NF- κ B1 (P=0.0459) and STAT3 (p=0.0479) in dystrophic thymus of mdx related to C57Bl mice (Figure 2F). These data are supportive of the altered architecture of the dystrophic thymus, since impairment of NF- κ B and STAT3 associated signalling has been described in impaired cellularity and development of mTECs and thymus dimension “ (lines 240-243). However, the quantification of the data in the westerns is questionable (the densitometry data don't reflect the images of the blots) and the entire blots are not shown in the raw data files so it is not possible to assess the quality of the antibody. In addition, they investigators did not assay NF κ B activity and their data show that there is not an increase in pSTAT3, so there is no support for the conclusion that either of these pathways is experiencing a change in activation. This part was already commented for referee n1 as follows.

Activation of canonical and non-canonical NF- κ B signaling pathways results in activation of respectively NF- κ B1/p50/p65 (NF- κ B1/RelA) and p52/RelB complexes. We used NF- κ B1 to evaluate the canonical pathway. As suggested, we also tried to evaluate the non-canonical pathway for RelB expression. In a new set of experiments, we first tested the rabbit anti-mouse RelB (sc-226) obtained from Santa Cruz but we found lots of unspecific bands. We then decided to move forward using the EMD Millipore RelB antibody (Millipore, 06-1105) as previously described by Finkin et al 12. However, not conclusive results have been found. Representative gel is shown below for this referee.

Our results may depend on the fact that RelB is an unstable protein in vivo and requires highly specific partners for its stabilization 13. Expression of STAT3 in normal thymic tissue, which is negative for pSTAT3, was previously described 14. Moreover, STAT3/p-STAT3 expression is prevalently detected in cortical compared to medullary TEC. These findings indicate that the phosphorylated (activated) p-STAT3, is differentially involved in TEC differentiation. Our STAT3/p- STAT3 results may be related on the reduced cortical area of thymus of mdx. Moreover p-STAT3 expression of thymus of mdx is also reduced even not significantly.

Since this part of experiments seems to be not conclusive, we increased the number of tested animals by WB providing a new set of data in the revised manuscript (n=6 animals/group) (new Figure 2F).

This issue was not properly addressed. First, this reviewer concerns specificity of antibodies in WB analysis, however, authors did not show entire blots in the Source_Data file (Membranes might be cut?). Furthermore, size markers should be indicated. Second, activation of NF- κ Bs is normally monitored by their nuclear translocations (or I κ Ba degradation), but not simply their expression levels. Authors should address these points.

[Comment 4] Figure 3 is purported to show “AIRE signaling pathway dysregulation in mTEC of mdx thymus.” The rationale for assaying the AIRE pathway is their claim that their preceding data sets showed that there was “dramatic loss of GHS-R, and defects in NF κ B signaling pathways and autophagy machinery “ (line 252) although that assertion was not supported by strong data (NF κ B protein, not activation, was assayed). Given the shockingly high levels of expression of AIRE and SIRT1 that they show in whole tissue extracts in their blots, the authors should provide evidence to confirm the specificity of the antibodies, e.g. preabsorption of the antibodies with the antigen before repeating the blots and show the preabsorption specifically eliminates the band they identify as the antigens they are assaying.

In order to obtain quantitative data from western blots, a rigorous methodology have been used for all antibodies tested as previously described (10.1007/s12033-013-9672-6). In detail, we calculated the dilution factor of samples that is required for protein loading in the quantitative linear dynamic range for each antibody. Furthermore, we selected the most appropriate normalization method based on reference signals obtained either by housekeeping proteins after immunochemical staining or total protein intensity on blotting membranes after total protein staining. Our WB protocols were standardized considering the sensitivity and specificity of the antibody as referred in literature (in this case for AIRE and SIRT1) to assure that the reported fold changes of the target protein are not an artifact of reference signal. However, we added more

WB results (n=6 animals/group) to address the point raised by this referee.

Again, this reviewer express concerns about specificity of antibodies in WB analysis. Unfortunately, author did not address this issue. The best way would be to use the thymus lysate from Aire KO or Sirt1 KO mice, however, these mice may not be easily available for authors. Instead, can authors show some reliable references in which these antibodies were used for WB analysis to detect expression of Aire and Sirt1 in the whole thymus lysates? Otherwise, authors should check the specificity by some experiments (e.g. pre-absorption) to address this issue.

[Comment 4-2] The western blots for dystrophin proteins show the antibody is highly non-specific which would make any attempts to quantify antigens by densitometry of blots unreliable.

We used a Western blot protocol that allows sensitive and accurate dystrophin quantification in preclinical and clinical studies (10.3233/JND-150113; 10.3233/JND-180357). Moreover, the antibodies tested are the same used for dystrophin diagnostic purpose in patients followed at the Neuromuscular Center of the Policlinico Hospital of Milan (Italy). Please see as example of similar dystrophin pattern of distribution the last publication of the Jerry Mendell group in JAMA Neurology (doi:10.1001/jamaneurol.2020.1484) published online June 15, 2020.

As described by authors, there is no doubt on the specificity against human dystrophin proteins. However, how about against the mouse proteins?

[Comment 4-3] Figure 5 is intended to show that "Adult dystrophic thymus transplantation into nude mice altered T cell development." However, whether transplantation altered development was not tested. The investigators showed some differences in the relative proportion of different T-cell populations in nude mice that received minced thymus from wild-type or mdx mice, but those differences may have reflected differences in the cells that occurred prior to their transplantation, not changes in development caused by transplantation. Furthermore, interpretation of the data would be facilitated if the authors reported how many cells were transplanted. If it varied between treatment groups that would affect the outcomes in subsequent experiments. Also, please report numbers of cells at each time point in figure 5b in addition to percentages, so changes in total numbers over time can be assessed. The western blot results of whole muscle extracts in figure 5c are very surprising because Tregs are exceeding few in muscle, even in fully-inflamed mdx muscle, which has made many other labs unable to detect FoxP3 in whole muscle extracts, even under pathological conditions when Treg numbers were elevated. The surprise is also increased by the finding of a strong signal for FoxP3 in the control nude mice, which do not have T-cells. Those observations support the possibility that the band that is labeled "FoxP3" is not actually FoxP3. It will be especially important to validate this antibody and blot. It would also be more convincing to other investigators in the field if there were also immunohistochemistry data with double-labeling for FoxP3 with CD4 or CD25 to validate these proposed differences in FoxP3 in these muscles. It is also puzzling that the authors show in figure 5f that some nude mice that did not receive transplanted cells had more CD3+ and CD4+ T-cells than mice that received transplantation. How would this be possible when nude mice have no T-cells, as confirmed in Figure 5b in this investigation?

T cell maturation was tested after thymus transplantation using a published method which is based on the transplantation of minced thymus (10.1038/jid.2012.492). The limitation of this approach is related to the fact that transplantation of thymic cells does not work because thymus cannot be reproduced without stromal tissue. Supplementary Figure 6 showed the presence of transplanted thymus under the kidney capsule. The weights of the transplanted minced thymi were similar in all experiments. Raw data of FACS analysis (Figure 5B) reporting the number of cells per time point/mouse were added in the additional file. As suggested by referee n2 we performed immunostaining of Foxp3 in the muscle of transplanted animals but we failed to detect any positive cells. This was also related by the low number of infiltrating cells found in muscle sections of transplanted nude mice. Moreover, Foxp3+ Tregs were never obtained by FACS analysis from 3 independent experiments. These data are in agreement with previous observations made by Burzyn et al that isolated few amounts of Tregs from the muscle of mdx (which are more infiltrated than nude mice)

(10.1016/j.cell.2013.10.054). Specificity of FoxP3 band in WB was demonstrated by using preabsorbed antibody that led to band disappearance, meaning that antibody (FJK-16s eBioscience) is well suited for detection of Foxp3 by WB analysis. To be consistent with our conclusions we added more WB results (n=4 animals/group) for Foxp3 (new Figure 5C). The percentages of infiltrating T-cells found in the muscle of nude untreated mice were less than 1% in all the experiments performed (see also Figure 9) and similar to what described for the C57Bl muscle in our previous works (10.1038/s41598-018-32613-w). As suggested by referee n1, we included in the Figures 5 and 9 a dashed line showing the percentage of CD25, CD4, CD8 found in untreated blood of nude mice as baseline. As previously published by other groups the blood of nude mouse showed T cell expression between 0.1- 2%.

This issue was not be properly addressed. This concern would be raised because this reviewer suspected specificity of the Foxp3 antibody in WB analysis. So, this reviewer also wondered why Foxp3 protein could be detected in the muscle of nude mice (PBS control). Since nude mice have a very small number of T cells, Treg cellularity should be extremely low in nude mouse. This is indeed puzzling and need to be addressed. Pre-absorption experiments may be necessary.

Author describe that “Raw data of FACS analysis (Figure 5B) reporting the number of cells per time point/mouse were added in the additional file.”, but I could not find out the cell number data.

[Comment 4-3] Data in Figure 6D is presented as a histogram with bars, instead of a scatter diagram. Please clarify what the data sets are in the raw data file for Figure 6D. It appears that 3 mice in each group were analyzed (T1, T2, T3)? Which mice were selected for analysis and what were the selection criteria?

In the new version of the manuscript (and in the Raw data) we add new counts in the graph of Figure 6D. We also specified the number of animals per group in the Raw data section and how we identified the different MyHC isoforms. In the new Figure 6D, the graph is now presented as a scatter diagram.

I did not find the scatter graph of Fig. 6D.

[Comment 4-4] Suppl Figure 2 purportedly shows an increase in fibrosis with a western (n= 2) of dubious quality showing an increase in collagen 1. [“statistically significant over-expression of collagen I “; line 379]. However, that increase in the proportion of the whole muscle that was comprised of collagen could just be secondary to a decrease in fiber size since controlled for mass loaded on gel for blot. Thus, there are no data to show that collagen over-expression occurred.

As suggested, we included in the Suppl Figure 2 new representative WB quantification of collagen I and OPN. Ponceau S staining of the proteins transferred to the membrane confirmed similar amounts of proteins between specimens (see below). Results in the text are expressed as the ratio of relative intensity of collagen I and OPN protein expression normalized to GAPDH as a loading control.

Data showed no significant or clear evidence that Col1 and OPN could be increased in the muscle of the MDX-transferred mice. Moreover, at least to me, intensities of some major bands visualized by Ponceau S staining shows a similar pattern with WB bands of Coll1 than GAPDH. I recommend authors to modify this part.

Response to Reviewer:

[Comment 3]

The authors also claim that Figure 2 shows “a significant down-regulation of NF-kB1 (P=0.0459) and STAT3 (p=0.0479) in dystrophic thymus of mdx related to C57Bl mice (Figure 2F). These data are supportive of the altered architecture of the dystrophic thymus, since impairment of NF-kB and STAT3 associated signalling has been described in impaired cellularity and development of mTECs and thymus dimension “ (lines 240-243). However, the quantification of the data in the westerns is questionable (the densitometry data don't reflect the images of the blots) and the entire blots are not shown in the raw data files so it is not possible to assess the quality of the antibody. In addition, they investigators did not assay

NFkB activity and their data show that there is not an increase in pSTAT3, so there is no support for the conclusion that either of these pathways is experiencing a change in activation.

This part was already commented for referee n1 as follows.

Activation of canonical and non-canonical NF-kB signaling pathways results in activation of respectively NFkB1/

p50/p65 (NF-kB1/RelA) and p52/RelB complexes. We used NF-kB1 to evaluate the canonical pathway. As suggested, we also tried to evaluate the non-canonical pathway for RelB expression. In a new set of experiments, we first tested the rabbit anti-mouse RelB (sc-226) obtained from Santa Cruz but we found lots of unspecific bands. We then decided to move forward using the EMD Millipore RelB antibody (Millipore, 06-1105) as previously described by Finkin et al 12. However, not conclusive results have been found. Representative gel is shown below for this referee.

Our results may depend on the fact that RelB is an unstable protein in vivo and requires highly specific partners for its stabilization 13. Expression of STAT3 in normal thymic tissue, which is negative for pSTAT3, was previously described 14. Moreover, STAT3/p-STAT3 expression is prevalently detected in cortical compared to medullary TEC. These findings indicate that the phosphorylated (activated) p-STAT3, is differentially involved in TEC differentiation. Our STAT3/p- STAT3 results may be related on the reduced cortical area of thymus of mdx. Moreover p-STAT3 expression of thymus of mdx is also reduced even not significantly.

Since this part of experiments seems to be not conclusive, we increased the number of tested animals by WB providing a new set of data in the revised manuscript (n=6 animals/group) (new Figure 2F).

This issue was not properly addressed. First, this reviewer concerns specificity of antibodies in WB analysis, however, authors did not show entire blots in the Source_Data file (Membranes might be cut?). Furthermore, size markers should be indicated. Second, activation of NF-kBs is normally monitored by their nuclear translocations (or I kBα degradation), but not simply their expression levels. Authors should address these points.

The entire blots were not shown since membranes were cut to allow multiple detections (as mentioned by this referee); however, as suggested, we indicated in this figure (and throughout all the figures of the Manuscript) the MW size. Furthermore, we improved NF-kB data performing western blot of IKKi (now included in the new version of Figure 2). IKKi was selected in agreement with previous works demonstrating that IKKi is predominantly expressed in thymus (doi: 10.1093/intimm/11.8.1357) and its over-expression is associated with I kBα phosphorylation and NF-kB activation (10.1016/j.cell.2007.03.052; 10.1158/0008-5472.CAN-05-1602). In these new experiments we found that IKKi was significantly downregulated in thymus of mdx and the reduction of IKKi protein level was well correlated with NF-kB reduction (new Figure 2F). We would like to thank this reviewer in helping us to address this critical point.

Regarding the concerns about the specificity of the NF-kB, STAT3 and pSTAT3 antibodies, we summarized a list of references for the antibodies selected for performing WBs:

NF-kB p65 (A-12) (sc-514451, Santa Cruz Biotechnology) references:

- ✓ Mol Med Rep. doi: 10.3892/mmr.2017.6837.
- ✓ J Neurophysiol. doi: 10.1152/jn.00936.2016.
- ✓ Experimental Cell Research. doi.org/10.1016/j.yexcr.2018.06.020
- ✓ Front Pharmacol. doi: 10.3389/fphar.2018.00647
- ✓ Biomedicine & Pharmacotherapy. doi.org/10.1016/j.biopha.2018.07.034
- ✓ Aging. doi: 10.18632/aging.101602
- ✓ Cell Death Dis. doi: 10.1038/s41419-018-0867-4
- ✓ Genes Genomics. doi: 10.1007/s13258-020-00922-y.
- ✓ The Journal of Biological Chemistry. doi: 10.1074/jbc.RA119.010648
- ✓ Exp Ther Med. doi: 10.3892/etm.2020.8706

- ✓ Oncotarget. doi: 10.18632/oncotarget.27487
- ✓ Metabolism. doi.org/10.1016/j.metabol.2019.154013

STAT3 (ab68153, Abcam) references:

- ✓ Cell Death Dis. doi: 10.1038/s41419-019-1917-2
- ✓ EBioMedicine. doi: 10.1016/j.ebiom.2019.05.042
- ✓ Int J Nanomedicine. doi: 10.2147/IJN.S200480
- ✓ Int J Oncol. doi: 10.3892/ijo.2019.4919
- ✓ Exp Ther Med. doi: 10.3892/etm.2020.8540
- ✓ Cell Death Dis. doi: 10.1038/s41419-018-1132-6
- ✓ J Cancer. doi: 10.7150/jca.28476
- ✓ Oncol Lett. doi: 10.3892/ol.2019.10384
- ✓ Sci Rep. doi: 10.1038/s41598-018-38068-3
- ✓ Cell Death Dis. doi: 10.1038/s41419-019-1417-4

Phospho-STAT3 (ab76315, Abcam) references:

- ✓ Cell Death Dis. doi: 10.1038/s41419-019-1917-2
- ✓ Int J Nanomedicine. doi: 10.2147/IJN.S200480
- ✓ Cellular Signalling. doi.org/10.1016/j.cellsig.2019.05.010
- ✓ Int J Oncol. doi: 10.3892/ijo.2019.4919
- ✓ Sci Rep. doi: 10.1038/s41598-018-38068-3
- ✓ Cell Death Dis. doi: 10.1038/s41419-019-1417-4
- ✓ Am J Cancer Res. doi: 10.194/699-713.
- ✓ Front Immunol. doi: 10.3389/fimmu.2019.02900
- ✓ Experimental Neurology. doi.org/10.1016/j.expneurol.2020.113359
- ✓ J Cancer. doi: 10.7150/jca.42850

[Comment 4]

Figure 3 is purported to show “AIRE signaling pathway dysregulation in mTEC of mdx thymus.” The rationale for assaying the AIRE pathway is their claim that their preceding data sets showed that there was “dramatic loss of GHS-R, and defects in NFκB signaling pathways and autophagy machinery” (line 252) although that assertion was not supported by strong data (NFκB protein, not activation, was assayed). Given the shockingly high levels of expression of AIRE and SIRT1 that they show in whole tissue extracts in their blots, the authors should provide evidence to confirm the specificity of the antibodies, e.g. preabsorption of the antibodies with the antigen before repeating the blots and show the preabsorption specifically eliminates the band they identify as the antigens they are assaying.

In order to obtain quantitative data from western blots, a rigorous methodology have been used for all antibodies tested as previously described (10.1007/s12033-013-9672-6). In detail, we calculated the dilution factor of samples that is required for protein loading in the quantitative linear dynamic range for each antibody. Furthermore, we selected the most appropriate normalization method based on reference signals obtained either by housekeeping proteins after immunochemical staining or total protein intensity on blotting membranes after total protein staining. Our WB protocols were standardized considering the sensitivity and specificity of the antibody as referred in literature (in this case for AIRE and SIRT1) to assure that the reported fold changes of the target protein are not an artifact of reference signal. However, we added more WB results (n=6 animals/group) to address the point raised by this referee.

Again, this reviewer express concerns about specificity of antibodies in WB analysis. Unfortunately, author did not address this issue. The best way would be to use the thymus lysate from Aire KO or Sirt1 KO mice, however, these mice may not be easily available for authors. Instead, can authors show some reliable references in which these antibodies were used for WB analysis to detect expression of Aire and Sirt1 in the whole thymus lysates? Otherwise, authors should check the specificity by some experiments (e.g. pre-absorption) to address this issue.

The use of thymus lysates from Aire KO or Sirt1 KO mice would undoubtedly be ideal to test the specificity of the antibodies. Unfortunately, as observed by this referee, these murine models are not easily available for authors. To address the antibody specificity concerns, we found literature reporting that the AIRE antibody we used in our WB experiments was previously employed for proteomic experiments of several murine

tissues, even though not in the whole thymus lysate (10.1038/ni.3675). To avoid this bias, as suggested by referee we performed pre-absorption experiments. In details, pre-absorption was performed by incubating the AIRE primary antibody (14-5934-82, 5H12, eBioscience) with a large molar excess (10-fold) of the mouse AIRE Recombinant Protein (OPCA12022; Aviva Systems Biology Corporation) O/N at 4°C in blocking buffer. Samples from thymus and brain of C57Bl mice were resolved on polyacrylamide gels (10%) and transferred to nitrocellulose membranes (Bio-Rad Laboratories, California, USA). Filter was incubated with Ponceau solution to demonstrate the correct transfer process (A). After washing, membrane was cut into four slices. Two filters (thymus C57Bl, n=2; Brain C57Bl, n=2) were incubated with Blocking Buffer for 1h at RT. After blocking step, filters were incubated O/N at +4°C with AIRE (14-5934-82, 5H12, eBioscience) primary antibody O/N +4° in blocking buffer (1:500). After washing, membranes were incubated for 1h30' with anti-mouse secondary antibody and developed by enhanced chemiluminescence (ECL) (Amersham Biosciences, USA). Bands were visualized using an Odyssey Infrared Imaging System (Li-COR Biosciences, USA) (B and D). The remaining two filters (thymus C57Bl, n=2; Brain C57Bl, n=2) were incubated with Blocking Buffer for 1h at RT. After blocking step, filters were incubated O/N at +4°C with pre-adsorbed AIRE (14-5934-82, 5H12, eBioscience) primary antibody (1:500) in blocking buffer. After washing, membranes were incubated for 1h30' with anti-mouse secondary antibody and developed by enhanced chemiluminescence (ECL) (Amersham Biosciences, USA). Bands were visualized using an Odyssey Infrared Imaging System (Li-COR Biosciences, USA) (C and E).

Pre-absorption experiments confirmed the specificity of AIRE antibody (14-5934-82, 5H12, eBioscience).

Regarding the expression of SIRT-1 protein, according to the suggestions of the referee, we performed new WB experiments using a different antibody that recognize the amino terminus of SIRT-1 (2192247; Millipore). This antibody was previously validated on whole murine thymus as published in Nature Immunology (doi:10.1038/ni.3194). The new western blots for SIRT-1 proteins show the increase of SIRT-1 in C57Bl wild type versus dystrophic thymus samples (now shown in revised Figure 3B) confirming the previous results obtained with the antibody against the C-terminal domain of SIRT-1 (sc-74465; Santa Cruz).

[Comment 4-2]

The western blots for dystrophin proteins show the antibody is highly non-specific which would make any attempts to quantify antigens by densitometry of blots unreliable.

We used a Western blot protocol that allows sensitive and accurate dystrophin quantification in preclinical and clinical studies (10.3233/JND-150113; 10.3233/JND-180357). Moreover, the antibodies tested are the same used for dystrophin diagnostic purpose in patients followed at the Neuromuscular Center of the Policlinico Hospital of Milan (Italy). Please see as example of similar dystrophin pattern of distribution the last publication of the Jerry Mendell group in JAMA Neurology (doi:10.1001/jamaneurol.2020.1484) published online June 15, 2020.

As described by authors, there is no doubt on the specificity against human dystrophin proteins. However, how about against the mouse proteins?

At the protein level, the carboxy terminus of dystrophin protein have been shown to be highly conserved throughout human and mouse with 93% of homology for the epitope recognized by NCL-DYS2 (between amino acids 3669 and 3685). As specified in the Datasheet, the Novocastra™ Lyophilized Mouse Monoclonal Antibody Dystrophin (C-terminus) NCL-DYS2 antibody strongly cross-reacts with skeletal, cardiac and smooth muscle dystrophin from mouse, rat, rabbit, dog, chicken and hamster. Here is a brief list of references reporting the use of this antibody in mouse tissues:

- ✓ PLoS ONE. doi:10.1371/journal.pone.0002419
- ✓ Science doi: 10.1126/science.aad5143

- ✓ PLoS ONE. Doi: 10.1371/journal.pone.0230083. hal-02509545
- ✓ Nature, doi:10.1006/mthe.2002.0675
- ✓ Circ Res. doi: 10.1161/CIRCRESAHA.
- ✓ *Circ Res.* doi:CIRCRESAHA.107.162982
- ✓ Bioche and Biop Res Com, doi.org/10.1016/j.bbrc.2017.08.048
- ✓ PLoS ONE . doi:10.1371/journal.pone.0021618

[Comment 4-3]

Figure 5 is intended to show that “Adult dystrophic thymus transplantation into nude mice altered T cell development. “ However, whether transplantation altered development was not tested. The investigators showed some differences in the relative proportion of different T-cell populations in nude mice that received minced thymus from wild-type or mdx mice, but those differences may have reflected differences in the cells that occurred prior to their transplantation, not changes in development caused by transplantation. Furthermore, interpretation of the data would be facilitated if the authors reported how many cells were transplanted. If it varied between treatment groups that would affect the outcomes in subsequent experiments. Also, please report numbers of cells at each time point in figure 5b in addition to percentages, so changes in total numbers over time can be assessed. The western blot results of whole muscle extracts in figure 5c are very surprising because Tregs are exceeding few in muscle, even in fully-inflamed mdx muscle, which has made many other labs unable to detect FoxP3 in whole muscle extracts, even under pathological conditions when Treg numbers were elevated. The surprise is also increased by the finding of a strong signal for FoxP3 in the control nude mice, which do not have T-cells. Those observations support the possibility that

the band that is labeled “FoxP3” is not actually FoxP3. It will be especially important to validate this antibody and blot. It would also be more convincing to other investigators in the field if there were also immunohistochemistry data with double-labeling for FoxP3 with CD4 or CD25 to validate these proposed differences in FoxP3 in these muscles. It is also puzzling that the authors show in figure 5f that some nude mice that did not receive transplanted cells had more CD3+ and CD4+ T-cells than mice that received transplantation. How would this be possible when nude mice have no T-cells, as confirmed in Figure 5b in this investigation?

T cell maturation was tested after thymus transplantation using a published method which is based on the transplantation of minced thymus (10.1038/jid.2012.492). The limitation of this approach is related to the fact that transplantation of thymic cells does not work because thymus cannot be reproduced without stromal tissue. Supplementary Figure 6 showed the presence of transplanted thymus under the kidney capsule. The weights of the transplanted minced thymi were similar in all experiments. Raw data of FACS analysis (Figure 5B) reporting the number of cells per time point/mouse were added in the additional file. As suggested by referee n2 we performed immunostaining of Foxp3 in the muscle of transplanted animals but we failed to detect any positive cells. This was also related by the low number of infiltrating cells found in muscle sections of transplanted nude mice. Moreover, Foxp3+ Tregs were never obtained by FACS analysis from 3 independent experiments. These data are in agreement with previous observations made by Burzyn et al that isolated few amounts of Tregs from the muscle of mdx (which are more infiltrated than nude mice) (10.1016/j.cell.2013.10.054). Specificity of FoxP3 band in WB was demonstrated by using preabsorbed antibody that led to band disappearance, meaning that antibody (FJK-16s eBioscience) is well suited for detection of Foxp3 by WB analysis. To be consistent with our conclusions we added more WB results (n=4 animals/group) for Foxp3 (new Figure 5C). The percentages of infiltrating T-cells found in the muscle of nude untreated mice were less than 1% in all the experiments performed (see also Figure 9) and similar to what described for the C57Bl muscle in our previous works (10.1038/s41598-018-32613-w). As suggested by referee n1, we included in the Figures 5 and 9 a dashed line showing the percentage of CD25, CD4, CD8 found in untreated blood of nude mice as baseline. As previously published by other groups the blood of nude mouse showed T cell expression between 0.1- 2%.

This issue was not be properly addressed. This concern would be raised because this reviewer suspected specificity of the Foxp3 antibody in WB analysis. So, this reviewer also wondered why Foxp3 protein could be detected in the muscle of nude mice (PBS control). Since nude mice have a very small number of T cells, Treg cellularity should be extremely low in nude mouse. This is indeed puzzling and need to be addressed. Pre-absorption experiments may be necessary. Author describe that “Raw data of FACS analysis (Figure 5B) reporting the number of cells per time point/mouse were added in the additional file.”, but I could not find out the cell number data.

Regarding the concerns about the specificity of the Foxp3 antibody (FJK-16s) (1:500, 14-5773-82, eBioscience) we summarized a list of references using this antibody for WB:

- ✓ *Int J Mol Med.* doi: 10.3892/ijmm.2018.3618

- ✓ *J Biol Chem.* doi: 10.1074/jbc.M807322200
- ✓ *PLoS One.* doi: 10.1371/journal.pone.0007890
- ✓ *Nat Immunol.* doi: 10.1038/ni.3835

In addition, as requested, we performed pre-absorption experiments. In details, pre-absorption was performed by incubating the FOXP3 primary antibody (FJK-16s, 14-5773-82, eBioscience) with a large molar excess (10-fold) of the Mouse FOXP3 Recombinant Protein (LS-G14413-10; LifeSpan BioSciences, Inc.) O/N at 4°C in blocking buffer. Samples were resolved on polyacrylamide gels (10%) and transferred to nitrocellulose membranes (Bio-Rad Laboratories, California, USA). Filter was incubated with Ponceau solution to demonstrate the correct transfer process (A). After washing, membrane was cut into two slices. The first filter was incubated with Blocking Buffer for 1h at RT. After blocking step, filter was incubated O/N at +4°C with Foxp3 (FJK-16s, 14-5773-82, eBioscience) primary antibody O/N +4° in blocking buffer (1:500). After washing, membrane was incubated for 1h30' with anti-rat secondary antibody and developed by enhanced chemiluminescence (ECL) (Amersham Biosciences, USA). Bands were visualized using an Odyssey Infrared Imaging System (Li-COR Biosciences, USA) (B). The second filter was incubated with Blocking Buffer for 1h at RT. After blocking step, filter was incubated O/N at +4°C with pre-adsorbed Foxp3 (FJK-16s, 14-5773-82, eBioscience) primary antibody (1:500) in blocking buffer. After washing, membrane was incubated for 1h30' with anti-rat secondary antibody and developed by enhanced chemiluminescence (ECL) (Amersham Biosciences, USA). Bands were visualized using an Odyssey Infrared Imaging System (Li-COR Biosciences, USA) (C).

Pre-absorption experiments confirmed the specificity of Foxp3 antibody (FJK-16s, 14-5773-82, eBioscience).

Although we proved the Foxp3 antibody specificity, we agree with both reviewers that Treg cellularity should be extremely low in nude mouse thus no expression of Foxp3 should be expected in WBs. The presence of faint bands on the WBs of nude PBS control mice might depend on the BALB/c strain that presents high amount of Tregs expressing Foxp3 compared to other strain of mice such as NOD strain (doi.org/10.1016/j.imlet.2020.04.006). Furthermore, we cannot exclude Foxp3 expression on other cell populations as endothelial cells and macrophages, as previously described (DOI 10.1007/s00262-014-1581-4; doi:10.1038/mt.2013.219; doi: 10.4049/jimmunol.180.8.5163). To address this issue, we have performed Real Time experiments that confirmed significant differences in *Foxp3* expression between experimental groups as already outlined in the manuscript (nude+PBS vs. nude+MDX: $p=0.0058$; nude+MDX vs. nude+C57Bl: $p=0.0067$ with One-way ordinary ANOVA). Of note, RT-PCR experiments highlight hardly detectable *Foxp3* expression in muscles of nude mice treated with PBS.

Nevertheless, the interpretation of these data remains misunderstanding leading us to decide to exclude the Foxp3 WB experiments in the revised manuscript. Finally, we included in the Raw data of FACS analysis of Figure 5B the number of cells per μl that was calculated as previously described (DOI: [10.1002/cyto.b.20500](https://doi.org/10.1002/cyto.b.20500)). Briefly, the percentages of CD25+CD3+ or CD4+CD3+ or CD8+CD3+ T-cells were multiplied by the absolute lymphocyte count derived from the automated hematology analyzer (calculated as the product of the white blood cell count (WBC) and the lymphocyte percentage).

[Comment 4-3]

Data in Figure 6D is presented as a histogram with bars, instead of a scatter diagram. Please clarify what the data sets are in the raw data file for Figure 6D. It appears that 3 mice in each group were analyzed (T1, T2, T3)? Which mice were selected for analysis and what were the selection criteria? *In the new version of the manuscript (and in the Raw data) we add new counts in the graph of Figure 6D. We also specified the number of animals per group in the Raw data section and how we identified the different MyHC isoforms. In the new Figure 6D, the graph is now presented as a scatter diagram.*

I did not find the scatter graph of Fig. 6D.

Please accept our apologize for that misunderstanding. The scatter graph of Fig. 6D is now reported in the new version of Figure 6.

[Comment 4-4]

Suppl Figure 2 purportedly shows an increase in fibrosis with a western (n= 2) of dubious quality showing an increase in collagen 1. [“statistically significant over-expression of collagen I “; line 379]. However, that increase in the proportion of the whole muscle that was comprised of collagen could just be secondary to a decrease in fiber size since controlled for mass loaded on gel for blot. Thus, there are no data

to show that collagen over-expression occurred.

As suggested, we included in the Suppl Figure 2 new representative WB quantification of collagen I and OPN. Ponceau S staining of the proteins transferred to the membrane confirmed similar amounts of proteins between specimens (see below). Results in the text are expressed as the ratio of relative intensity of collagen I and OPN protein expression normalized to GAPDH as a loading control.

Data showed no significant or clear evidence that Col1 and OPN could be increased in the muscle of the MDX-transferred mice. Moreover, at least to me, intensities of some major bands visualized by Ponceau S staining shows a similar pattern with WB bands of Coll1 than GAPDH. I recommend authors to modify this part.

As suggested, we modified the description of Supplementary figure 2 in the new version of the Manuscript as follow: “Collagen I and the idiopathic marker of fibrosis osteopontin (OPN) protein levels of TAs of Tnu^{MDX} mice were not statistically different when compared to TAs of Tnu^{C57Bl} and nude^{PBS} mice (Figure S2B). However, RT-qPCR analysis of TAs revealed an over-expression of collagen 3a^{42,43} in Tnu^{MDX} related to Tnu^{C57Bl} (P=0.0012) and nude^{PBS} mice (P=0.0211) (Figure S2C)”.

REVIEWERS' COMMENTS

Reviewer #2 (Remarks to the Author):

Thank you for the responses. Authors addressed all concerns raised by reviewers.

No other requests were added by the Reviewer #2

REVIEWERS' COMMENTS

Reviewer #2 (Remarks to the Author):

Thank you for the responses. Authors addressed all concerns raised by reviewers.